# Replicable Distribution Testing

Ilias Diakonikolas
University of Wisconsin-Madison
ilias@cs.wisc.edu

Jingyi Gao
University of Wisconsin-Madison
jingyig@cs.wisc.edu

Daniel M. Kane
University of California, San Diego
dakane@cs.ucsd.edu

Sihan Liu
University of California, San Diego
La Jolla, CA
sil046@ucsd.edu

Christopher Ye
University of California, San Diego
La Jolla, CA
czye@ucsd.edu

## Abstract

We initiate a systematic investigation of distribution testing in the framework of algorithmic replicability. Specifically, given independent samples from a collection of probability distributions, the goal is to characterize the sample complexity of replicably testing natural properties of the underlying distributions. On the algorithmic front, we develop new replicable algorithms for testing closeness and independence of discrete distributions. On the lower bound front, we develop a new methodology for proving sample complexity lower bounds for replicable testing that may be of broader interest. As an application of our technique, we establish near-optimal sample complexity lower bounds for replicable uniformity testing—answering an open question from prior work—and closeness testing.

## 1   Introduction

Algorithmic replicability has emerged as a fundamental notion in modern statistics and machine learning to ensure consistency of algorithm outputs in the presence of randomness in input datasets. The formal notion of replicability, proposed in Impagliazzo et al. [2022], is as follows.

**Definition 1.1** (Replicability Impagliazzo et al. [2022])**.** *A randomized algorithm* $\mathcal{A} : \mathcal{X}^n \mapsto \mathcal{Y}$ *is* $\rho$-*replicable if for all distributions* $\mathbf{p}$ *on* $\mathcal{X}$, $\mathrm{Pr}_{r,T,T'}\left(\mathcal{A}(T;r) = \mathcal{A}(T';r)\right) \geq 1 - \rho$, *where* $T, T'$ *are i.i.d. samples taken from* $\mathbf{p}$, *and* $r$ *denotes the internal randomness of the algorithm* $\mathcal{A}$.

Since its introduction, replicability has been considered in the context of a wide range of machine learning tasks, including multi-arm bandits Esfandiari et al. [2022], clustering Esfandiari et al. [2023], reinforcement learning Karbasi et al. [2023], Eaton et al. [2023], halfspace learning Kalavasis et al. [2024], and high-dimension statistics Hopkins et al. [2024]. A related line of work explored the connection between replicability and other algorithmic stability notions such as differential privacy Bun et al. [2023a], Moran et al. [2023], total variation indistinguishability Kalavasis et al. [2023], global stability Chase et al. [2023], and one-way perfect generalization Bun et al. [2023a], Moran et al. [2023].

In this work, we initiate a systematic study of replicability in distribution testing, a central area in property testing and statistics that aims to ascertain whether an unknown distribution satisfies a certain property or is "far" from satisfying that property. Specifically, we focus on replicable testing of discrete distributions in total variation distance, which encompasses canonical problems such as uniformity/identity testing, closeness testing, and independence testing. Formally, we have:

39th Conference on Neural Information Processing Systems (NeurIPS 2025).

**Definition 1.2** (($\epsilon, \rho$)-replicable testing of property $\mathcal{P}$). *Let $\mathcal{P}$ be a property consisting of $k$-tuples of distributions, and $\epsilon, \rho \in (0, 1/4)$. Given sample access to a collection of distributions $\mathbf{p}^{(1)}, \cdots, \mathbf{p}^{(k)}$, we say a randomized algorithm $\mathcal{A}$ solves $(\epsilon, \rho)$-replicable $\mathcal{P}$-testing if $\mathcal{A}$ is $\rho$-replicable and can distinguish between the following cases with probability at least $1 - \rho$: completeness case $(\mathbf{p}^{(1)}, \cdots, \mathbf{p}^{(k)}) \in \mathcal{P}$ or soundness case $k^{-1} \sum_{i=1}^{k} d_{TV}(\mathbf{p}^{(i)}, \mathbf{q}^{(i)}) \geq \varepsilon$ for all tuples $(\mathbf{q}^{(1)}, \cdots, \mathbf{q}^{(k)}) \in \mathcal{P}$. In particular, (a) uniformity testing over the domain $[n]$ corresponds to the case that $k = 1$, and $\mathcal{P}$ consists of only the uniform distribution over $[n]$; (b) closeness testing over the domain $[n]$ corresponds to the case that $k = 2$, and $\mathcal{P}$ consists of all pairs of distributions $(\mathbf{p}, \mathbf{p})$ over $[n]$; (c) independence testing over domain $[n_1] \times [n_2]$ corresponds to the case that $k = 1$, and $\mathcal{P}$ consists of all product distributions over that domain.*

After the pioneering works formulating this field Batu et al. [2000], Goldreich and Ron [2011] from a TCS perspective, a line of work has given efficient testers (without the replicability requirement) achieving information theoretically optimal sample complexities for the aforementioned problems; see Paninski [2008], Valiant and Valiant [2017] for uniformity/identity testing, Batu et al. [2000], Valiant [2008], Batu et al. [2013], Chan et al. [2014] for closeness testing, and Batu et al. [2001], Acharya et al. [2015], Diakonikolas and Kane [2016] for independence testing. More broadly, substantial progress has been made on testing a wide range of natural properties; see, e.g., Batu et al. [2002, 2004], Acharya et al. [2011], Levi et al. [2011], Diakonikolas et al. [2015], Canonne et al. [2017], Daskalakis et al. [2018], Canonne et al. [2018, 2022b,a] for a sample of works, and Rubinfeld [2012], Canonne [2020, 2022] for surveys on the topic.

Despite the maturity of the field of distribution testing, the relevant literature contains only a single prior work addressing replicability: a recent paper by Liu and Ye [2024] designs replicable testers for the task of uniformity testing (and identity testing via a standard reduction technique), and demonstrates that the sample complexity is nearly tight within a *restricted* class of algorithms. Specifically, their techniques are insufficient to establish lower bounds against *general* uniformity testers, or to design sample-optimal replicable testers for other distribution testing problems.

An instructive parallel arises in the context of differentially private (DP) distribution testing, where similar challenges in designing testers with extra stability constraints have been addressed; see Cai et al. [2017], Aliakbarpour et al. [2018], Acharya et al. [2018], Aliakbarpour et al. [2019]. Although generic reductions from DP to replicable algorithms exist, see, e.g., Bun et al. [2023b], they incur polynomial overheads in sample complexity. This motivates our goal: to develop a principled and fine-grained understanding of the sample complexity cost of replicability in distribution testing.

## 1.1 Our results

Our first main contribution is a new lower bound framework for replicable distribution testing that yields unconditional lower bounds against all testers, without additional assumptions. As a first application of our framework, we show that the replicable uniformity tester proposed in Liu and Ye [2024] is indeed nearly optimal, thus settling the main open problem left by their work.

**Theorem 1.3** (Sample Complexity of Replicable Uniformity Testing). *The sample complexity of $(\varepsilon, \rho)$-replicable uniformity testing over $[n]$ is $\tilde{\Theta}(\sqrt{n}\varepsilon^{-2}\rho^{-1} + \varepsilon^{-2}\rho^{-2})$.*

We believe that the framework is broadly applicable to establishing lower bounds for other replicable distribution testing problems. In particular, as an additional application, we derive the following tight lower bound for replicable closeness testing.

**Theorem 1.4** (Lower Bound of Replicable Closeness Testing). *The sample complexity of $(\varepsilon, \rho)$-replicable closeness testing over $[n]$ is at least $\tilde{\Omega}(n^{2/3}\varepsilon^{-4/3}\rho^{-2/3} + \sqrt{n}\varepsilon^{-2}\rho^{-1} + \varepsilon^{-2}\rho^{-2})$.*

On the algorithmic front, we provide new replicable testers for closeness and independence testing. For closeneness testing, we show:

**Theorem 1.5** (Replicable Closeness Tester). *The sample complexity of $(\varepsilon, \rho)$-replicable closeness testing over $[n]$ is at most $\tilde{O}(n^{2/3}\varepsilon^{-4/3}\rho^{-2/3} + \sqrt{n}\varepsilon^{-2}\rho^{-1} + \varepsilon^{-2}\rho^{-2})$.*

Note that Theorem 1.5, together with Theorem 1.4, give a tight characterization of the sample complexity of replicable closeness testing up to polylogarithmic factors. For independence testing, we show that:

**Theorem 1.6** (Replicable Independence Tester). *The sample complexity of $(\varepsilon, \rho)$-replicable independence testing over $[n_1] \times [n_2]$ for $n_1 \geq n_2$ is at most $\tilde{O}\left(\frac{n_1^{2/3} n_2^{1/3}}{\rho^{2/3} \varepsilon^{4/3}} + \frac{\sqrt{n_1 n_2}}{\rho \varepsilon^2} + \frac{1}{\varepsilon^2 \rho^2}\right)$.* [1]

Perhaps surprisingly, our upper bounds point to an intriguing conceptual connection between replicability and distribution testing in the *high success probability regime*. In particular, the functional forms of the sample complexities of (non-replicable) uniformity, closeness, and independence testing up to error probability $\delta$ have been characterized in Diakonikolas et al. [2018, 2021] to be precisely the sample complexity upper bounds of the corresponding replicable testing problems after replacing $\rho^{-1}$ with $\sqrt{\log(1/\delta)}$ [2]. Moreover, all known replicable distribution testers (including ours) leverage the statistics developed in the context of high probability distribution testers. We leave it as an interesting open problem whether there exists generic reduction from high-probability testers to replicable ones or vice versa. Lastly, with this connection in mind, it is a plausible conjecture that our sample complexity upper bound for replicable independence testing is nearly optimal. We leave this as an open question.

### 1.2 Technical Overview

We start with a description of our lower bound framework, which is the main technical contribution of this work, followed by our upper bounds.

**Replicable Testing Lower Bounds** In what follows, we will sketch the overall framework for showing lower bounds against replicable uniformity and closeness testing, and point to the specific lemmas used in deriving the uniformity testing lower bound for concreteness. Let $\mathcal{A}$ be a randomized tester for the testing problem that uses significantly fewer samples than the target lower bound. Our end goal is to show that if $\mathcal{A}$ satisfies the correctness requirements of the corresponding testing problem, then $\mathcal{A}$ cannot be replicable. Towards this goal, we begin with the same reasoning steps as the ones in Liu and Ye [2024]. In particular, we construct a meta-distribution $\mathcal{M}_\xi$, parametrized by $\xi \in [0, \varepsilon]$, over potential hard instances of the testing problem such that (i) $\mathcal{M}_0$ and $\mathcal{M}_\varepsilon$ correspond to instances that should be respectively accepted and rejected by the tester, and (ii) it should be hard to distinguish a random instance from $\mathcal{M}_\xi$ versus a random instance from $\mathcal{M}_{\xi + \varepsilon\rho}$.

After that, using the same argument as Liu and Ye [2024], we can deduce that if we sample $\xi \sim \mathcal{U}([0, \varepsilon])$, then the average acceptance probability of the tester under $\mathcal{M}_\xi$, i.e., $\mathbb{E}_{\mathbf{p} \sim \mathcal{M}_\xi}[\Pr_{S \sim \mathbf{p}}[\mathcal{A} \text{ accepts } S]]$, will be close to $1/2$ with probability at least $\Omega(\rho)$ over the randomness of $\xi$. See Lemma 3.5 for the formal statement. If $\mathcal{M}_\xi$ were to contain just a single distribution instance $\mathbf{p}_\xi$, then the statement would directly imply that $\mathcal{A}$ is not replicable under $\mathbf{p}_\xi$, and this would conclude the lower bound argument. Of course, $\mathcal{M}_\xi$ is in reality a meta-distribution over (exponentially) many different instances by design (see Definition 3.2). To overcome this issue, Liu and Ye [2024] takes advantage of the fact that the instances from the meta-distribution are identical up to permutation of the domain elements. As such, if one makes the *additional assumption* that the output of the tester is invariant up to domain relabeling (in other words, the tester is *symmetric*), then it is not hard to show that the acceptance probability of the tester under each individual distribution must be the same as the overall averaged acceptance probability under $\mathcal{M}_\xi$, and the proof is complete.

Our proof circumvents this difficulty with a fundamentally different approach that allows us to avoid making *any* assumptions on the tester. As one of our main technical contributions, we show that even when the tester is *not* symmetric, the acceptance probability under a random choice of $\mathbf{p} \sim \mathcal{M}_\xi$ must nonetheless concentrate around its expectation, as long as the tester is still moderately replicable under $\mathbf{p}$, i.e., replicable with probability $1 - 1/\text{polylog}(n)$. See Lemma 3.6 for the formal statement. Towards this goal, consider the joint distribution of two random sample sets $S, S'$ generated as follows: pick a random distribution $\mathbf{p} \sim \mathcal{M}_\xi$ and then sample $S, S'$ independently from $\mathbf{p}^{\otimes m}$. The distribution of $S'$ conditioned on $S$ then naturally defines a random walk $\mathbf{RW}_\xi$ on the space of all possible sample sets. For convenience, denote by $\mathbf{RW}_\xi^k(\mathbf{p})$ the distribution over sample sets $\bar{S}$ obtained by first sampling $T \sim \mathbf{p}$ and then performing $k$ steps of the random walk. Lying in the heart

---

[1] While our replicable closeness tester runs in linear time in sample size, our replicable independence tester requires polynomial runtime. This is due to an extra "averaging" operation applied to make the statistic more stable (see Section 1.2). We leave it for future work to explore whether its runtime can be further improved.

[2] For example, the sample complexity of high probability closeness testing has been shown to be $\Theta\left(n^{2/3} \varepsilon^{-4/3} \log^{1/3}(1/\delta) + \sqrt{n} \varepsilon^{-2} \sqrt{\log(1/\delta)} + \varepsilon^{-2} \log(1/\delta)\right)$ by Diakonikolas et al. [2021]

of our proof is the following two structural claims: (1) for most $\mathbf{p} \sim \mathcal{M}_\xi$, the acceptance probability of the tester under $\mathbf{RW}_\xi^{\mathrm{polylog}(n)}(\mathbf{p})$ is roughly the same as that of under $\mathbf{p}$. (2) the random walk $\mathbf{RW}_\xi$ has mixing time at most $\mathrm{polylog}(n)$. Combining the two claims gives that the acceptance probability under most $\mathbf{p} \sim \mathcal{M}_\xi$ must be roughly the same, as the acceptance probability under the stationary distribution of the random walk (which by construction is exactly equal to the expected acceptance probability under $\mathcal{M}_\xi$).

It then remains for us to establish these two claims. The proof of (1) mainly follows from the definition of the random walk, and the assumption that the tester is moderately replicable. See Lemma 3.9 and its proof for details. The canonical way for showing (2) is to bound from below the eigenvalue gaps of the transition matrix of the random walk. To analyze this, we note that after a careful use of Poissonization (see Definition D.1 for the definition of Poisson sampling), we can make $\mathbf{RW}_\xi$ a product of $n$ independent random walks. Formally, since the number of samples is Poissonized, the sample frequency of each bucket is independent, even conditioned on the choice of distribution $\mathbf{p} \sim \mathcal{M}_\xi$. It then suffices for us to bound the mixing time of much simpler random walks on the sample frequencies of each individual domain element. Fortunately, the eigenvalue gap of this random walk can be analyzed conveniently using elementary properties of the Poisson distributions. See Lemma 3.8 and its proof for details. The formal proofs of Theorem 1.3 and the relevant lemmas are deferred to Appendix E.

Lastly, the same framework also applies to the proof of Theorem 1.5. The main change needed is to replace the meta-distribution $\mathcal{M}_\xi$ to be the standard hard instance for closeness testing. See, e.g., Valiant [2008], Chan et al. [2014], Diakonikolas and Kane [2016] for the construction of the hard instance. The formal proof can be found in Appendix F.

**Replicable Testing Upper Bounds** We begin with the observation that many testers from the literature share the following nice form: compute a test statistic $Z$ and compare it to a threshold $R$. Usually, the analysis (without replicability requirements) involves showing that $\mathbb{E}[Z] = 0$ in the completeness case, $\mathbb{E}[Z] \gg R$ in the soundness case, and $\mathrm{Var}[Z]$ is at most a small constant multiple of $R^2$. For testers of this form, we can employ the same strategy as the one used in Liu and Ye [2024] to transform them into replicable testers: we can compute the same test statistic $Z$, and then compare it to a *randomly chosen* threshold $r$ between $0$ and $R/2$. In particular, if we further have that $\mathrm{Var}[Z] \ll R^2\rho^2$ (at the cost of taking more samples), the variance bound on $Z$ implies that $Z$ computed with different sample sets drawn from the same underlying distribution are likely to be close to each other. Consequently, the values of $Z$ in two runs are unlikely to be separated by a randomly chosen $r$, ensuring replicability.

For closeness testing, the high probability tester from Diakonikolas et al. [2021] satisfies exactly the conditions needed, and in turn yields our replicable tester after combining it with the random thresholding strategy. The formal proof of Theorem 1.5 is given in Appendix C.

Designing good testers for replicable independence testing turns out to be significantly more involved, as even the known high probability independence testers do not satisfy the required variance bounds within our sample complexity budget. In its essence, the bottleneck lies in an extra randomized "flattening" procedure employed by the tester, which significantly increases the overall variance of the final statistic computed. Specifically, the procedure utilizes a random subset of the input samples to "split" domain elements with large mass into sub-elements. This step aims to ensure that there will be no extremely heavy elements after the procedure (otherwise, the tester may fail to satisfy even the basic correctness requirements). To show correctness of their tester, Diakonikolas et al. [2021] demonstrated that (1) the flattening procedure preserves the product/non-product structure of the original distribution, and (2) the variance of the final test statistic *conditioned on* the flattening samples (and some other technical conditions) is small. Notably, a bound on the total variance of the test statistic $Z$ is not needed in their context, as the above two properties suffice for them to show upper/lower bounds on $Z$ in the completeness/soundness cases. Yet, when replicability is of concern, we do need to show that $Z$ concentrates around a small interval. As a result, the lack of a good bound on the total variance (as compared to just conditional variance) of the final test statistic turns out to be a major technical obstacle in converting the high probability tester into a replicable one. In fact, the randomness in using different samples for flattening purposes can easily cause the total variance of $Z$ to be much larger than the conditional variance.

To overcome this difficulty, we leverage the following idea from Aliakbarpour et al. [2019] (in the context of differentially private testing): to make a test statistic computed with internal randomness more stable, we can replace it with the *averaged* version of it. In particular, we apply this idea to the statistic $Z$ computed by the high probability independence statistic, and obtain a new averaged statistic $Z_a$—essentially, the expected value of $Z$ averaged over all possible partitions of the input samples into flattening samples and testing samples (see Definition 2.6 for the formal definition). As our main technical lemma, we show that the total variance of this averaged test statistic $Z_a$ can be bounded from above by $N$–the expected value of the number of *non-singleton* samples, i.e., the testing samples which still collide with another testing sample after the flattening procedure (see Lemma 2.9). At a high level, our argument uses an Efron-Stein style inequality that bounds the variance by the sum of the expected square differences of the test statistic $Z_a$ caused by removal of each individual sample. Suppose that there are in total $m$ samples and the probability of selecting a sample for flattening is $p$. We then proceed by a case analysis. If the sample removed is used for computing the final test statistic, we show that the (non-averaged) test statistic $Z$ will only be different if the sample also happens to be a singleton sample after flattening, which happens with probability roughly $O(N/m)$. If the sample is selected for flattening, we show that removing it can change the test statistic $Z$ by at most $N$ divided by the number of flattening samples, which is roughly $O(pm)$. Consequently, the contribution to the variance of $Z_a$ in this case is at most $(p\, N/(mp))^2 \leq N^2/m^2$, which is also $O(N/m)$.

It remains for us to control the non-singleton sample count $N$. Fortunately, Diakonikolas et al. [2021] already established sharp bounds on the expected value of $N$, when $\mathbf{p}$ is known to be a product distribution. This then motivates us to run a pre-test to check whether $\mathbb{E}[N]$ is within a constant factor of the desired bound, before computing the averaged independence statistic $Z_a$. In particular, we consider the statistic $N_a$, defined similarly to $Z_a$ as the expected non-singleton sample count $N$ averaged over the random choice of the flattening sample set, and use an almost identical argument to show that $\mathrm{Var}[N_a]$ can also be bounded by $O(\mathbb{E}[N])$ (see Lemma 2.11). Equipped with the variance bound, it is not hard to show that comparing $N_a$ with an appropriately chosen random threshold yields a tester that replicably determines whether the magnitude of $\mathbb{E}[N]$ is within a constant factor of the bound it should satisfy when $\mathbf{p}$ is a product distribution. If we pass this test, we can then proceed to apply the main test, which compares $Z_a$ to a randomized threshold. This concludes our proof sketch. The relevant lemma statements can be found in Section 2. The proofs of Theorem 1.6 and the relevant lemmas are deferred to Appendix B.

**Preliminaries** Let $[n] = \{1, \ldots, n\}$. We use $n$ to denote domain size and $m$ to denote sample complexity. We use bold letters (e.g. $\mathbf{p}, \mathbf{q}$) to denote distributions or measures and $\mathbf{p}(i)$ to denote the mass of $i$ under $\mathbf{p}$. Let $\mathrm{Poi}(\lambda)$ denote a Poisson distribution with parameter $\lambda$ and $\mathrm{PoiS}(m, \mathbf{p})$ denote $m' \sim \mathrm{Poi}(m)$ i.i.d. samples from $\mathbf{p}$. Let $\mathrm{Bernoulli}(\alpha)$ denote a Bernoulli distribution with parameter $\alpha$. Let $\mathcal{U}(S)$ denote the uniform distribution over set $S$, where $S$ can be either a discrete set of points or an interval. For any distribution $\mathbf{p}$, let $\mathbf{p}^{\otimes m}$ denote $m$ i.i.d. samples from $\mathbf{p}$. We use "algorithm" and "tester" interchangeably. For a multiset $S$ of samples, we denote the set of all elements appearing in $S$ by $supp(S)$.

## 2 Replicable Independence Testing Algorithm

In this section, we give our replicable independence tester. At a high level, we compute the same statistic used by the high probability independence tester from Diakonikolas et al. [2021], but average over the internal randomness of the tester to enhance replicability.

Our starting point is the (randomized) flattening technique developed in Diakonikolas and Kane [2016], Diakonikolas et al. [2021] that helps decrease the $\ell_2$ norm of input distributions while maintaining the properties to be tested in total variation distance. The original description is as follows. First, one draws a set of samples $X$, and randomly partitions $X$ into a flattening sample set, and a testing sample set. Next, one uses the flattening samples to determine the number of sub-bins for each original domain element, and then randomly assigns original testing samples to the sub-bins. For our analysis, it is more convenient to consider an equivalent random process, where we randomly sort all samples, partition them into flattening and testing samples, and make two testing samples be in the same sub-bin if and only if they are originally from the same bin and there are no flattening samples from the same bin between them. The formal description is as follows.

**Definition 2.1.** *Let $X = \{X_1, \cdots, X_m\}$ be a multiset of samples over $[n]$, and $F \in \{0,1\}^m$ be a binary vector. Then the randomized flattening procedure $X^f := \{X_\ell^f\}_{\ell:F_\ell=0} \leftarrow Flatten(\{X_\ell\}_{\ell=1}^m; F)$ is as follows. (1) Assign a random order $\sigma$ to the samples. (2) For each sample $X_\ell$, count the number of samples $X_{\ell'}$ before it according to $\sigma$ such that $X_{\ell'} = X_\ell$ and $F_{\ell'} = 1$. Denote the number as $f_\ell$. (3) For each $\ell$ such that $F_\ell = 0$, set $X_\ell^f \leftarrow (X_\ell, f_\ell)$. Moreover, given a parameter $\alpha \in (0,1)$, we denote by $Flatten(X; \alpha)$ the randomized sample set obtained from $X^f \leftarrow Flatten(X; F)$, where $F \sim Bernoulli(\alpha)^{\otimes m}$.*

In independence testing, we need to perform the flattening operation on the marginals of multi-dimensional distribution independently. For clarity, we formalize this operation below.

**Definition 2.2.** *Let $\alpha, \beta \in (0,1)$, and $P = \{P_\ell = (X_\ell, Y_\ell)\}_{\ell=1}^m$ be a multiset of samples over $[n_1] \times [n_2]$. Then the multi-dimensional flattening operation $P^f := \{P_\ell^f\} \leftarrow Flatten(\{P_\ell\}_{\ell=1}^m; \alpha, \beta)$ is as follows. (1) Choose $F^x \sim Bernoulli(\alpha)^{\otimes m}$ and $F^y \sim Bernoulli(\beta)^{\otimes m}$. (2) $\{X_\ell^f\}_{\ell:F_\ell^x=0} \leftarrow Flatten(\{X_\ell\}_{\ell=1}^m; F^x)$, $\{Y_\ell^f\}_{\ell:F_\ell^y=0} \leftarrow Flatten(\{Y_\ell\}_{\ell=1}^m; F^y)$. (3) Map $P_\ell$ to $P_\ell^f \leftarrow (X_\ell^f, Y_\ell^f)$ if $F_\ell^x = F_\ell^y = 0$. When flattening two bags of samples $A$ and $B$ together, we denote by $A^f \cup B^f \leftarrow Flatten(A \cup B; \alpha, \beta)$, where $A^f(B^f, resp.)$ contains all elements mapped from $A(B, resp.)$.*

Another key idea behind the tester from Diakonikolas et al. [2021] is to use the samples from $\mathbf{p}$ to simulate samples from another product distribution $\mathbf{q}$ that equals to the product of the marginals of $\mathbf{p}$. In particular, a sample from $\mathbf{q}$ can be simulated by taking two samples from $\mathbf{p}$, and combining the first coordinate of the first sample to the second coordinate of the second sample. Hence, we can readily assume that we have sample access to both $\mathbf{p}$ and $\mathbf{q}$.

**Definition 2.3** (Product of Marginals). *Given a distribution $\mathbf{p}$ over $[n_1] \times [n_2]$, we say $\mathbf{q}$ is the product of marginals of $\mathbf{p}$ if the marginals of $\mathbf{q}$ agree with that of $\mathbf{p}$ and $\mathbf{q}$ is a product distribution.*

Given the samples from the original distribution, and the ones from the product of the marginals, the final step of the tester from Diakonikolas et al. [2021] is to compute the closeness test statistic, which we reiterate below.

**Definition 2.4** (Closeness Statistic). *Given two bags of samples $S_p, S_q$ over some finite discrete domain, the closeness statistic $Z_C(S_p, S_q)$ is defined as follows. (1) For each sample in $S_p \cup S_q$, mark it independently with probability $1/2$. (2) For $i \in supp\,(S_p \cup S_q)$, let $T_i^{p_0}, T_i^{q_0}$ be the number of times the element $i$ appears marked in $S_p, S_q$, and $T_i^{p_1}, T_i^{q_1}$ be the corresponding counts of the unmarked samples. (3) Compute $Z_C(S_p, S_q) \leftarrow |T_i^{p_0} - T_i^{q_0}| + |T_i^{p_1} - T_i^{q_1}| - |T_i^{p_0} - T_i^{p_1}| - |T_i^{q_0} - T_i^{q_1}|$.*

A useful fact of this test statistic is that any singleton sample does not contribute to its value.

**Fact 1.** *Consider two sets of samples $S_p, S_q$ over some finite discrete domain. Assume that $P$ is a singleton sample among $S_p \cup S_q$. It holds that $\mathbb{E}[Z_C(S_p, S_q)] = \mathbb{E}[Z_C(S_p \backslash \{P\}, S_q \backslash \{P\})]$, where the randomness is over the internal randomness of the test statistic $Z$.*

We are now ready to state the tester from Diakonikolas et al. [2021], which forms the building block of our replicable independence tester.

---

**Algorithm 1** INDEPENDENCESTATS

    **Input:** a sample set $S_p$ from the unknown distribution $\mathbf{p}$ over $[n_1] \times [n_2]$, where $n_1 \geq n_2$, and another sample set $S_q$ from $\mathbf{q}$, the product of marginals of $\mathbf{p}$.
    **Parameters:** domain sizes $n_1 > n_2$ , tolerance $\epsilon \in (0, 1/4)$ , replicability $\rho \in (0, 1/4)$.
    **Output:** A test statistic related to whether these samples came from an independent distribution.

1: Set $m = \tilde{\Theta}\left(n_1^{2/3} n_2^{1/3} \rho^{-2/3} \varepsilon^{-4/3} + \sqrt{n_1 n_2} \rho^{-1} \varepsilon^{-2} + \rho^{-2} \varepsilon^{-2}\right)$.
2: Set $\alpha = \min(n_1/(100m), 1/100)$, $\beta = n_2/(100m)$.
3: Compute the flattened samples $S_p^f \cup S_q^f \leftarrow$ Flatten $(S_p \cup S_q; \alpha, \beta)$.
4: Abort and return 0 if $|S_p| - |S_p^f| > 10n_1$ or $|S_q| - |S_q^f| > 10n_2$.
5: Sample $\ell, \ell' \sim$ Poi$(m)$. Abort and return 0 if $\ell > |S_p^f|$ or $\ell' > |S_q^f|$.
6: Keep only the first $\ell$ samples of $S_p^f$ and only the first $\ell'$ samples of $S_q^f$.
7: Compute and return the closeness test statistic $Z_C(S_p^f, S_q^f)$.

---

A basic property we need is that the final test statistic computed has a wide expectation gap.

**Lemma 2.5** (Expectation Gap of Independence Statistics; Section 3.2 and Claim 4.14 of Diakonikolas et al. [2021]). *Let $\mathbf{p}$ be some unknown distribution over $[n_1] \times [n_2]$, $\mathbf{q}$ be the product of marginals of $\mathbf{p}$, and $m$ be defined as in Line 1 of* INDEPENDENCESTATS. *Let $S_p, S_q$ be samples from $\mathbf{p}, \mathbf{q}$ respectively with size $|S_p| = |S_q| = 100m$, and $Z \leftarrow$* INDEPENDENCESTATS$(S_p, S_q)$. *Define $G := \min\left(\varepsilon m, m^2 \varepsilon^2/(n_1 n_2), m^{3/2}\varepsilon^2/\sqrt{n_1 n_2}\right)$ If $\mathbf{p}$ is a product distribution, then $\mathbb{E}[Z] \leq C_{I_1} G$; If $\mathbf{p}$ is $\varepsilon$-far from any product distribution in TV distance, then $\mathbb{E}[Z] > C_{I_2} G$ for some constants $C_{I_1} < C_{I_2}$.*

To make the final test statistic computed by INDEPENDENCESTATS more replicable, we consider a new test statistic $Z_a$ computed by averaging over the internal randomness of INDEPENDENCESTATS.

**Definition 2.6** (Averaged Independence Statistic). *Let $S_p, S_q$ be samples over $[n_1] \times [n_2]$. We define $Z_a(S_p, S_q)$ to be the expected value of* INDEPENDENCESTATS $(S_p, S_q)$ *averaged over the internal randomness of* INDEPENDENCESTATS.

An important quantity in analyzing the concentration of the above new statistic is the concept of non-singleton sample counts.

**Definition 2.7** (Non-singleton Sample Count). *Let $S$ be a set of samples over some finite discrete domain. We define the non-singleton sample count $N(S)$ as the total number of samples within $S$ that collide with another sample.*

Specifically, we focus on the non-singleton sample count of the flattened sample sets $S_p^f, S_q^f$ constructed by INDEPENDENCESTATS.

**Definition 2.8** (Non-singleton Sample Count of Flattened Samples). *Let $S_p, S_q$ be two arbitrary sets of samples over $[n_1] \times [n_2]$. Consider the two random (truncated) flattened sample sets $S_p^f, S_q^f$ constructed by* INDEPENDENCESTATS$(S_p, S_q)$ *on Line 7 [3]. We define $N_a(S_p, S_q) := \mathbb{E}\left[N(S_p^f \cup S_q^f)\right]$, where the expectation is over the internal randomness of* INDEPENDENCESTATS.

Our main insight is that we can bound the variance of the new statistic $Z_a(S_p, S_q)$ by the expected value of $N_a(S_p, S_q)$. This key technical lemma is as follows. The proof is deferred to Appendix B.

**Lemma 2.9** (Bound Variance of Averaged Independence Statistic in Expected Non-Singleton Sample Count). *Let $\mathbf{p}$ be a distribution over $[n_1] \times [n_2]$, and $\mathbf{q}$ be the product of marginals of $\mathbf{p}$. Let $S_p, S_q$ be samples from $\mathbf{p}, \mathbf{q}$ respectively. Consider the averaged independence statistics $Z_a(S_p, S_q)$, and the averaged non-singleton sample count $N_a(S_p, S_q)$. Then it holds that $\mathrm{Var}[Z_a(S_p, S_q)] \leq O(\log^3(n_1 n_2))\mathbb{E}[N_a(S_p, S_q)]$, where the randomness is over the samples $S_p, S_q$.*

It then remains for us to control $N_a(S_p, S_q)$. Fortunately, the expected value of the non-singleton count has already been shown to be small by Diakonikolas et al. [2021] when the underlying distribution $\mathbf{p}$ is known to be a product distribution.

**Lemma 2.10** (Expected Non-singleton Sample Count under Product Distribution, Lemma 4.9 of Diakonikolas et al. [2021]). *Let $m \in \mathbb{Z}_+, \alpha, \beta \in (0, 1)$ be defined as in Line 1 and Line 2 from* INDEPENDENCESTATS *respectively. Let $S$ be samples from a product distribution $\mathbf{q}$ over $[n_1] \times [n_2]$ with $|S| = 100m$. Consider the random variable $N(S^f)$, where $S^f \leftarrow Flatten(S; \alpha, \beta)$. Then there exists a universal constant $C_N$ such that $\mathbb{E}[N(S^f)] \leq C_N \max\left(m^2/(n_1 n_2), m/n_2\right)$, where the randomness is over the internal randomness of $Flatten(\cdot)$ as well as the samples.*

This motivates a two-stage testing strategy: we can first test that the expected value of $N_a(S_p \cup S_q)$ is sufficiently small, and then compute the averaged statistic $Z_a(S_p, S_q)$. To ensure replicability of the first testing stage, we also need to control the variance of the averaged non-singleton sample count $N_a(S_p \cup S_q)$. Fortunately, the variance can be bounded in the same way as the variance of the averaged independence statistic $Z_a(S_p \cup S_q)$. See Appendix B for the detailed argument.

**Lemma 2.11** (Bound Variance of Averaged Non-Singleton Sample Count). *Let $\mathbf{p}$ be a distribution over $[n_1] \times [n_2]$, and $\mathbf{q}$ be the product of marginals of $\mathbf{p}$. Let $S_p, S_q$ be samples from $\mathbf{p}, \mathbf{q}$ respectively. Consider the random variable $N_a(S_p, S_q)$ defined as in Definition 2.8. Then it holds that $\mathrm{Var}[N_a(S_p, S_q)] \leq O(\log^3(n_1 n_2))\mathbb{E}[N_a(S_p, S_q)]$, where the randomness is over $S_p, S_q$.*

---

[3] We think of the two sets as being empty if the algorithm aborts before reaching Line 7

Using Lemma 2.11, we show that we can replicably test whether $\mathbb{E}[N_a(S_p, S_q)]$ is on the order of $\max\left(m^2/(n_1 n_2), m/\min(n_1, n_2)\right)$ by simply drawing random sample sets $S_p, S_q$, (approximately) computing $N_a(S_p, S_q)$, and then comparing it with an appropriately chosen random threshold.

We are now ready to present our full independence tester. The full analysis and proof of Theorem 1.6 can be found in Appendix B.

---

**Algorithm 2** REPINDEPENDENCESTATS

---

    **Input:** sample access to an unknown distribution $\mathbf{p}$ over $[n_1] \times [n_2]$
    **Parameter:** $\epsilon \in (0, 1/4)$ tolerance, $\rho \in (0, 1/4)$ replicability.
    **Output:** Whether $\mathbf{p}$ is a product distribution.

1: Let $m$ be defined as in Line 1 of INDEPENDENCESTATS.
2: $\bar{S}_p \leftarrow 100m$ samples from $\mathbf{p}$, and $\bar{S}_q \leftarrow 100m$ samples from $\mathbf{q}$, the product of marginals of $\mathbf{p}$.
3: Estimate $N_a(\bar{S}_p, \bar{S}_q)$ (see Definition 2.8) up to error $o(1)$ by running INDEPENDENCESTATS$(\bar{S}_p, \bar{S}_q)$ with fresh randomness for sufficiently many times.
4: Draw $r \sim \mathcal{U}([2C_N, 100C_N])$, where $C_N$ is the constant from Lemma 2.10.
5: Reject if (estimated) $N_a(\bar{S}_p, \bar{S}_q) > r \max\left(m^2/(n_1 n_2), m/n_2\right)$.
6: $S_p \leftarrow 100m$ samples from $\mathbf{p}$, and $S_q \leftarrow 100m$ samples from $\mathbf{q}$, the product of marginals of $\mathbf{p}$.
7: Estimate $Z_a(S_p, S_q)$ (see Definition 2.6) up to error $o(1)$ by running INDEPENDENCESTATS$(S)$ with fresh randomness for sufficiently many times.
8: Draw $r \sim \mathcal{U}([C_{I_1}, C_{I_2}])$, where $C_{I_1}, C_{I_2}$ are constants from Lemma 2.5.
9: Reject if (estimated) $Z_a(S_p, S_q) > r \min\left(\varepsilon m, m^2 \varepsilon^2/(n_1 n_2), m^{3/2} \varepsilon^2/\sqrt{n_1 n_2}\right)$. Otherwise, accept.

---

# 3 Lower Bounds for Replicable Uniformity Testing

In this section, we show the sample complexity lower bound $\tilde{\Omega}\left(\epsilon^{-2}\rho^{-2} + \sqrt{n}\epsilon^{-2}\rho^{-1}\right)$ for $(\epsilon, \rho)$-replicable uniformity testing over $[n]$. The $\tilde{\Omega}(\epsilon^{-2}\rho^{-2})$ part follows from the lower bound in Lemma 7.2 of Impagliazzo et al. [2022] for the naive case when $n = 2$, i.e. distinguishing a fair from biased coin so we focus on establishing the lower bound $\tilde{\Omega}(\sqrt{n}\epsilon^{-2}\rho^{-1})$. As such, we assume that $\tilde{o}\left(\sqrt{n}\varepsilon^{-2}\rho^{-1}\right) = \varepsilon^{-2}\rho^{-2}$, which implies the implicit bound $\sqrt{n}\varepsilon^{-2}\rho^{-1} = \tilde{o}\left(n\varepsilon^{-2}\right)$, throughout this section. To begin with, we apply a common technique called Poissonization. Specifically, it reduces the task into showing lower bounds against *Poissonized* testers allowing a more flexible sampling process from non-negative measures in place of the standard testers that are restricted to take a fixed number of samples from distributions. Formally, the Poissonized tester is defined as follows.

**Definition 3.1** (Poissonized Tester and Poisson Sampling). *Given a non-negative measure $\mathbf{p}$ over $[n]$ and an integer $m$, the Poisson sampling model samples a number $m' \sim Poi\left(m\|\mathbf{p}\|_1\right)$, and draws $m'$ samples from $\mathbf{p}/\|\mathbf{p}\|_1$. Let $T \in \mathbb{R}^n$ be the random vector where $T_i$ counts the number of element $i$ seen among the samples. We write $PoiS(m, \mathbf{p})$ to denote the distribution of the random vector $T$. We say $\mathcal{A}$ is a Poissonized tester with sample complexity $m$ if it takes as input $T \sim PoiS(m, \mathbf{p})$.*

Based on this, we can relax the hard instances for uniformity testing to be in general non-negative measures over $[n]$. The definition below describes the meta-distribution $\mathcal{M}_\xi$ over non-negative measures over $[n]$ that forms the family of hard instances for uniformity testing.

**Definition 3.2** (Uniformity Hard Instance). *For $\xi \in [0, \varepsilon]$, we define $\mathcal{M}_\xi$ to be the distribution over non-negative measures $\mathbf{p}_\xi$ defined as follows: $\mathbf{p}_\xi(i) = \frac{1+\xi}{n}$ with probability $\frac{1}{2}$ and $\frac{1-\xi}{n}$ otherwise. The hard instance $\mathcal{H}_U$ for replicable uniformity testing is given by first $\xi \sim \mathcal{U}([0, \varepsilon])$, then $\mathbf{p}_\xi \sim \mathcal{M}_\xi$.*

Using a standard minimax-style argument from Impagliazzo et al. [2022], it suffices to give a lower bound for deterministic algorithms (fixed random seed $r$) on a random instance from $\mathcal{H}_U$. More specifically, the task can be reduced to showing that any deterministic algorithm that satisfies *distributional correctness* w.r.t. $\mathcal{M}_0$ and $\mathcal{M}_\varepsilon$ cannot at the same time satisfy *distributional replicability* with respect to $\mathcal{H}_U$.

**Definition 3.3** (Distributional Correctness/Replicability). *Let $\mathcal{M}_0, \mathcal{M}_\varepsilon, \mathcal{H}$ be meta-distributions over non-negative measures over $[n]$. Let $\mathcal{A}$ be a Poissonized tester with sample complexity $m$. (1)*

*We say $\mathcal{A}$ is $\delta$-correct w.r.t. $\mathcal{M}_0$ and $\mathcal{M}_\varepsilon$ if $\Pr_{r,\mathbf{p}\sim\mathcal{M}_0,T\sim PoiS(m,\mathbf{p})}[\mathcal{A}(T) = Accept] \geq 1 - \delta$ and $\Pr_{r,\mathbf{p}\sim\mathcal{M}_\varepsilon,T\sim PoiS(m,\mathbf{p})}[\mathcal{A}(T) = Accept] \leq \delta$. (2) We say $\mathcal{A}$ is $\rho$-replicable with respect to $\mathcal{H}$ if it holds that $\Pr_{\mathbf{p}\sim\mathcal{H},T,T'\sim PoiS(m,\mathbf{p})}[\mathcal{A}(T) \neq \mathcal{A}(T')] \leq \rho$.*

We defer an elaboration on the reduction from showing lower bounds against deterministic Poissonized testers to those against randomized standard testers to Appendix D. Equipped with this reduction, the proof of Theorem 1.3 can then be reduced to the following main result of this section:

**Proposition 3.4.** *Let $\mathcal{M}_\xi$ be the meta-distribution parametrized by $\xi \in [0, \varepsilon]$ defined as in Definition 3.2, $\mathcal{A}$ be a deterministic Poissonized tester with sample complexity $m = \tilde{o}\left(\sqrt{n}\varepsilon^{-2}\rho^{-1}\right)$. If $\mathcal{A}$ is $0.1$-correct with respect to $\mathcal{M}_0$ and $\mathcal{M}_\varepsilon$, then $\mathcal{A}$ cannot be $\rho \log^{-2} n$-replicable with respect to $\mathcal{H}_U$.*

From now on, we focus on the proof of Proposition 3.4. Since $\mathcal{A}$ is assumed to be a deterministic tester, we note that $\Pr_{T,T'\sim\text{PoiS}(m,\mathbf{p})}[\mathcal{A}(T) \neq \mathcal{A}(T')] \geq 0.1$ holds as long as the acceptance probability of $\mathcal{A}(T)$ lies in the interval $[1/3, 2/3]$. For convenience, we define the function $\text{Acc}_m(\mathbf{p}, \mathcal{A}) := \Pr_{T\sim\text{PoiS}(m,\mathbf{p})}[\mathcal{A}(T) = \text{Accept}]$. It then suffices for us to show

$$\Pr_{\mathbf{p}\sim\mathcal{H}_U}\left[\text{Acc}_m(\mathbf{p}, \mathcal{A}) \in (\log^{-2} n, 1 - \log^{-2} n)\right] \geq \rho. \tag{1}$$

Recall that $\mathcal{H}_U$ is defined to first select $\xi$ randomly from $[0, \varepsilon]$, and then sample from the distribution family $\mathcal{M}_\xi$. Hence, towards showing Equation (1), we will first show the intermediate result that the *average* acceptance probability $\mathbb{E}_{\mathbf{p}\sim\mathcal{M}_\xi}[\text{Acc}_m(\mathbf{p}, \mathcal{A})]$ is close to $1/2$ with probability at least $\rho$ over the random choice of $\xi$. At a high-level, we observe that the expected acceptance probability function must evaluate to exactly $1/2$ for some $\xi \in (0, \varepsilon)$ due to continuity, and then draws tools from information theory to show that the function is in general $0.1(\varepsilon\rho)^{-1}$-Lipschitz with respect to the parameter $\xi$. The argument is similar to the one employed in Liu and Ye [2024], and so we defer it to Appendix E.1. The formal statement is given below.

**Lemma 3.5.** *Let $\mathcal{A}$ be a deterministic Poissonized tester that is $0.1$-correct w.r.t. $\mathcal{M}_0$ and $\mathcal{M}_\varepsilon$ and $m = \tilde{o}(\sqrt{n}\varepsilon^{-2}\rho^{-1})$, then $\Pr_{\xi\sim\mathcal{U}([0,\varepsilon])}\left[\mathbb{E}_{\mathbf{p}\sim\mathcal{M}_\xi}[Acc_m(\mathbf{p}, \mathcal{A})] \in (1/3, 2/3)\right] \geq \rho$.*

To conclude the proof of Equation (1), we then relate the acceptance probability $\text{Acc}_m(\mathbf{p})$ for a random $\mathbf{p} \sim \mathcal{M}_\xi$ to its expected value. To achieve the goal, the authors from Liu and Ye [2024] exploit the assumption that the underlying tester $\mathcal{A}$ is symmetric. Our main technical contribution here is that we managed to remove this assumption. In particular, we show that even if the underlying tester is not symmetric, the acceptance probability $\text{Acc}_m(\mathbf{p})$ will nonetheless satisfy strong concentration properties as long as the tester is still moderately replicable w.r.t. $\mathcal{H}_U$.

**Lemma 3.6** (Concentration of Acceptance Probabilities). *Let $\xi \in (0, \varepsilon)$ and $\mathcal{A}$ be a deterministic tester that is $\log^{-2} n$-replicable with respect to $\mathcal{H}_U$. Assume that $m = \tilde{o}(n\varepsilon^{-2})$. Then it holds $\Pr_{\mathbf{p}\sim\mathcal{M}_\xi}\left(\left|Acc_m(\mathbf{p}, \mathcal{A}) - \mathbb{E}_{\mathbf{p}'\sim\mathcal{M}_\xi}[Acc_m(\mathbf{p}', \mathcal{A})]\right| > \frac{1}{4}\right) \leq \frac{1}{2}$.*

The formal proof is deferred to Appendix E.2. At a high level, we construct a random walk on the sample space whose stationary distribution is the same as $T \sim \mathbf{p}$, where $\mathbf{p} \sim \mathcal{M}_\xi$.

**Definition 3.7** (Sample Random Walk). *Let $\mathcal{M}$ be a meta-distribution over non-negative measures over $[n]$, and $m \in \mathbb{Z}_+$. The sample random walk $\mathbf{RW}_{m,\mathcal{M}}$ is defined on the graph whose vertex set is $\mathbb{N}^n$ (where each vertex corresponds to a sample count vector $T$) and transitions $(T_1, T_2)$ are defined by the conditional distribution of $T_2$ given $T_1$ induced by the joint distribution given by the following process: (1) Choose $\mathbf{p} \sim \mathcal{M}$. (2) $T_1, T_2$ are sampled independently from $PoiS(m, \mathbf{p})$. Moreover, for a sample count vector $T$, we denote by $\mathbf{RW}_{m,\mathcal{M}}^k(T)$ the random variable representing the outcome after $k$ steps of the random walk $\mathbf{RW}_{m,\mathcal{M}}$ from $T$. For a non-negative measure $\mathbf{p}$ over $[n]$, we denote by $\mathbf{RW}_{m,\mathcal{M}}^k(\mathbf{p})$ the distribution of $\mathbf{RW}_{m,\mathcal{M}}^k(T)$, where $T \sim PoiS(m, \mathbf{p})$.*

For simplicity, we write $\mathbf{RW}_{m,\xi} := \mathbf{RW}_{m,\mathcal{M}_\xi}$ where $\mathcal{M}_\xi$ is the meta-distribution given in Definition 3.2. The random walk turns out to mix very rapidly.

**Lemma 3.8.** *Let $\xi \in (0, \varepsilon)$ and $m = \tilde{o}(n\varepsilon^{-2})$. Then $\mathbf{RW}_{m,\xi}$ has mixing time $\tau(\delta) = O(\log(n/\delta))$.*

The proof is deferred to Appendix E.2. To see that the random walk is fast mixing, we observe that $\mathbf{RW}_{m,\xi}$ is a product of $n$ independent random walks induced by the following process on the sample counts $t_1, t_2 \in \mathbb{N}$ for each domain element: 1) choose $\lambda \sim \mathcal{U}(\{\lambda_+ := (1+\xi)/n, \lambda_- := (1-\xi)/n\})$

and 2) $t_1, t_2$ are sampled independently from $\mathrm{PoiS}(m, \lambda)$. We show that the total variation distance between $\mathrm{PoiS}(m, \lambda_+)$ and $\mathrm{PoiS}(m, \lambda_-)$ is not too large, so that most initial states $t_1$ are about as equally likely to be generated by $\mathrm{PoiS}(m, \lambda_+)$ as to be generated by $\mathrm{PoiS}(m, \lambda_-)$. Consequently, the distribution of $\lambda$ conditioned on $t_1$ will be close to the uniform distribution over $\lambda_+, \lambda_-$, further implying that the conditional distribution of the next state $t_2$ will be close to the stationary distribution.

The fast mixing time of the random walk then allows us to approximate the stationary distribution by $\mathbf{RW}_{m,\xi}^k(\mathbf{p})$ for some $k = \mathrm{polylog}(n)$. As a result, we can write $\left| \mathbb{E}_{T \sim \mathrm{PoiS}(m, \mathbf{p})}[\mathcal{A}(T)] - \mathbb{E}_{\mathbf{p}' \sim \mathcal{M}_\xi}\left[\mathbb{E}_{T \sim \mathrm{PoiS}(m, \mathbf{p}')}[\mathcal{A}(T)]\right] \right| \approx \left| \mathbb{E}_{T \sim \mathrm{PoiS}(m, \mathbf{p})}[\mathcal{A}(T)] - \mathbb{E}_{T' \sim \mathbf{RW}_{m,\xi}^k(\mathbf{p})}[\mathcal{A}(T')] \right|$. Since $\mathcal{A}$ is replicable with respect to $\mathcal{H}_U$, we can use the triangle inequality and some simple algebraic manipulation to further bound the above by the sum of the terms

$$\Pr_{T \sim \mathbf{RW}_{m,\xi}^{i-1}(\mathbf{p}),\, T' \sim \mathbf{RW}_{m,\xi}^i(\mathbf{p})}[\mathcal{A}(T) \neq \mathcal{A}(T')] \,, \tag{2}$$

where $i \in [k]$. While it is challenging to establish a uniform bound on the terms for an arbitrary non-negative measure $\mathbf{p}$, it turns out that this is not so hard for an "average" $\mathbf{p} \sim \mathcal{M}_\xi$. At a high level, if we consider the expected value of the disagreement probability in Equation (2) over $\mathbf{p} \sim \mathcal{M}_\xi$, the term simplifies to $\Pr_{T \sim \pi_\xi, T' \sim \mathbf{RW}_{m,\xi}(T)}[\mathcal{A}(T) \neq \mathcal{A}(T')] = \Pr_{\mathbf{p} \sim \mathcal{M}_\xi, T, T' \sim \mathrm{PoiS}(m, \mathbf{p})}[\mathcal{A}(T) \neq \mathcal{A}(T')]$, where the equality follows from the definition of the stationary distribution $\pi_\xi$ of the random walk. Therefore, the expected value of the disagreement probability cannot be too large as long as the tester $\mathcal{A}$ is still moderately replicable with respect to $\mathcal{H}_U$. The formal statement is given below, and the proof is deferred to Appendix E.3.

**Lemma 3.9** (Indistinguishability of Random Walk Step). *Let $\mathcal{A}$ be a deterministic uniformity tester and $\mathbf{p} \sim \mathcal{M}_\xi$. Define $\kappa := \Pr_{\mathbf{p} \sim \mathcal{M}_\xi, T, T' \sim PoiS(m, \mathbf{p})}[\mathcal{A}(T) \neq \mathcal{A}(T')]$. With probability at least $1/2$, it holds that $\sum_{i=1}^k \Pr_{T \sim \mathbf{RW}_{m,\xi}^{i-1}(\mathbf{p}),\, T' \sim \mathbf{RW}_{m,\xi}(T)}[\mathcal{A}(T) \neq \mathcal{A}(T')] < 2k\kappa$, where the randomness is over choice of $\mathbf{p} \sim \mathcal{M}_\xi$.*

The proof of Lemma 3.6 then largely follows from Lemmas 3.8 and 3.9. After that, the proof of Proposition 3.4 follows from Lemmas 3.5 and 3.6. See Appendix E.4 for the formal arguments.

## Acknowledgments and Disclosure of Funding

Ilias Diakonikolas and Jingyi Gao are supported by NSF Medium Award CCF-2107079 and an H.I. Romnes Faculty Fellowship. Daniel Kane and Sihan Liu are supported by NSF Medium Award CCF-2107547 and NSF Award CCF-1553288 (CAREER). Christopher Ye is supported by NSF Medium Award AF-2212136 and HDR TRIPODS Phase II grant 2217058 (EnCORE Institute).

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

# Supplementary Material

## A  Additional Preliminaries

### A.1  Probability and Information Theory

In this subsection, we present some basic lemmas about probability and information theory.

**Lemma A.1** (Poisson Concentration (see e.g. Canonne [2019])). *Let $X \sim \mathrm{Poi}\,(\lambda)$. Then for any $x > 0$,*

$$\max(\Pr(X \geq \lambda + x), \Pr(X \leq \lambda - x)) < e^{-\frac{x^2}{2(\lambda + x)}}.$$

**Claim A.2** (Asymptotic Upper Bound of Mutual Information). *Let $X$ be an unbiased uniform random bit, and $M$ be a discrete random variable s.t. $\Pr[M = a | X = 0] = \Theta(1)\Pr[M = a | X = 1]$, for all $a$ within the support of $M$, then the mutual information between the two random variables satisfies that $I(X : M) = O(1) \sum_a \frac{(\Pr[M=a|X=0] - \Pr[M=a|X=1])^2}{\Pr[M=a|X=0] + \Pr[M=a|X=1]}$.*

*Proof.* Denote $\alpha := \Pr[M = a, X = 1], \beta := \Pr[M = a, X = 0]$ for simplicity. Since $\Pr[X = 1] = \Pr[X = 0] = 1/2 \implies \beta = \Theta(\alpha)$ then $\frac{(\beta - \alpha)^2}{\beta + \alpha} = \Theta\left(\frac{(\beta - \alpha)^2}{\alpha}\right) = \Theta\left(\frac{(\beta - \alpha)^2}{\beta}\right)$. By definition,

$$
\begin{aligned}
I(X : M) &= \sum_a \sum_{i=0,1} \Pr[X = i, M = a] \log\left(\frac{\Pr[X = i, M = a]}{\Pr[X = i]\Pr[M = a]}\right) \\
&= \frac{1}{2} \sum_a \left(\beta \log\left(\frac{2\beta}{\beta + \alpha}\right) + \alpha \log\left(\frac{2\alpha}{\beta + \alpha}\right)\right)
\end{aligned}
$$

rearranging

$$
= \Theta(1) \sum_a \left(\beta \log\left(\frac{1}{1 + \frac{\alpha - \beta}{2\beta}}\right) + \alpha \log\left(\frac{1}{1 - \frac{\alpha - \beta}{2\alpha}}\right)\right).
$$

Denote $A := \frac{\alpha - \beta}{2\beta}$ and $B := \frac{\alpha - \beta}{2\alpha}$, then by Taylor expansion of $\log\left(\frac{1}{1+A}\right)$ and $\log\left(\frac{1}{1-B}\right)$ we have that

$$
\begin{aligned}
I(X : M) &= \Theta(1) \sum_a \left(\beta \left(\sum_{n=1}^{\infty} (-1)^n \frac{A^n}{n}\right) + \alpha \left(\sum_{n=1}^{\infty} \frac{B^n}{n}\right)\right) \\
&= \Theta(1) \sum_a \left(\sum_{\substack{n=3 \\ n \text{ odd}}} \frac{1}{n}(\alpha B^n - \beta A^n) + \sum_{\substack{n=2 \\ n \text{ even}}} \frac{1}{n}(\beta A^n + \alpha B^n)\right) \\
&\leq O(1) \sum_a 2 \sum_{n=2}(\alpha B^n + \beta A^n) \leq O(1)\alpha B^2 = \sum_a O\left(\frac{(\beta - \alpha)^2}{\alpha}\right)
\end{aligned}
$$

as desired. $\qquad\square$

### A.2  Random Walks

Let **RW** denote a random walk with transition matrix $P$ where $P(x, y)$ denotes the probability of transitioning to state $y$ from state $x$. $P^t(x, y)$ hence denotes the probability of landing in a state $y$ after $t$ steps if one starts from state $x$. We give some elementary properties of random walks important to our analysis.

**Definition A.3.** *A random walk **RW** is irreducible if for all $x, y$, there exists $t > 0$ such that $P^t(x, y) > 0$.*

**Definition A.4.** *For a given state $x$, let $\mathcal{T}(x) := \{t > 0 \text{ s.t. } P^t(x, x) > 0\}$. The period of $x$ is the greatest common divisor of $\mathcal{T}(x)$. A random walk is aperiodic if the period of any of its states is $1$.*

A random walk that is both irreducible and aperiodic is called ergodic.

We say a distribution $\pi$ over the states of a random walk is stationary if it stays invariant after one step of the random walk. A useful fact is that any ergodic random walk has a unique stationary distribution.

**Fact 2.** *Any irreducible and aperiodic random walk has a unique stationary distribution.*

While any irreducible and aperiodic random walk is guaranteed to converge to its stationary distribution, we are interested in a quantitative bound on the convergence rate. In what follows, we define the concept of the mixing time of a random walk and give the relevant preliminaries (see e.g. Levin et al. [2006], Guruswami [2016] for a detailed survey).

**Definition A.5.** *The mixing time $\tau(\delta)$ of an ergodic random walk $\mathbf{RW}$ with stationary distribution $\pi$ is defined by*

$$\tau(\delta) = \max_{i \in \Omega} \min \left\{ t \text{ s.t. } (\forall t' \geq t) \sum_{j \in \Omega} \left| P_{ij}^t - \pi_j \right| < \delta \right\}.$$

We require the following facts regarding mixing time. First, the mixing time of a product random walk can be bounded via the mixing times of the individual coordinates (up to polynomial factors in the dimension).

**Lemma A.6.** *Let $\mathbf{RW}$ be a random walk over the product space $\mathcal{X}^n$, where the $i$-th coordinate follows an independent random walk $\mathbf{RW}_i$ over $\mathcal{X}$. Assume that $\mathbf{RW}_i$ has mixing time $\tau_i(\delta)$. Then the mixing time of $\mathbf{RW}$ satisfies that $\tau(\delta) \leq \max_i \tau_i(\delta/n)$.*

*Proof.* Let $P^{(i)}$ denote the transition matrix for each coordinate $i$ and $\pi^{(i)}$ denote the stationary distribution. Then,

$$\left\| \prod_{\ell=1}^n P^{(\ell)^t} - \prod_{\ell=1}^n \pi^{(\ell)} \right\|_1 \leq \sum_{\ell=1}^n \left\| P^{(\ell)^t} - \pi^{(\ell)} \right\|_1.$$

This concludes the proof of Lemma A.6. $\qquad\square$

For any ergodic random walk $\mathbf{RW}$ with transition matrix $P$ and stationary distribution $\pi$, we let $\lambda$ denote the eigenvalues of $P$. The following lemma relates the mixing time to the absolute spectral gap $\lambda_*$ of the transition matrix (or alternative the relaxation time of the random walk).

**Definition A.7.** *Let $\mathcal{E}$ be the set of eigenvalues of the transition matrix $P$. The absolute spectral gap of a Markov chain with transition matrix $P$ is $\gamma_* = 1 - \lambda_*$, where $\lambda_* = \max |\lambda|$ s.t. $\lambda \neq 1$ and $\lambda \in \mathcal{E}$.*

**Definition A.8.** *The relaxation time of a Markov chain is $t_{\mathrm{rel}} = \frac{1}{\gamma_*}$.*

Another property important to the mixing time of the random walk is the detailed balance criteria. If a random walk satisfies this criteria, then we say it is reversible.

**Definition A.9** (Detailed Balance Criteria)**.** *A random walk is reversible if and only if for all states $x, y$, $\pi(x)P(x,y) = \pi(y)P(y,x)$.*

We are now ready to state the mixing time of a Markov chain in terms of its relaxation time (or the inverse of the absolute spectral gap).

**Theorem A.10** (Theorem 12.5 of Levin et al. [2006])**.** *For an ergodic and reversible Markov chain, its mixing time satisfies that*

$$\tau(\delta) \geq (t_{\mathrm{rel}} - 1) \log(1/2\delta).$$

# B   Replicable Independence Testing Algorithm

In this section, we provide omitted proofs and analysis for our replicable independence tester, and then conclude the proof of Theorem 1.6.

We begin with a useful property of the flattening procedure (see Definition 2.1) — it ensures that there will be no "heavy" bins after the operation with high probability .

**Lemma B.1.** *Let $S$ be a set of samples over $[n]$ with $|S| = \text{poly}(n)$, and $S^f = \text{Flatten}(S; \alpha)$. Denote by $T^f$ the sample count vector of the flattened samples. For any constant $C$, it holds that $T_i^f \leq O\left(\alpha^{-1} \log n\right)$ for all $i \in \text{supp}\left(S^f\right)$ with probability at least $1 - n^{-C}$, where the randomness is over the internal randomness of $\text{Flatten}(\cdot)$.*

*Proof.* After sorting the samples in $S$, we note that the position of the first flattening sample follows exactly a geometric distribution with mean $1/\alpha$. Denote by $Y$ its position. We have that $\Pr[Y \geq t] = (1-\alpha)^{t-1} = \exp\left((t-1)\log(1-\alpha)\right) \leq \exp\left(-\alpha(t-1)\right)$. In particular, this implies that $\Pr[Y \geq a\alpha^{-1}\log(n)] \leq n^{-a}$ for any number $a > 0$. This shows that with probability at least $n^{-a}$ it holds that the number of samples falling in the first bin is at most $a\alpha^{-1}\log(1/n)$. If we choose $a$ to be a sufficiently large constant, we then have that $Y \leq O\left(\alpha^{-1}\log(n)\right)$ with probability at least $1 - 1/\text{poly}(n)$. Since $Y$ is also the number of samples within the first sub-bin, Lemma B.1 then follows applying this argument to all subsequent samples and the union bound. $\qquad\square$

Recall that a key technical step is to bound the variance of the averaged independence statistic $Z_a\left(S_p, S_q\right)$ by the expected value of the non-singleton sample count $N_a(S_p, S_q)$.

*Proof of Lemma 2.9.* We will bound the variance of $Z_a(S_p, S_q)$ by the expected sum over samples of the squared difference in the final test statistic by removing each sample.

In particular, suppose that $S_p, S_q$ contains the samples $\{P_\ell\}_{\ell=1}^k$. For convenience, we denote by $S_{p,-\ell}, S_{q,-\ell}$ the corresponding set after removing the sample $P_\ell$.[4] We then have the following inequality that bounds from above the variance.

$$\text{Var}[Z_a(S_p, S_q)] \leq O(1)\, \mathbb{E}\left[\sum_{\ell=1}^k \left(Z_a\left(S_{p,-\ell}, S_{q,-\ell}\right) - Z_a\left(S_p\, S_q\right)\right)^2\right], \tag{3}$$

where the randomness is over the samples. Fix some sample sets $S_p, S_q$. Consider the random variables $Z = \textsc{IndependenceStats}\left(S_p, S_q\right)$ and $Z_{-\ell} = \textsc{IndependenceStats}\left(S_{p,-\ell}, S_{q,-\ell}\right)$. We claim that it suffices for us to show that

$$\sum_{\ell=1}^k \left(\mathbb{E}[Z] - \mathbb{E}[Z_{-\ell}]\right)^2 \leq O\left(\log^2(n_1 n_2)\right) N_a(S_p, S_q), \tag{4}$$

where the expectation is over the internal randomness of $\textsc{IndependenceStats}$. After that, taking expectation over the randomness of the samples on both sides of Equation (4) and combining it with Equation (3) then concludes the proof.

It then remains for us to show Equation (4). Recall that the tester first partitions the samples into flattening samples and testing samples randomly. We denote by $S_p^f, S_q^f$ the flattened testing samples constructed in Line 3 from the original sample set $S_p, S_q$, and $S_{p,-\ell}^f, S_{q,-\ell}^f$ the ones constructed from the leave-one-out sample sets $S_{p,-\ell}, S_{q,-\ell}$.

Denote by $F_\ell^x, F_\ell^y \in \{0,1\}$ the indicator variables of whether $P_\ell$ is selected for row or column flattening purpose respectively (see Definition 2.2) while constructing $S_p^f, S_q^f$. We then break into cases based on the values of $F_\ell^x, F_\ell^y$.

In the first case, we have that $F_\ell^x = F_\ell^y = 0$. This suggests that the $\ell$-th sample is not selected as a flattening sample. Hence, there exists a flattened version of $P_\ell$, which we denote by $P_\ell^f$, within $S_p^f \cup S_q^f$. In this case, $S_{p,-\ell}^f, S_{q,-\ell}^f$ are obtained by deleting exactly $P_\ell^f$ from $S_p^f, S_q^f$. There are then again two sub-cases. Either $P_\ell^f$ is a singleton sample among $S_{p,-\ell}^f \cup S_{q,-\ell}^f$. In that case, we must have $Z_C(S_p^f, S_q^f) = Z_C(\bar{S}_{p,-\ell}^f, \bar{S}_{q,-\ell}^f)$ by Fact 1. Otherwise, we have $\left|Z(S_p^f, S_q^f) - Z(\bar{S}_{p,k}^f, \bar{S}_{q,k}^f)\right| \leq O(1)$ as the closeness test statistic is Lipchitz in its inputs. As a result, it follows that

$$\sum_{\ell=1}^k \left(\mathbb{E}[Z\, \mathbb{1}\{F_\ell^x = F_\ell^y = 0\}] - \mathbb{E}[Z_{-\ell}]\right)^2 \leq \mathbb{E}[N(S_p^f \cup S_q^f)]. \tag{5}$$

---

[4]If $P_\ell \notin S_p$, then $S_{p,-\ell} = S_p$, and the same for $S_{q,-\ell}$.

Next, consider the case that $F_\ell^x = 1$ and $F_\ell^y = 0$, which happens with probability at most $p_x :=$ $\min\left(n_1/(100m), 1/100\right)$. This suggests that the $P_\ell$ is selected as a row flattening sample. Denote by $K_\ell$ the number of samples lying in the same row as $P_\ell$. Consider the following coupling between $(S_p^f, S_q^f)$ (conditioned on $F_\ell^x = 1$ and $F_\ell^y = 0$) and $(S_{p,-\ell}^f, S_{q,-\ell}^f)$: (1) Pick a random sub-row $a$ (in the flattened domain) weighted by the total sample count of sub-row $a$ within $S_{p,-\ell}^f \cup S_{q,-\ell}^f$ divided by $K_\ell$, (2) subdivide the sub-row into two sub-rows $a_1, a_2$, and (3) randomly assign the samples from $a$ to $a_1, a_2$. Denote by $T_i$ the total number of samples among $S_{p,-\ell}^f \cup S_{q,-\ell}^f$ within the sub-row $i$, and $N_i$ the corresponding non-singleton sample count. Note that if a flattened sample is a singleton sample among $S_{p,-\ell}^f \cup S_{q,-\ell}^f$, then it remains a singleton sample after the subdivision, and hence has no impact on the final closeness statistic. Therefore, such subdivision can change the final closeness statistic by at most $N_i$. On the other hand, the probability of the sub-row $i$ being selected and the sample $\ell$ being selected for flattening purpose is at most $p_x T_i/K_\ell$. Hence, the averaged statistic changes by at most

$$\left(\mathbb{E}[Z\,\mathbb{1}\{F_\ell^x = 1, F_\ell^y = 0\}] - \mathbb{E}[Z_{-\ell}]\right)^2 \le O(1) \left(\sum_{i:\text{sub-rows of the row of}P_\ell} \mathbb{E}\left[\frac{N_i p_x T_i}{K_\ell}\right]\right)^2.$$

Summing over all $\ell'$ such that $P_\ell$ and $P_\ell'$ lie in the same row then gives that

$$\sum_{\ell':P_\ell \text{ lies in the same row as } P_{\ell'}} \left(\mathbb{E}[Z\,\mathbb{1}\{F_\ell^x = 1, F_\ell^y = 0\}] - \mathbb{E}[Z_{-\ell}]\right)^2 \le O(1)\,\mathbb{E}^2\left[\sum_{i:\text{sub-rows of the row of}P_\ell} N_i p_x T_i\right]/K_\ell.$$

By Lemma B.1, we have that $T_i$ is at most $\log(n_1)p_x^{-1}$ with probability at least $1 - 1/\text{poly}\,(n_1)$. Besides, $\sum_{i:\text{sub-rows of the row of}P_\ell} N_i$ is always at most $K_\ell$. It then follows that

$$\sum_{\ell':P_\ell \text{ lies in the same row as } P_{\ell'}} \left(\mathbb{E}[Z\,\mathbb{1}\{F_\ell^x = 1, F_\ell^y = 0\}] - \mathbb{E}[Z_{-\ell}]\right)^2$$

$$\le O(\log^2 n)\,\mathbb{E}^2\left[\sum_{i:\text{sub-rows of the row of}P_\ell} N_i\right]/K_\ell$$

$$\le O(\log^2 n)\,\mathbb{E}\left[\sum_{i:\text{sub-rows of the row of}P_\ell} N_i\right].$$

Note that the non-single sample count can only increase conditioned on that $P_\ell$ is not selected for flattening, which happens with at least constant probability. As a result, the expected number of non-singleton samples among $S_{p,-\ell}^f \cup S_{q,-\ell}^f$ is always at most a constant factor of the expected number of non-singleton samples among $S_p^f \cup S_q^f$. Summing over all $\ell$ then gives that

$$\sum_{\ell=1}^k \left(\mathbb{E}[Z\,\mathbb{1}\{F_\ell^x = 1, F_\ell^y = 0\}] - \mathbb{E}[Z_{-\ell}]\right)^2 \le O(\log^2 n_1)\,\mathbb{E}\left[N(S_p^f, S_q^f)\right]. \tag{6}$$

This then concludes the analysis of the second case.

In the third case, we assume that $F_\ell^x = 0, F_\ell^y = 1$. Using an argument that is almost identical to the second case, one can show that

$$\sum_{\ell=1}^k \left(\mathbb{E}[Z\,\mathbb{1}\{F_\ell^x = 0, F_\ell^y = 1\}] - \mathbb{E}[Z_{-\ell}]\right)^2 \le O(\log^2 n_2)\,\mathbb{E}\left[N(S_p^f, S_q^f)\right]. \tag{7}$$

as this corresponds to the case when the $k$-th sample is chosen for column flattening.

Finally, for $F_k^x = F_k^y = 1$, we can use an argument that is almost identical to the second case to show that

$$\sum_{\ell=1}^k \left(\mathbb{E}[Z\,\mathbb{1}\{F_\ell^x = 1, F_\ell^y = 1\}] - \mathbb{E}[Z\,\mathbb{1}\{F_\ell^x = 0, F_\ell^y = 1\}]\right)^2 \le O(\log^2 n_1)\,\mathbb{E}\left[N(S_p^f, S_q^f)\right].$$

It then follows from the triangle inequality that

$$\sum_{\ell=1}^{k} \left( \mathbb{E}[Z \, \mathbb{1}\{F_\ell^x = 0, F_\ell^y = 1\}] - \mathbb{E}[Z_{-\ell}] \right)^2$$

$$\leq O(1) \left( \sum_{\ell=1}^{k} \left( \mathbb{E}[Z \, \mathbb{1}\{F_\ell^x = 1, F_\ell^y = 1\}] - \mathbb{E}[Z \, \mathbb{1}\{F_\ell^x = 0, F_\ell^y = 1\}] \right)^2 \right.$$

$$\left. + \sum_{\ell=1}^{k} \left( \mathbb{E}[Z \, \mathbb{1}\{F_\ell^x = 0, F_\ell^y = 1\}] - \mathbb{E}[Z_{-\ell}] \right)^2 \right) \leq O(\log^2 n_1) \mathbb{E}\left[ N(S_p^f, S_q^f) \right]. \tag{8}$$

Combining the case analysis (Equations (5) to (8)) then yields that

$$\sum_{\ell=1}^{k} \left( \mathbb{E}[Z] - \mathbb{E}[Z_{-\ell}] \right)^2 \leq O(\log^2 n_1) \mathbb{E}\left[ N(S_p^f \cup S_q^f) \right] = O(\log^2 n_1) N_a(S_p, S_q).$$

This concludes the proof of Equation (4) as well as Lemma 2.9. $\qquad \square$

Recall that we adhere to a two-stage testing strategy, where we first test the size of the expected non-singleton sample count before computing the averaged independence statistics. In what follows, we provide the proof which bounds the variance of the averaged non-singleton sample count by its expected value.

*Proof of Lemma 2.11.* Denote by $N(S_p^f, S_q^f)$ the number of non-singleton samples among $S_p^f \cup S_q^f$. Recall that the averaged non-singleton sample count $N_a(S_p, S_q)$ is simply $\mathbb{E}\left[ N(S_p^f, S_q^f) \right]$, where the randomness is over the flattened sample set $S_p^f, S_q^f$. Similar to the closeness statistic $Z_C(S_p^f, S_q^f)$, $N(S_p^f, S_q^f)$ has the two following properties: (1) $N(S_p^f, S_q^f)$ is invariant if one removes any singleton sample from $S_p^f$ or $S_q^f$ and changes by 1 if one removes a non-singleton sample. We note that these are the only two properties used in Lemma 2.9 to show that the variance of the averaged statistic $Z_a(S_p, S_q)$ can be bounded from above by $O(\log^2(n_1 n_2)) \mathbb{E}\left[ N_a(S_p, S_q) \right]$. Hence, we can use the same argument to show that $\mathrm{Var}\left[ N_a(S_p, S_q) \right] \leq O(\log^2(n_1 n_2)) \mathbb{E}\left[ N_a(S_p, S_q) \right]$, and this concludes the proof of Lemma 2.11. $\qquad \square$

We are now ready to show the full analysis of our replicable independence tester, and the proof of Theorem 1.6.

*Proof of Theorem 1.6.* Recall that the algorithm has two steps. In the first step, it verifies that the size of the expected value of the non-singleton sample count is not large by comparing $N_a(S_p, S_q)$ with a random threshold. In the second step, it computes the averaged independence statistics $Z_a(S_p, S_q)$ with fresh samples, and compare it with another appropriately chosen random threshold.

We first analyze the correctness and replicability of the first step. Let $m$ be defined as in Line 1 of INDEPENDENCESTATS, and $S_p, S_q$ be sample sets with size $100m$. By Lemma 2.10, the expected number of non-singleton sample count $\mathbb{E}\left[ N_a(S_p, S_q) \right]$ is at most $C_N \max\left( m^2/(n_1 n_2), m/n_2 \right)$ for some constant $C_N$ if the underlying distribution $\mathbf{p}$ is indeed a product distribution. By Lemma 2.11, we have that $\mathrm{Var}\left[ N_a(S_p, S_q) \right] \leq \mathbb{E}\left[ N_a(S_p, S_q) \right]$. We first show the validity of the following bound:

$$\log^2(n_1) \max\left( m^2/(n_1 n_2), m/n_2 \right) \ll \rho^2 \left( \max\left( m^2/(n_1 n_2), m/n_2 \right) \right)^2. \tag{9}$$

In particular, we will see that for this step, it is sufficient if $m \gg n_1^{2/3} n_2^{1/3} \rho^{-2/3} + \sqrt{n_1 n_2} \rho^{-1}$. We begin with a case analysis. In the first case, we have that $m^2/(n_1 n_2)$ is the dominating term in Equation (9). It is not hard to verify that

$$m^2/(n_1 n_2) \leq \rho^2 m^4/(n_1 n_2)^2$$

as long as $m \gg \sqrt{n_1 n_2} \rho^{-1}$. So Equation (9) easily holds in this case. In the second case, we have that $m^2/(n_1 n_2) \leq m/n_2$ and so $m/n_2$ is the dominating term. In this case, we need to show that $m/n_2 \leq \rho^2 (m/n_2)^2$, which is true as long as $m \geq n_2 \rho^{-2}$. In particular, the case assumption

indicates that $n_1^{2/3} n_2^{1/3} \rho^{-2/3} \ll m < n_1$. This implies that $n_1 \gg n_2 \rho^{-2}$, which further implies that $m \gg \sqrt{n_1 n_2} \rho^{-1} > n_2 \rho^{-2}$. This hence concludes the proof of Equation (9).

To argue the correctness of the tester, we analyze the completeness and the soundness cases separately. Denote by $G := C_N \max\left(m^2/(n_1 n_2), m/n_2\right)$. In the completeness case, the expectation is at most $G$, and the variance is at most $O(1) \, \log^2(n_1) G$. By Chebyshev's inequality and Equation (9), the statistic $N_a(S_p, S_q)$ will be at most $2G$ with high constant probability. In the soundness case, suppose $\mathbb{E}\left[N_a(S_p, S_q)\right] \geq 101 G$. It is not hard to verify that $G \gg \log^2(n_1)$ as our choice of $m$ ensures that $m \gg \log^2(n_1)\sqrt{n_1 n_2} \geq \log^2(n_1) n_2$. In particular, this implies that $\sqrt{\log(n_1) G} \ll G$. In this case, by Chebyshev's inequality, $N_a(S_p, S_q)$ is at least $101G - \sqrt{\log^2(n_1) G} \geq 100 G$. The above ensures that the tester will be correct with high constant probability. Combining this with the standard median trick then ensures correctness with probability at least $1 - \rho$ at the cost of increasing the sample complexity by an extra $\log(1/\rho)$ factor.

To argue the replicability of the tester when we are in neither the completeness nor the soundness case, we note that the variance is at most $\log(n_1) G$. By Chebyshev's inequality and Equation (9), we have that the test statistic will concentration around an interval of size $\sqrt{\log(n_1) G} \ll \rho G$ with high constant probability. Again, combining this with the median trick ensures that $N_a(S_p, S_q)$ will lie in an interval (around its expected value) of size $\rho G$ with probability at least $1 - \rho$ (at the cost of increasing the sample complexity by an extra factor of $\log(1/\rho)$). Conditioned on that, we have that the tester will be replicable as long as the random threshold lies outside this interval of size $\rho G$, which happens with probability at least $1 - \rho$. We can therefore conclude that the tester is replicable with probability at least $1 - 2\rho$ by the union bound.

Conditioned on that the first-stage testing passes, we hence must have that

$$\mathrm{Var}[Z_a(S_p, S_q))] \leq \log^2(n_1) \, \mathbb{E}[N_a(S_p, S_q)] \leq O\left(\log^2(n_1)\right) \left(m^2/(n_1 n_2) + m/n_2\right).$$

Besides, since $Z_a(S_p, S_q)$ is the average over some statistic that is Lipchitz in the input samples, we also have the trivial variance bound $\mathrm{Var}[Z_a(S_p, S_q)] \leq O(m)$. Again, we begin with a quantitative bound that will be useful for both the replicability and correctness analysis:

$$\sqrt{\log^2(n_1) \min(m, m^2/(n_1 n_2) + m/n_2)} \ll \rho \min\left(\varepsilon m, m^2 \varepsilon^2/(n_1 n_2), m^{3/2} \varepsilon^2/\sqrt{n_1 n_2}\right). \quad (10)$$

Again, we proceed by a case analysis. Suppose the right hand side evaluates to $\varepsilon m$. We note that $\log(n_1)\sqrt{m} \ll \rho \varepsilon m$ as long as $m \gg \log^2(n_1) \rho^{-2} \varepsilon^{-2}$. So Equation (10) clearly holds in this case. Suppose that the right hand side evaluates to $m^2 \varepsilon^2/(n_1 n_2)$. The case assumption implies that $m^{1/2} \leq \sqrt{n_1 n_2}$, which further implies that $m \leq n_1 n_2$. Since we always have $m \gg n_2$, this suggests that $m/n_2$ will be the dominating term on the left hand side. However, we always have $\log(n_1)\sqrt{m/n_2} \ll \rho m^2 \varepsilon^2/(n_1 n_2)$ as long as $m \gg \log^{2/3}(n_1) n_1^{2/3} n_2^{1/3} \rho^{-2/3} \varepsilon^{-4/3}$. This verifies the validity of Equation (10) in this case. The last case is when the right hand side evaluates to $m^{3/2} \varepsilon^2/\sqrt{n_1} n_2$. In this case, it suffices to show that $\log(n_1)\sqrt{m} \ll \rho m^{3/2} \varepsilon^2/\sqrt{n_1 n_2}$, which is true as long as $m \gg \log(n_1)\sqrt{n_1 n_2}\varepsilon^{-2}\rho^{-1}$. This concludes the proof of Equation (10).

To argue the correctness of the second stage, we again break into the completeness and the soundness cases. For convenience, denote by $H_E := \min\left(\varepsilon m, m^2 \varepsilon^2/(n_1 n_2), m^{3/2} \varepsilon^2/\sqrt{n_1 n_2}\right)$ and $H_V := \log^2(n_1) \min(m, m^2/(n_1 n_2) + m/n_2)$. In the completeness case, by Lemma 2.5, we have that $\mathbb{E}[Z_a(S_p, S_q)] \leq C_{I_1} H_E$. By Equation (10) and Chebyshev's inequality, we have that $Z_a(S_p, S_q) \leq C_{I_1} H_E + O\left(\sqrt{H_V}\right) \leq (C_{I_1} + o(1)) \, H_E$ with high constant probability. In the soundness case, by Lemma 2.5, we have that $\mathbb{E}[Z_a(S_p, S_q)] \geq C_{I_2} H_E$. By Equation (10) and Chebyshev's inequality, we have that $Z_a(S_p, S_q) \geq C_{I_2} H_E - O\left(\sqrt{H_V}\right) \leq (C_{I_2} - o(1)) \, H_E$ with high constant probability. This shows that the test on $Z_a(S_p, S_q)$ is correct with high constant probability. Combining this with the median trick ensures correctness with probability at least $1 - \rho$ at the cost of increasing the sample complexity by an extra factor of $\log(1/\rho)$.

To argue the replicability of the second stage, we note that $Z_a(S_p, S_q)$ must lie in an interval around its expected value with size at most $\sqrt{H_V} \ll \rho H_E$ with high constant probability. Again, combining this with the median trick ensures that $Z_a(S_p, S_q)$ must lie in an interval $L$ of size $\rho H_E$ with probability at least $1 - \rho$ (at the cost of increasing the sample complexity by an extra factor of

$\log(1/\rho)$). Thus, the tester will be replicable as long as the random threshold chosen uniformly random from $[C_{I_1}H_E, C_{I_2}H_E]$ falls outside of this interval $L$. This happens with probability at least $1 - \frac{\rho H_E}{(C_{I_2}-C_{I_1})H_E} \geq 1 - O(\rho)$. We can then conclude that the tester is replicable with probability at least $1 - O(\rho)$ by the union bound.

Lastly, it is clear from the description of REPINDEPENDENCESTATS that the tester draws $\Theta(m)$ many samples, and $m$ (Line 1 of INDEPENDENCESTATS) is within the sample budget of Theorem 1.6. This concludes the proof of Theorem 1.6. $\qquad\square$

## C   Replicable Closeness Testing Algorithm

In this section, we present a replicable closeness tester with optimal sample complexity.

---

**Algorithm 3** REPCLOSENESSTESTER$((\mathbf{p}, \mathbf{q}), \epsilon, \rho, n)$

---

    **Input:** sample access to distribution $\mathbf{p}$ and $\mathbf{q}$ supported on $[n]$.
    **Parameter:** $\epsilon \in (0, 1/4)$ tolerance, $\rho \in (0, 1/4)$ replicability, $n$ support size.
    **Output:** ACCEPT if $\mathbf{p} = \mathbf{q}$, REJECT if $d_{\mathrm{TV}}(\mathbf{p}, \mathbf{q}) \geq \epsilon$.
1:  $m \leftarrow \tilde{\Theta}\left(\frac{n^{2/3}}{\rho^{2/3}\epsilon^{4/3}} + \frac{\sqrt{n}}{\epsilon^2\rho^1} + \frac{1}{\rho^2\epsilon^2}\right)$,
    $(m_{\mathbf{p}}, m'_{\mathbf{p}}, m_{\mathbf{q}}, m'_{\mathbf{q}}) \leftarrow \mathsf{Multinom}(4m, (1/4, 1/4, 1/4, 1/4))$.
2:  Draw two multisets $D_1, D_2$ of iid samples from $p$ of sizes $m_p, m'_p$ respectively; and two multisets $D_3, D_4$ of iid samples from $p$ of sizes $m_q, m'_q$ respectively. $\forall i \in [n]$ let $X_i, X'_i, Y_i, Y'_i$ be the occurrence of $i$ in $D_1, D_2, D_3, D_4$, respectively.
3:  Compute the statistic $\forall i \in [n], Z_i \leftarrow |X_i - Y_i| + |X'_i - Y'_i| - |X_i - X'_i| - |Y_i - Y'_i|$ and $Z \leftarrow \sum_{i=1}^n Z_i$.
4:  Set threshold $r \leftarrow C_1\sqrt{m} + r_0\left(R - C_1\sqrt{m}\right)$ where $r_0 \leftarrow \mathsf{Unif}\left(\frac{1}{4}, \frac{3}{4}\right)$ and $R, C_1$ are given in Lemma C.1.
5:  **return** ACCEPT if $Z \leq r$. REJECT otherwise.

---

To show Theorem 1.5, the key idea is that firstly, to guarantee correctness the threshold we randomly picked needs to fall between an upper bound on the test statistic of the completeness case and a lower bound on the test statistic of the soundness case whp and the proof follows from Diakonikolas et al. [2021]; secondly, to guarantee replicability we need to further make sure that the randomly picked threshold falls in the high confidence interval of the statistic with probability $< \rho$, so that upon multiple runs, the algorithm gives same answers whp. Remark that the main difference between replicable closeness tester and high confidence closeness tester is that the former needs to whp output the same result upon receiving different sample set even in the case when $0 < d_{\mathrm{TV}}(\mathbf{p}, \mathbf{q}) < \epsilon$, yet there's no requirement on the behavior of the latter in such case.

The main ingredients are the two following facts: a concentration bound on statistic $Z$ and the expectation gap between the case when $\mathbf{p} = \mathbf{q}$ and the case when $d_{\mathrm{TV}}(\mathbf{p}, \mathbf{q}) \geq \epsilon$. Luckily, both facts were shown in Diakonikolas et al. [2021].

**Lemma C.1.** *(Expectation Gap, Lemma 3.3 in Diakonikolas et al. [2021]) Given $m, \epsilon, \rho, n, Z$ as specified in Algorithm 3, there exists universal constants $C_1, C_2 > 0$ s.t.*

1. *If $\mathbf{p} = \mathbf{q}$, $\mathbb{E}[Z] \leq C_1\sqrt{m}$;*
2. *If $d_{TV}(\mathbf{p}, \mathbf{q}) \geq \epsilon$, $\mathbb{E}[Z] \geq R := C_2 \min\left(\epsilon m, \frac{m^2\epsilon^2}{n}, \frac{m^{3/2}\epsilon^2}{n^{1/2}}\right)$. In particular, $R \geq C_2\sqrt{m\log(1/\rho)}$.*

**Lemma C.2.** *(Concentration bound on $Z$, Section 3.2 in Diakonikolas et al. [2021]) Given $m, \rho, Z$ as specified in Algorithm 3, there exists a universal constant $C > 0$ such that $\Pr\left[|Z - \mathbb{E}[Z]| \geq C\sqrt{m\log(1/\rho)}\right] < \frac{\rho}{2}$ where $C\sqrt{m\log(1/\rho)} < \frac{1}{4}\left(C_2\sqrt{m\log(1/\rho)} - C_1\sqrt{m}\right)$.*

We are now ready to show the proof of Theorem 1.5.

*Proof of Theorem 1.5.* We first argue the correctness. From Lemma C.1 and Lemma C.2, with probability $\geq 1 - \rho/2$ threshold $r$ falls between the value of $Z$ for the completeness case and the soundness case, whence successfully separates two cases,

We next show replicability. We break into 3 cases based on the value of $R$. Essentially we need to show that for each case the ratio $\frac{\text{concentration bound}}{\text{expectation gap}} \leq \frac{\rho}{6}$.

- When $R = C_2 \epsilon m$, since $m \geq \frac{36(2C+C_1/6)^2}{C_2^2} \cdot \frac{\log(1/\rho)}{\epsilon^2 \rho^2}$, we have that $\frac{2C\sqrt{m\log(1/\rho)}}{C_2 \epsilon m - C_1\sqrt{m}} \leq \frac{\rho}{6}$.
- When $R = C_2 \frac{m^2 \epsilon^2}{n}$, since $m \geq \frac{4(2C+C_1/6)^{2/3}}{C_2^{2/3}} \cdot \frac{\log^{1/3}(1/\rho)n^{2/3}}{\epsilon^{4/3}\rho^{2/3}}$, we have that $\frac{2C\sqrt{m\log(1/\rho)}}{C_2 \frac{m^2 \epsilon^2}{n} - C_1\sqrt{m}} \leq \frac{\rho}{6}$.
- When $R = C_2 \frac{m^{3/2} \epsilon^2}{n^{1/2}}$, since $m \geq \frac{6(2C+C_1/6)}{C_2} \cdot \frac{\log^{1/2}(1/\rho)\sqrt{n}}{\rho\epsilon^2}$, we have that $\frac{2C\sqrt{m\log(1/\rho)}}{C_2 \frac{m^{3/2} \epsilon^2}{n^{1/2}} - C_1\sqrt{m}} \leq \frac{\rho}{6}$.

By a union bound, Algorithm 3 is $\rho$-replicable. $\qquad\square$

# D   Poissonization and Internal Randomness Elimination

Let $\mathcal{A}$ be a replicable tester that satisfies the correctness requirement of the corresponding testing problem. To show a sample complexity lower bound against $\mathcal{A}$, we often construct a meta-distribution $\mathcal{B}_\xi$ parametrized by a positive number $\xi \in [0, \varepsilon]$ over potential testing instances. In particular, $\mathcal{B}_\xi$ will be constructed such that $\mathcal{B}_0$ represents a collection of instances satisfying the property to be tested while $\mathcal{B}_\varepsilon$ represents ones that are "far" from satisfying the property.

Our end goal is to show that $\mathcal{A}$ cannot be $\rho$-replicable under a random choice of $\mathbf{p} \sim \mathcal{B}_\xi$, where $\xi \sim \mathcal{U}([0, \varepsilon])$, with non-trivial probability.

There are two common techniques towards the goal. Firstly, the tester is usually assumed to take a fixed number of samples from a probability distribution. Nonetheless, a common practice in distribution testing is to first show lower bounds in the so-called Poisson sampling model, which allows for the more general sampling process for pseudo-distributions, i.e., non-negative measures over the discrete domain, and is often more amenable to analyze. After that, one can use a reduction-based argument to translate the lower bound back to the standard sampling model.

**Definition D.1** (Poisson Sampling). *Given a non-negative measure $\mathbf{p}$ over $[n]$ and an integer $m$, the Poisson sampling model samples a number $m' \sim Poi(m\|\mathbf{p}\|_1)$, and draws $m'$ samples from $\mathbf{p}/\|\mathbf{p}\|_1$. Define $T \in \mathbb{R}^n$ to be the random vector where $T_i$ counts the number of element $i$ seen. We write $PoiS(m, \mathbf{p})$ to denote the distribution of the random vector $T$. We say $\mathcal{A}$ is a Poissonized tester with sample complexity $m$ if it takes as input a sample count vector $T \sim PoiS(m, \mathbf{p})$.*

Secondly, the tester $\mathcal{A}$ is in general allowed to use internal randomness. Yet, since we have already fixed the hard instance meta-distribution over the testing instances, a common approach in showing replicability lower bounds is to use a minimax style argument that allows us to fix a "good" random string $r$ such that the induced deterministic algorithm $\mathcal{A}(; r)$ enjoys about the same correctness and replicability guarantees under the meta-distribution as the original randomized algorithm. This then allows us to focus on analyzing the replicability of deterministic algorithms under $\mathbf{p} \sim \mathcal{B}_\xi$. To facilitate the discussion of the minimax argument, we introduce the notion of distributional correctness and replicability.

**Definition D.2** (Distributional Correctness/Replicability). *Let $\mathcal{B}_0, \mathcal{B}_\varepsilon, \mathcal{H}$ be meta-distributions over non-negative measures over $[n]$. Let $\mathcal{A}$ be a Poissonized tester with sample complexity $m$.*

- *We say $\mathcal{A}$ is $\delta$-correct with respect to $\mathcal{B}_0$ and $\mathcal{B}_\varepsilon$ if $\Pr_{r,\mathbf{p}\sim\mathcal{B}_0, T\sim PoiS(m,\mathbf{p})}[\mathcal{A}(T) = Accept] \geq 1 - \delta$ and $\Pr_{r,\mathbf{p}\sim\mathcal{B}_\varepsilon, T\sim PoiS(m,\mathbf{p})}[\mathcal{A}(T) = Accept] \leq \delta$.*
- *We say $\mathcal{A}$ is $\rho$-replicable with respect to $\mathcal{H}$ if it holds that $\Pr_{\mathbf{p}\sim\mathcal{H}, T, T'\sim PoiS(m,\mathbf{p})}[\mathcal{A}(T) \neq \mathcal{A}(T')] \leq \rho$.*

*The notions of distributional correctness/replicability for a non-Poissonized tester taking $m$ samples are defined similarly with the sampling process $S \sim (\mathbf{p}/\|\mathbf{p}\|_1)^{\otimes m}$ instead of $T \sim PoiS(m, \mathbf{p})$.*

To make our lower bound arguments more modular, we prove the following meta-lemma that allows us to focus on lower bounds against deterministic algorithm within the Poisson sampling model.

**Lemma D.3.** *Let $\mathcal{B}_\xi$ be a meta-distribution parametrized by a number $\xi \in (0, \varepsilon)$, $\mathcal{H}$ be a meta-distribution. Both $\mathcal{B}_\xi$ and $\mathcal{H}$ are over non-negative measures $\mathbf{p}$ over a finite universe $\mathcal{X}$ satisfying $\|\mathbf{p}\|_1 \in (0.5, 2)$. Let $\delta, \rho \in (0, 1/3)$, and $m$ be a positive integer satisfying $m \geq \log(10/\delta) + \log(10/\rho)$. Consider the following two statements:*

- *For any deterministic Poissonized tester $\mathcal{A}$ with sample complexity $m$, if $\mathcal{A}$ is $\delta$-correct with respect to $\mathcal{B}_0$ and $\mathcal{B}_\varepsilon$, then $\mathcal{A}$ cannot be $\rho$-replicable with respect to $\mathcal{H}$.*
- *For any randomized tester $\mathcal{A}$ that consumes $m' \leq m/10$ samples over $\mathcal{X}$, if $\mathcal{A}$ is $\delta/10$-correct with respect to $\mathcal{B}_0$ and $\mathcal{B}_\xi$, then $\mathcal{A}$ cannot be $\rho/10$-replicable with respect to $\mathcal{H}$.*

*The first statement implies the second statement.*

*Proof.* Let $\mathcal{A}$ be a randomized tester that consumes $m'$ samples. Consider the negation of the second statement. In particular, assume that $\mathcal{A}$ is $\delta/10$-correct with respect to $\mathcal{B}_0$ and $\mathcal{B}_\varepsilon$ as well as $\rho/10$ replicable with respect to $\mathcal{H}$. We show that this will contradict the first statement.

By Markov's inequality, with probability at least $2/3$ over the choice of the random string $r$, we have that the induced deterministic tester $\mathcal{A}(; r)$ is $0.3\delta$-correct with respect to $\mathcal{B}_0$ and $\mathcal{B}_\varepsilon$. Similarly, with probability at least $2/3$ over the choice of $r$, $\mathcal{A}(; r)$ is $0.3\rho$-replicable with respect to $\mathcal{H}$. By the union bound, with probability at least $1/3$ over the choice of $r$, $\mathcal{A}(; r)$ is at the same time $0.3\rho$-replicable with respect to $\mathcal{H}$. and $0.3\delta$-correct with respect to $\mathcal{B}_0$ and $\mathcal{B}_\varepsilon$.

We will now convert the tester into a Poissonized one. In particular, consider the Poissonized tester $\bar{\mathcal{A}}$ obtained as follows. We first take $k \sim \text{Poi}(m)$ samples from the underlying distribution $\mathbf{p}/\|\mathbf{p}\|_1$. If $k \geq m'$, we take the first $k$ samples, and feed it to $\mathcal{A}(; r)$. If $k < m'$, we simply return reject. Since we assume $\|\mathbf{p}\|_1 \in (0.5, 2)$ and $m \geq \log(10/\delta) + \log(10/\rho)$, it then follows from standard Poisson concentration that $k \geq m$ with probability at least $1 - \min(\delta, \rho)$. In particular, this implies that $\bar{\mathcal{A}}$ is a Poissonized tester with sample complexity $m$ that is at the same time $0.4\rho$-replicable w.r.t. $\mathcal{H}$ and $0.4\delta$-correct with respect to $\mathcal{B}_0$ and $\mathcal{B}_\varepsilon$. This therefore contradicts the first statement of the lemma. $\qquad\square$

# E  Omitted Proofs for Replicable Uniformity Testing Lower Bounds

In this section, we provide the omitted proofs for replicable uniformity testing. We give the proofs of Lemma 3.5, Lemma 3.9, Lemma 3.6, Proposition 3.4, and finally Theorem 1.3. Remark that since in Section 3 we assumed that $\tilde{\Theta}(\sqrt{n}\epsilon^{-2}\rho^{-1})$ dominates $\epsilon^{-2}\rho^{-2}$, throughout this section we have the implicit upper bound $\sqrt{n}\varepsilon^{-2}\rho^{-1} = \tilde{o}\left(n\varepsilon^{-2}\right)$.

## E.1  Bounding the Average Acceptance Probability for Uniformity Testing

In this subsection, we provide the proof of Lemma 3.5.

At a high level, we appeal to the same argument as in Liu and Ye [2024] to analyze the expected acceptance probability of the tester. The framework proceeds as follows. Fix any $\epsilon_0 < \epsilon_1$ in $[0, \epsilon]$ such that $\epsilon_1 - \epsilon_0 < \epsilon\rho$. Let $X$ be an unbiased random bit, $\tilde{\mathbf{p}} \sim \mathcal{M}_{\epsilon_X}$ be defined as in Definition 3.2, and $T \sim \text{PoiS}(m, \tilde{\mathbf{p}})$ be defined as in Definition D.1. Then, the mutual information between $X$ and $T$ is bounded from above by a function of the parameters $m$, $n$, $\epsilon$, and $\rho$, as stated formally in Lemma E.1. Secondly, given the mutual information bound, we know that with limited amount of samples, for any pair of $\epsilon_0, \epsilon_1 \in [0, \epsilon]$ that are $\rho\epsilon$ close to each other, $\mathbb{E}_{\mathbf{p} \sim \mathcal{M}_{\epsilon_0}}[\text{Acc}_m(\mathbf{p}, \mathcal{A})]$ and $\mathbb{E}_{\mathbf{p} \sim \mathcal{M}_{\epsilon_1}}[\text{Acc}_m(\mathbf{p}, \mathcal{A})]$ must be close to each other. See Lemma E.3 for the formal statement. Lastly, given $\mathcal{A}$ as above and is $0.1$-correct w.r.t. $\mathcal{M}_0$ and $\mathcal{M}_\epsilon$, then the acceptance probability function should satisfy that $\mathbb{E}_{\mathbf{p} \sim \mathcal{M}_0}[\text{Acc}_m(\mathbf{p}, \mathcal{A})] \geq 0.9$ and $\mathbb{E}_{\mathbf{p} \sim \mathcal{M}_\epsilon}[\text{Acc}_m(\mathbf{p}, \mathcal{A})] < 0.1$. Thus, by the mean value theorem there exists $\xi^* \in (0, \epsilon)$ such that $\mathbb{E}_{\mathbf{p} \sim \mathcal{M}_{\xi^*}}[\text{Acc}_m(\mathbf{p}, \mathcal{A})] = \frac{1}{2}$. Furthermore, from the above Lipschitzness of $\mathbb{E}_{\mathbf{p} \sim \mathcal{M}_\xi}[\text{Acc}_m(\mathbf{p}, \mathcal{A})]$ in $\xi$ we know that for at least $\rho$ fraction of $\xi \in [0, \epsilon]$, $\mathbb{E}_{\mathbf{p} \sim \mathcal{M}_\xi}[\text{Acc}_m(\mathbf{p}, \mathcal{A})] \in (1/3, 2/3)$, which concludes the proof of Lemma 3.5.

We begin by by showing the mutual information bound.

**Lemma E.1** (Mutual Information Bound for Uniformity Testing Hard Instance)**.** *Let $m = o(n\varepsilon^{-2}\log^{-2} n)$, $\epsilon_0 < \epsilon_1 \in [0, \epsilon]$ be such that $\epsilon_1 - \epsilon_0 < \epsilon\rho$, $X$ be an unbiased random bit,*

$\mathcal{M}_{\epsilon_X}$ be the distribution over measures defined as in Definition 3.2, $\tilde{\mathbf{p}} \sim \mathcal{M}_{\epsilon_X}$, $T \sim PoiS(m, \tilde{\mathbf{p}})$. Then the mutual information $I(X : T_1, \cdots, T_n)$ satisfies:

$$I(X : T_1, \cdots, T_n) = O\left(\frac{m^2}{n}\epsilon^4\rho^2\log^4 n\right) + o(1).$$

*Proof of Lemma E.1.* Let $\delta := \epsilon_1 - \epsilon_0 = O(\epsilon\rho)$. Since $M_i's$ are conditionally independent conditioned on $X$, we have that

$$I(X : T_1, \cdots, T_n) \le \sum_{i=1}^{n} I(X : T_i) = nI(X : T_1).$$

Therefore, it suffices show that $I(X : T_1) = O\left(\frac{m^2}{n^2}\epsilon^4\rho^2\log^4 n\right) + o\left(\frac{1}{n}\right).$

We start by expanding the conditional probabilities of $T_1$ conditioned on value of $X$.

$$T_1|(X = 0) \sim \frac{1}{2}\text{Poi}\left(\frac{m}{n}(1 + \epsilon_0)\right) + \frac{1}{2}\text{Poi}\left(\frac{m}{n}(1 - \epsilon_0)\right),$$

and similarly,

$$T_1|(X = 1) \sim \frac{1}{2}\text{Poi}\left(\frac{m}{n}(1 + \epsilon_1)\right) + \frac{1}{2}\text{Poi}\left(\frac{m}{n}(1 - \epsilon_1)\right),$$

then we can expand $\Pr[T_1 = a|X = 0]$ and $\Pr[T_1 = a|X = 1]$ accordingly. Indeed,

$$\Pr[T_1 = a|X = 0] = \frac{1}{2a!}\left(\frac{m}{n}\right)^a(1 + \epsilon_0)^a\exp\left(-\frac{m}{n}(1 + \epsilon_0)\right) + \frac{1}{2}\frac{1}{a!}\left(\frac{m}{n}\right)^a(1 - \epsilon_0)^a\exp\left(-\frac{m}{n}(1 - \epsilon_0)\right)$$

$$= \frac{1}{2a!}\left(\frac{m}{n}\right)^a\exp\left(-\frac{m}{n}\right)\left(\exp\left(-\frac{m\epsilon_0}{n}\right)(1 + \epsilon_0)^a + \exp\left(\frac{m\epsilon_0}{n}\right)(1 - \epsilon_0)^a\right),$$

$$\Pr[T_1 = a|X = 1] = \frac{1}{2a!}\left(\frac{m}{n}\right)^a\exp\left(-\frac{m}{n}\right)\left(\exp\left(-\frac{m(\epsilon_0 + \delta)}{n}\right)(1 + \epsilon_0 + \delta)^a + \exp\left(\frac{m(\epsilon_0 + \delta)}{n}\right)(1 - \epsilon_0 - \delta)^a\right).$$

Since $\delta = o(\epsilon)$, $\Pr[T_1 = a|X = 0] = \Theta\left(\Pr[T_1 = a|X = 1]\right)$. By Claim A.2, it suffices to show that $\bar{I} := \sum_a \frac{(\Pr[T_1=a|X=0]-\Pr[T_1=a|X=1])^2}{\Pr[T_1=a|X=0]+\Pr[T_1=a|X=1]} = O\left(\frac{m^2}{n^2}\epsilon^4\rho^2\log^4 n\right) + o\left(\frac{1}{n}\right).$

Let

$$f_a(y) := \exp\left(-\frac{my}{n}\right)(1 + y)^a + \exp\left(\frac{my}{n}\right)(1 - y)^a. \tag{11}$$

Then it holds

$$\bar{I} = O(1)\sum_{a=0}^{\infty}\frac{1}{a!}\left(\frac{m}{n}\right)^a\exp\left(-\frac{m}{n}\right)\frac{(f_a(\epsilon_0) - f_a(\epsilon_0 + \delta))^2}{f_a(\epsilon_0) + f_a(\epsilon_0 + \delta)} =: O(1)\sum_{a=0}^{\infty}\bar{I}_a,$$

where for simplicity, denote $\bar{I}_a := \frac{1}{a!}\left(\frac{m}{n}\right)^a\frac{(f_a(\epsilon_0)-f_a(\epsilon_0+\delta))^2}{f_a(\epsilon_0)+f_a(\epsilon_0+\delta)}$. Then by the mean value theorem $\bar{I}_a \le \delta^2\frac{\max_{y\in[\epsilon_0,\epsilon_0+\delta]}\left(\frac{\partial}{\partial y}f_a(y)\right)^2}{f_a(\epsilon_0)+f_a(\epsilon_0+\delta)}$, whence to bound $\bar{I}_a$ from above, it suffices to bound the denominator of RHS from below and the numerator of RHS from above separately. We next break into 3 cases depending on the size of $\frac{m}{n}$.

**Case 1:** For the sublinear regime, i.e., $\frac{m}{n} \le 1/2$, we break into 3 cases depending on the value of $a$.

when $a = 0$, applying the mean value theorem gives that $|f_0(\epsilon_0) - f_0(\epsilon_0 + \delta)| \le -\frac{m}{n}\delta\frac{2m(\epsilon_0+\delta)}{n}\exp\left(\frac{m(\epsilon_0+\delta)}{n}\right)$. Since $f_0(\varepsilon_0) + f_0(\varepsilon_1) = \Omega(1)$, we have that $\bar{I}_0 = O(1)(f_0(\epsilon_0) - f_0(\epsilon_0 + \delta))^2 = O\left(\frac{m^4}{n^4}\epsilon_0^2\delta^2\right) = O\left(\frac{m^2}{n^2}\epsilon_0^2\delta^2\right).$

when $a = 1$,

$$|f_1(\epsilon_0) - f_1(\epsilon_0 + \delta)|$$
$$\leq |f_0(\epsilon_0) - f_0(\epsilon_0 + \delta)|$$
$$+ \left| \epsilon_0 \left( \exp\left(\frac{m\epsilon_0}{n}\right) - \exp\left(-\frac{m\epsilon_0}{n}\right) \right) - (\epsilon_0 + \delta) \left( \exp\left(-\frac{m(\epsilon_0 + \delta)}{n}\right) + \exp\left(\frac{m(\epsilon_0 + \delta)}{n}\right) \right) \right|$$
$$\leq O\left(\frac{m^2}{n^2}\epsilon_0\delta\right) + \epsilon_0 \left| \exp\left(-\frac{m\epsilon_0}{n}\right) - \exp\left(-\frac{m(\epsilon_0 + \delta)}{n}\right) \right| + \epsilon_0 \left| \exp\left(\frac{m(\epsilon_0 + \delta)}{n}\right) - \exp\left(\frac{m\epsilon_0}{n}\right) \right|$$
$$+ \delta \left| \exp\left(\frac{m(\epsilon_0 + \delta)}{n}\right) - \exp\left(-\frac{m(\epsilon_0 + \delta)}{n}\right) \right|$$
$$= O\left(\frac{m}{n}\epsilon_0\delta\right). \qquad \text{(the mean value theorem)}$$

Combining with the fact that $f_1(\varepsilon_0) + f_1(\varepsilon_1) = \Omega(1)$, we have that $\bar{I}_1 = O(1)(f_1(\epsilon_0) - f_1(\epsilon_0 + \delta))^2 = O\left(\frac{m^2}{n^2}\epsilon^2\delta^2\right)$.

when $a \geq 2$,

$$\frac{\partial}{\partial y}f_a(y) = -\frac{m}{n}\exp\left(-\frac{my}{n}\right)(1 + y)^a + a(1 + y)^{a-1}\exp\left(-\frac{my}{n}\right)$$
$$+ \frac{m}{n}\exp\left(\frac{my}{n}\right)(1 - y)^a - a(1 - y)^{a-1}\exp\left(\frac{my}{n}\right).$$

Before bounding $\left|\frac{\partial}{\partial y}f_a(y)\right|$, we introduce a technical claim that is helpful in the rest of the proof to show the monotonicity of specific families of functions.

**Claim E.2.** *For $a, b, s, d, x \in \mathbb{R}$, when $s + dx \geq 0$, if $dk \geq (\leq, resp.)b(s + dx)$ then $\exp(a - bx)(s + dx)^k$ is nondecreasing(nonincreasing, resp.) as a function of $x$.*

*Proof of Claim E.2.* $\frac{\partial}{\partial x}\left(\exp(a - bx)(s + dx)^k\right) = \exp(a - bx)(s + dx)^{k-1}[dk - b(s + dx)]$, then if $dk \geq b(s + dx)$, we have that $\frac{\partial}{\partial x}\left(\exp(a - bx)(s + dx)^k\right) \geq 0$. Similar argument applies when $dk \leq b(s + dx)$. $\qquad\square$

When $y \in [\epsilon_0, \epsilon_0 + \delta]$, by Claim E.2, since $a - 1 \geq 1 \geq (1 + y)\frac{m}{n}$,

$$\left|\frac{\partial}{\partial y}f_a(y)\right| \leq \exp\left(-\frac{my}{n}\right)\left[\frac{m}{n}\left((1 + y)^a - (1 - y)^a\right) + a\left((1 + y)^{a-1} - (1 - y)^{a-1}\right)\right]$$
$$\leq \frac{m}{n}a(1 + y)^{a-1}2y + a(a - 1)(1 + y)^{a-2}2y = O\left(2^a a^2 y\right).$$

Thus, we have that $\max_{y \in [\epsilon_0, \epsilon_0 + \delta]}\left(\frac{\partial}{\partial y}f_a(y)\right)^2 = \max_{y \in [\epsilon_0, \epsilon_0 + \delta]}\left(O\left(2^{2a}a^4 y^2\right)\right) = O\left(4^a a^4 \epsilon^2\right)$. Since $f_a(\epsilon_0) + f_a(\epsilon_0 + \delta) = \Omega(1)$,

$$\sum_{a=2}^{\infty}\bar{I}_a \leq O(\delta^2\epsilon^2)\sum_{a=2}^{\infty}\frac{4^a a^4}{a!}\left(\frac{m}{n}\right)^a.$$

Since $\sum_{a=2}^{\infty}\frac{4^a a^4}{a!}$ is a converging series, it can be bounded by $O(1)$. Therefore, $\sum_{a=2}^{\infty}\bar{I}_a \leq O(\delta^2\epsilon^2)\sum_{a=2}^{\infty}\left(\frac{m}{n}\right)^a = O\left(\frac{m^2}{n^2}\delta^2\epsilon^2\right)$.

In conclusion, from the above three cases, $\bar{I} = O\left(\frac{m^2}{n^2}\delta^2\epsilon^2\right)$.

**Case 2:** For the superlinear regime when $n/2 \leq m \leq o\left(\frac{n}{\epsilon^2 \log^2 n}\right)$, we start by noticing that when $a$ deviates far enough from $\frac{m}{n}$, the sum of all such $\bar{I}_a$ is negligible. More specifically, let $\lambda = \frac{m(1-\epsilon-\delta)}{n}$,

and $c > 0$ be a constant such that $\exp\left(-\frac{x^2}{2(\lambda+x)}\right)\Big|_{x=c\log n\sqrt{m/n}} \le \frac{1}{n^2}$ then by Lemma A.1,

$$\sum_{\substack{a \ge \lfloor \lambda + c\log n\sqrt{m/n}\rfloor \\ a \le \lceil \lambda - c\log n\sqrt{m/n}\rceil}} \frac{(\Pr[T_1 = a|X=0] - \Pr[T_1 = a|X=1])^2}{\Pr[T_1 = a|X=0] + \Pr[T_1 = a|X=1]} = o\left(\frac{1}{n}\right).$$

Therefore, to compute $\bar{I}$, it suffices to consider $\bar{I}_a$ when $a \in \left[\lambda - c\log n\sqrt{m/n}, \lambda + c\log n\sqrt{m/n}\right]$. Instead of directly bounding $|f_a(\epsilon_0) - f_a(\epsilon_1)|$, we separate $f_a(y)$ into parts. In particular, by the Taylor expansion of $\exp(x)$

$$f_a(y) = \exp\left(-\frac{m}{n}y + a\log(1+y)\right) + \exp\left(\frac{m}{n}y + a\log(1-y)\right)$$

$$= 2 + a\log(1-y^2) + \sum_{i=2}^{\infty} \frac{1}{i!}\left(\left(-\frac{m}{n}y + a\log(1+y)\right)^i + \left(\frac{m}{n}y + a\log(1-y)\right)^i\right).$$

Define $g_a(y) := a\log(1-y^2)$ and $h_a(y) := \sum_{i=2}^{\infty} \frac{1}{i!}\left(-\frac{m}{n}y + a\log(1+y)\right)^i$. Then it follows that $f_a(y) = 2 + g_x(y) + h_a(y) + h_a(-y)$.

On one hand,

$$|g_a(\epsilon_0) - g_a(\epsilon_1)| = \left|a\log\left(\frac{1-\epsilon_0^2}{1-\epsilon_1^2}\right)\right| \le a\left|1 - \frac{1-\epsilon_0^2}{1-\epsilon_1^2}\right| \qquad (\text{for } x \ge 1, |\log(x)| \le x - 1)$$

$$= \frac{a}{1-\epsilon_1^2}(\epsilon_0 + \epsilon_1)|\epsilon_0 - \epsilon_1|$$

$$= O\left(\frac{m}{n}\epsilon\delta\log n\right). \tag{12}$$

On the other hand,

$$\frac{\partial}{\partial y}h_a(y) = \left(-\frac{m}{n} + \frac{a}{1+y}\right)\sum_{i=2}^{\infty} \frac{1}{(i-1)!}\left(-\frac{m}{n}y + a\log(1+y)\right)^{i-1}$$

$$= \left(-\frac{m}{n} + \frac{a}{1+y}\right)\left(\exp\left(-\frac{m}{n}y\right)(1+y)^x - 1\right).$$

When $y \in [\epsilon_0, \epsilon_1]$,

$$\left|-\frac{m}{n} + \frac{a}{1+y}\right| \le \left|-\frac{m}{n} + \frac{1}{1+y}\left(\frac{m}{n}(1-\epsilon_0 - \delta) + c\log n\sqrt{m/n}\right)\right|$$

$$\le \left|\frac{-y-\epsilon_0-\delta}{1+y}\frac{m}{n}\right| + \frac{c}{1+y}\log n\sqrt{m/n}$$

$$= O(1)\left(\frac{m}{n}\epsilon_0 + \log n\sqrt{m/n}\right) = O\left(\log n\sqrt{m/n}\right),$$

where the last equality follows from the fact that $\sqrt{\frac{m}{n}} = o\left(\frac{1}{\epsilon\log n}\right) = o\left(\frac{\log n}{\epsilon}\right)$ implies that $\log n\sqrt{m/n} \gg \frac{m}{n}\epsilon$. For $\left|\exp\left(-\frac{m}{n}y\right)(1+y)^a - 1\right|$, from Claim E.2, since $a > m/n(1+y)$, $\exp\left(-\frac{my}{n}\right)(1+y)^a$ nondecreasing and takes value 1 when $y = 0$. Therefore, we can remove absolute value directly, which gives that

$$\left|\exp\left(-\frac{m}{n}y\right)(1+y)^a - 1\right| \le \left(\sum_{i=0}^{\infty}\frac{y^i}{i!}\right)^{-m/n}(1+y)^a - 1 \le (1+y)^{-m/n}(1+y)^x - 1$$

$$\le 2(1+y)^{\log n\sqrt{m/n}} - 1 \le \sum_{k=1}^{\lceil\log n\sqrt{m/n}\rceil}\binom{\lceil\log n\sqrt{m/n}\rceil}{k}y^k$$

$$\le O(1)\sum_{k=1}^{\lceil\log n\sqrt{m/n}\rceil}\left(\frac{\epsilon e\log n\sqrt{m/n}}{k}\right)^k.$$

To show that $O\left(\epsilon \log n \sqrt{m/n}\right)$ dominates the last term, it suffices to show that $\frac{\epsilon e \log n \sqrt{m/n}}{k} = o(1)$ i.e. $\epsilon = o\left(\frac{1}{\log n \sqrt{m/n}}\right)$, which is equivalent to showing that $\sqrt{m/n} = o\left(\frac{1}{\log n\epsilon}\right)$. This is true if and only if $m = o\left(\frac{n}{\epsilon^2 \log^2 n}\right)$ as assumed in the premise.

Therefore, we have that $\max_{y \in [\epsilon_0, \epsilon_1]} \left|\frac{\partial}{\partial y} h_a(y)\right| = O\left(\epsilon \log^2 n \frac{m}{n}\right)$. By the mean value theorem, we have that

$$|h_a(\epsilon_0) - h_a(\epsilon_1)| = O\left(\epsilon \delta \log^2 n \frac{m}{n}\right), \tag{13}$$

and

$$|h_a(-\epsilon_0) - h_a(-\epsilon_1)| = O\left(\epsilon \delta \log^2 n \frac{m}{n}\right). \tag{14}$$

Combining Equation (12), Equation (13), and Equation (14), we have that

$$|f_a(\epsilon_0) - f_a(\epsilon_1)| = O\left(\epsilon \delta \log^2 n \frac{m}{n}\right). \tag{15}$$

We now consider bounding from below the denominator $f_a(\epsilon_0) + f_a(\epsilon_1)$. Recall that from Equation (11), we have that

$$f_a(y) = \exp\left(-\frac{my}{n}\right)(1+y)^a + \exp\left(\frac{my}{n}\right)(1-y)^a.$$

Since $1 + x \geq \exp(x - x^2)$, we have that $(1+y)^a \geq \exp(ay - ay^2)$. This implies that

$$
\begin{aligned}
f_a(\epsilon_0) + f_a(\epsilon_1) &\geq \Omega(1) \exp\left((\epsilon + \delta)\left(-\frac{m}{n} + a\right) - a(\epsilon + \delta)^2\right) \\
&= \Omega(1) \exp\left(-\frac{m}{n}(\epsilon + \delta)^2 - (\epsilon + \delta)c \log n \sqrt{m/n} - a(\epsilon + \delta)^2\right) \\
&= \Omega(1) \exp\left(-\frac{m}{n}\epsilon^2 - \epsilon \log n \sqrt{m/n} - \epsilon^2 \log n \sqrt{m/n}\right).
\end{aligned}
$$

Since $\frac{m}{n} = o\left(\frac{1}{\epsilon^2 \log^2 n}\right)$, we have that

$$f_a(\epsilon_0) + f_a(\epsilon_1) \geq \Omega(1) \frac{1}{\exp(o(1))} = \Omega(1). \tag{16}$$

Hence, by Equation (15) and Equation (16),

$$
\begin{aligned}
\bar{I} &= O(1) \sum_{a=0}^{\infty} \frac{1}{a!} \left(\frac{m}{n}\right)^a \exp\left(-\frac{m}{n}\right) \frac{(f_a(\epsilon_0) - f_a(\epsilon_0 + \delta))^2}{f_a(\epsilon_0) + f_a(\epsilon_0 + \delta)} \\
&= O\left(\epsilon^2 \delta^2 \log^4 n \frac{m^2}{n^2}\right) \sum_{\substack{a \geq \lfloor \lambda + c \log n \sqrt{m/n} \rfloor \\ a \leq \lceil \lambda - c \log n \sqrt{m/n} \rceil \\ a \in \mathbb{Z}}} \frac{1}{a!} \left(\frac{m}{n}\right)^a \exp\left(-\frac{m}{n}\right) + o\left(\frac{1}{n}\right).
\end{aligned}
$$

Since $\sum_{a = \lfloor \lambda + c \log n \sqrt{m/n} \rfloor}^{\lceil \lambda - c \log n \sqrt{m/n} \rceil} \frac{1}{a!} \left(\frac{m}{n}\right)^a < \sum_{a=0}^{\infty} \frac{1}{a!} \left(\frac{m}{n}\right)^a = \exp(m/n)$, this term cancels out with the succeeding $\exp(-m/n)$ term. Thus

$$\bar{I} \leq O\left(\epsilon^2 \delta^2 \log^4 n \frac{m^2}{n^2}\right) + o\left(\frac{1}{n}\right)$$

as desired.

The above 2 cases conclude the proof of Lemma E.1. $\square$

The second step of the framework is as follows.

**Lemma E.3** (Lipschitzness of Expected Acceptance Probability). *Assume that $m = \tilde{o}(\sqrt{n}\epsilon^{-2}\rho^{-1})$. Let $\mathcal{A}$ be a deterministic tester that takes a sample-count vector over $[n]$ as input. Let $\epsilon_0 < \epsilon_1 \in [0, \epsilon]$ be such that $\epsilon_0 - \epsilon_1 \leq \epsilon\rho$. Then it holds that*

$$|\mathbb{E}_{\mathbf{p}\sim\mathcal{M}_{\epsilon_0}}[Acc_m(\mathbf{p}, \mathcal{A})] - \mathbb{E}_{\mathbf{p}\sim\mathcal{M}_{\epsilon_1}}[Acc_m(\mathbf{p}, \mathcal{A})]| < 0.1\,,$$

*where $\mathrm{Acc}_m$ is the acceptance probability function defined as $\mathrm{Acc}_m(\mathbf{p}, \mathcal{A}) := \Pr_{T\sim\mathrm{PoiS}(m,\mathbf{p})}[\mathcal{A}(T) = \mathrm{Accept}]$.*

*Proof of Lemma E.3.* Assume for the sake of contradiction that $|\mathbb{E}_{\mathbf{p}\sim\mathcal{M}_{\epsilon_0}}[\mathrm{Acc}_m(\mathbf{p}, \mathcal{A})] - \mathbb{E}_{\mathbf{p}\sim\mathcal{M}_{\epsilon_1}}[\mathrm{Acc}_m(\mathbf{p}, \mathcal{A})]| \geq 0.1$. Let $X$ be an unbiased random bit, and $Y$ be the random variable defined as follows: let $\mathbf{p} \sim \mathcal{M}_{\epsilon_X}$, $T \sim \mathrm{PoiS}(m, \mathbf{p})$, then $Y = 1$ if $\mathcal{A}(T)$ accept, $Y = 0$ otherwise. It follows from the definition and the assumption that $\Pr[Y = 1|X = 0] = \mathbb{E}_{\mathbf{p}\sim\mathcal{M}_{\epsilon_0}}[\mathrm{Acc}_m(\mathbf{p}, \mathcal{A})]$ and $\Pr[Y = 1|X = 1] = \mathbb{E}_{\mathbf{p}\sim\mathcal{M}_{\epsilon_1}}[\mathrm{Acc}_m(\mathbf{p}, \mathcal{A})]$, which implies a mutual information bound of $I(X : T) \geq I(X : Y) = \Omega(1)$. This clearly contradicts the result from Lemma E.1, and hence concludes the proof of Lemma E.3. $\qquad\square$

We are ready to show the main result of this subsection.

*Proof of Lemma 3.5.* Since $\mathcal{A}$ is $0.1$-correct w.r.t. $\mathcal{M}_0$ and $\mathcal{M}_\varepsilon$, we have that $\mathbb{E}_{\mathbf{p}\sim\mathcal{M}_0}[\mathrm{Acc}_m(\mathbf{p}, \mathcal{A})] \geq 0.9$ and $\mathbb{E}_{\mathbf{p}\sim\mathcal{M}_\epsilon}[\mathrm{Acc}_m(\mathbf{p}, \mathcal{A})] < 0.1$. Furthermore, since $\mathbb{E}_{\mathbf{p}\sim\mathcal{M}_\xi}[\mathrm{Acc}_m(\mathbf{p}, \mathcal{A})]$ is a polynomial in $\xi$, it is continuous in $\xi$. Hence, by the mean value theorem, there exists $\xi^* \in (0, \epsilon)$ such that $\mathbb{E}_{\mathbf{p}\sim\mathcal{M}_{\xi^*}}[\mathrm{Acc}_m(\mathbf{p}, \mathcal{A})] = 1/2$. It follows immediately from E.3 that $\forall \xi \in [\xi^* - \rho\epsilon, \xi^* + \rho\epsilon]$ we have that

$$\mathbb{E}_{\mathbf{p}\sim\mathcal{M}_\xi}[\mathrm{Acc}_m(\mathbf{p}, \mathcal{A})] \in \left(\mathbb{E}_{\mathbf{p}\sim\mathcal{M}_{\xi^*}}[\mathrm{Acc}_m(\mathbf{p}, \mathcal{A})] - 0.1, \mathbb{E}_{\mathbf{p}\sim\mathcal{M}_{\xi^*}}[\mathrm{Acc}_m(\mathbf{p}, \mathcal{A})] + 0.1\right)$$
$$= (0.4, 0.6) \subset (1/3, 2/3).$$

In conclusion, if we uniformly at randomly select a $\xi \in [0, \epsilon]$, then once it falls in interval $[\xi^* - \rho\epsilon, \xi^* + \rho\epsilon]$ of length $2\rho\epsilon$, which happens with probability $2\rho\varepsilon$, we have that $\mathbb{E}_{\mathbf{p}\sim\mathcal{M}_\xi}[\mathrm{Acc}_m(\mathbf{p}, \mathcal{A})] \in (1/3, 2/3)$ as desired. $\qquad\square$

## E.2  Concentration of Acceptance Probability

In this subsection, we prove Lemma 3.6. Throughout this section we identify the Accept outcome with 1 so that $\mathrm{Acc}_m(\mathbf{p}, \mathcal{A}) = \Pr_{T\sim\mathrm{PoiS}(m,\mathbf{p})}[\mathcal{A}(T) = \mathrm{Accept}] = \mathbb{E}_{T\sim\mathrm{PoiS}(m,\mathbf{p})}[\mathcal{A}(T)]$.

A key technical result in this section is to bound the mixing time of the random walk $\mathbf{RW}_{m,\mathcal{M}_\xi}$, which we abbreviate as $\mathbf{RW}_{m,\xi}$ within this section for convenience.

**Lemma 3.8.** *Let $\xi \in (0, \varepsilon)$ and $m = \tilde{o}(n\varepsilon^{-2})$. Then $\mathbf{RW}_{m,\xi}$ has mixing time $\tau(\delta) = O(\log(n/\delta))$.*

We analyze the transition probability of the random walk $\mathbf{RW}_{m,\xi}$. We first note that drawing $\mathrm{Poi}\,(m)$ samples from $\mathbf{p} \sim \mathcal{M}_\xi$ is equivalent as drawing $\mathrm{Poi}\,(m\mathbf{p}_i)$ samples from each bucket independently where $\mathbf{p}_i \in \{\frac{1+\xi}{n}, \frac{1-\xi}{n}\}$ is the mass of bucket $i$ under the measure $\mathbf{p}$. Since the observed count of each bucket $i \in [n]$ is independent, we may decompose the random walk $\mathbf{RW}_{m,\xi}$ as a product of $n$ independent random walks.

**Definition E.4.** *The Coordinate Sample Random Walk $\mathbf{RW}_{m,\xi,i}$ is defined on the graph whose vertex set is $\mathbb{N}$ and transitions $(T_1[i], T_2[i])$ are defined by the conditional distribution of $T_2[i]$ given $T_1[i]$ induced by the joint distribution given by the following process:*

1. *Choose $\mathbf{p}_i \in \mathcal{U}\left(\left\{\frac{1+\xi}{n}, \frac{1-\xi}{n}\right\}\right)$.*
2. *$T_1[i], T_2[i]$ are sampled independently from $\mathrm{PoiS}(m, \mathbf{p}_i)$.*

*Given a sample count $T[i]$, we denote $\mathbf{RW}^k_{m,\xi,i}(T[i])$ the random variable representing the outcome after $k$ steps of random walk from $T$. For $\mathbf{p}_i \geq 0$, we denote by $\mathbf{RW}^k_{m,\xi,i}(\mathbf{p}_i)$ the distribution of $\mathbf{RW}^k_{m,\xi,i}(T[i])$ where $T[i] \sim \mathrm{PoiS}(m, \mathbf{p}_i)$.*

By independence, we have that

$$\mathbf{RW}_{m,\xi} = \prod_{i=1}^{n} \mathbf{RW}_{m,\xi,i}.$$

For a sample $T$, let $T[i]$ denote the empirical frequency of the $i$-th bucket in $T$. Let $T_1 \sim \mathbf{RW}_{m,\xi,i}(T_0)$. We can write the joint distribution of $T_0, T_1$ as

$$\Pr(T_0[i] = a, T_1[i] = b) = \frac{\left(e^{-2m(1+\xi)/n}\frac{((1+\xi)m/n)^{a+b}}{a!b!} + e^{-2m(1-\xi)/n}\frac{((1-\xi)m/n)^{a+b}}{a!b!}\right)}{2}$$

$$= \frac{1}{2a!b!}e^{-2m/n}\left(\frac{m}{n}\right)^{a+b}\left(e^{-2\xi m/n}(1+\xi)^{a+b} + e^{2\xi m/n}(1-\xi)^{a+b}\right).$$

Furthermore, we have that

$$\Pr(T_0[i] = a) = \sum_{b=0}^{\infty} \frac{\left(e^{-2(1+\xi)m/n}\frac{((1+\xi)m/n)^{a+b}}{a!b!} + e^{-2(1-\xi)m/n}\frac{((1-\xi)m/n)^{a+b}}{a!b!}\right)}{2}$$

$$= \frac{e^{-(1+\xi)m/n}((1+\xi)m/n)^a + e^{-(1-\xi)m/n}((1-\xi)m/n)^a}{2a!}$$

$$= \frac{1}{2a!}e^{-m/n}\left(\frac{m}{n}\right)^a\left(e^{-\xi m/n}(1+\xi)^a + e^{\xi m/n}(1-\xi)^a\right)$$

Combining the two gives that the probability of the transition $P(a,b) := \Pr(T_1[i] = b | T_0[i] = a)$ is

$$P(a,b) = \frac{\Pr(T_1[i] = b, T_0[i] = a)}{\Pr(T_0[i] = a)}$$

$$= \frac{1}{b!}e^{-m/n}\left(\frac{m}{n}\right)^b\left(\frac{e^{-2\xi m/n}(1+\xi)^{a+b} + e^{2\xi m/n}(1-\xi)^{a+b}}{e^{-\xi m/n}(1+\xi)^a + e^{\xi m/n}(1-\xi)^a}\right).$$

This defines the random walk $\mathbf{RW}_{m,\xi,i}$ for each $i \in [n]$ with transition probabilities given above. Given $\mathbf{RW}_{m,\xi,i}$ for all $i$, we can write the transition probability of $\mathbf{RW}_{m,\xi}$ from $a = (a_1, \ldots, a_n)$ to $b = (b_1, \ldots, b_n)$ as

$$\Pr(\mathbf{RW}_{m,\xi}(a) = b) = \prod_{i=1}^{n} \Pr\left(T_1[i] = b_i | T_0[i] = a_i\right)$$

$$= e^{-m} \prod_{i=1}^{n} \frac{1}{b_i!}\left(\frac{m}{n}\right)^{b_i}\left(\frac{e^{-2\xi m/n}(1+\xi)^{a_i+b_i} + e^{2\xi m/n}(1-\xi)^{a_i+b_i}}{e^{-\xi m/n}(1+\xi)^{a_i} + e^{\xi m/n}(1-\xi)^{a_i}}\right).$$

In particular, the stationary distribution of $\mathbf{RW}_{m,\xi}$ is the vector $\pi \in [m]^n$ given by

$$\pi((a_1, \ldots, a_n)) = \prod_{i=1}^{n} \Pr(T_0[i] = a_i)$$

$$= e^{-m} \prod_{i=1}^{n} \frac{1}{2a_i!}\left(\frac{m}{n}\right)^{a_i}\left(e^{-\xi m/n}(1+\xi)^{a_i} + e^{\xi m/n}(1-\xi)^{a_i}\right).$$

It is not hard to see that our random walk is ergodic and reversible.

**Lemma E.5.** *The random walk $\mathbf{RW}_{m,\xi}$ is ergodic and reversible.*

*Proof.* The random walk $\mathbf{RW}_{m,\xi}$ is ergodic since every transition is possible (including self-loops). Furthermore, $\mathbf{RW}_{m,\xi}$ is reversible since $\pi(i)P(i,j)$ is a joint distribution that is the same as $\pi(j)P(j,i)$. $\qquad\square$

We proceed to show that $\mathbf{RW}_{m,\xi,i}$ mixes rapidly.

**Lemma E.6.** *Suppose $m = o(n/\varepsilon^2)$. The random walk $\mathbf{RW}_{m,\xi,i}$ has mixing time $\tau(0.04) \leq 2$.*

*Proof.* Note that the random walk $\mathbf{RW}_{m,\xi,i}$ has transition probabilities from $Y_0 \geq 0$ to $Y_1 \geq 0$ given by the conditional distribution induced by the following joint distribution.

1. Let $X \sim \mathcal{U}(\{0,1\})$ be a uniformly random bit.
2. Independently sample $Y_0, Y_1 \sim \text{Poi}\left(m(1-\xi)/n\right)$ if $X = 0$ and otherwise sample $Y_0, Y_1 \sim \text{Poi}\left(m(1+\xi)/n\right)$ if $X = 1$.

A useful fact is that the total variation distance between $\text{Poi}\left(m(1-\xi)/n\right)$ and $\text{Poi}\left(m(1+\xi)/n\right)$ is small.

**Claim E.7.** *Let $\lambda_1 > \lambda_2 > 0$. Let $X \sim \text{Poi}(\lambda_1)$ and $Y \sim \text{Poi}(\lambda_2)$. Then*

$$d_{TV}(X,Y) \leq \sqrt{\frac{(\lambda_1 - \lambda_2)^2}{2\lambda_2}}.$$

*Proof.* We begin by bounding the KL-divergence as

$$D_{\mathrm{KL}} = \lambda_1 \log \frac{\lambda_1}{\lambda_2} + \lambda_2 - \lambda_1 \leq \lambda_1 \left(\frac{\lambda_1 - \lambda_2}{\lambda_2}\right) + \lambda_2 - \lambda_1 \leq \frac{(\lambda_1 - \lambda_2)^2}{\lambda_2}$$

where we have used $\log x \leq x - 1$ for $x > 0$. Now, using Pinsker's inequality, we can bound

$$d_{\mathrm{TV}}(X,Y) \leq \sqrt{\frac{(\lambda_1 - \lambda_2)^2}{2\lambda_2}}.$$

This concludes the proof of Claim E.7. $\qquad\square$

We handle the sub-linear and super-linear cases separately.

**Sub-linear Case: $m \leq n$.** Note that Claim E.7 implies that the total variation distance between $Z_0 \sim \text{Poi}\left(m(1-\xi)/n\right)$ and $Z_1 \sim \text{Poi}\left(m(1+\xi)/n\right)$ is at most

$$\sqrt{\frac{(2m\xi/n)^2}{2m(1-\xi)/n}} \leq \sqrt{\frac{2m\xi^2/n}{1-\xi}} \leq 2\xi.$$

where in the final inequality we have used $m/n \leq 1$ and $1 - \xi > 0.5$.

Now consider a step of the random walk from initial state $Y_0 = \ell$. The distribution of $Y_1$ is given by the mixture of two Poisson distributions

$$\Pr(Y_1 = k | Y_0 = \ell) = \Pr(X = 0 | Y_0 = \ell) \Pr(Z_0 = k) + \Pr(X = 1 | Y_0 = \ell) \Pr(Z_1 = k).$$

The total variation distance between this distribution and the stationary distribution $\pi$ is at most

$$\frac{1}{2} \sum_k |\Pr(Y_1 = k | Y_0 = \ell) - \pi(k)| \leq 2 \left|\Pr(X = 0 | Y_0 = \ell) - \frac{1}{2}\right| \xi \leq 2\xi.$$

Now, consider a step from $Y_1 = k$. By the total variation distance bound, we can conclude that an algorithm cannot distinguish $X = 0$ with advantage better than $2\xi < 0.2$. Therefore, we can bound $0.3 \leq \Pr(X = 0 | Y_1 = k) \leq 0.7$ otherwise the algorithm that returns $X$ with this conditional probability is a distinguisher. From our previous calculation we conclude that after two steps, the random walk mixes to within $0.4\xi < 0.04$ of the stationary distribution.

**Super-linear Case:** $n \leq m = o\left(\frac{n}{\varepsilon^2}\right)$**.** Following similar arguments as in the sub-linear case, the total variation distance between $Z_0, Z_1$ is at most $o\left(1\right) < 0.1$. As in the sub-linear case, no algorithm can distinguish $X = 0$ with advantage better than the total variation distance $0.1$. Since $\Pr(X = 0 | Y_1 = k) \leq 0.6$, we can conclude that the random walk mixes within $0.01$ of the stationary distribution in two steps. This concludes the proof of Lemma E.6. $\quad\square$

Given Lemma E.6, we can bound the relaxation time of each coordinate random walk via Theorem A.10. In particular, we have that for $\mathbf{RW}_{m,\xi,i}$, $t_{\text{rel}} \leq \frac{\tau(\delta)}{\log(1/2\delta)} + 1$. Combining this with Lemma E.6 gives that

$$t_{\text{rel}} \leq \frac{\tau(0.04)}{\log(1/0.08)} + 1 \leq \frac{2}{\log(1/0.08)} + 1 = O(1).$$

We are now ready to bound the mixing time of the product random walk $\mathbf{RW}_{m,\xi}$.

**Lemma E.8.** *Let $m = o\left(\frac{n}{\varepsilon^2}\right)$. Let $\gamma(x) \sim \mathrm{Poi}\left((1+\xi)m/n\right)$ or $\gamma(x) \sim \mathrm{Poi}\left((1-\xi)m/n\right)$ denote the initial distribution. Then, under either initial distribution $\gamma$, $\mathbf{RW}_{m,\xi,i}$ has mixing time:*
$$\tau_i(\delta) = O(\log(1/\delta)).$$

*Proof.* Consider a coordinate random walk $\mathbf{RW}_{m,\xi,i}$. Let $P$ denote the transition matrix and $P^t$ denote the transition matrix after $t$ steps. Let $\pi$ denote the stationary distribution. Recall that we have shown that $\mathbf{RW}_{m,\xi,i}$ has constant relaxation time and therefore constant absolute spectral gap $\lambda_*$. Given either initial distribution $\gamma(x)$, our goal is to bound the quantity

$$\sum_y \left| \left( \sum_x \gamma(x) P^t(x,y) \right) - \pi(y) \right|.$$

We begin with the following inequality that follows from the proof of Theorem 12.5 of Levin et al. [2006]. For any two states $x, y$,

$$\left| \frac{P^t(x,y)}{\pi(y)} - 1 \right| \leq \frac{\lambda_*^t}{\sqrt{\pi(x)\pi(y)}}.$$

Multiplying both sides by $\pi(y)\gamma(x)$ we obtain the inequality

$$\left| \gamma(x) P^t(x,y) - \gamma(x)\pi(y) \right| \leq \frac{\lambda_*^t \gamma(x) \sqrt{\pi(y)}}{\sqrt{\pi(x)}}.$$

The next claim bounds the ratio between $\gamma(x)$ and $\pi(x)$.

**Claim E.9.** *For any $x \geq 0$,*
$$\frac{\gamma(x)}{\pi(x)} \leq 2.$$

*Proof.* Let $Z_0 \sim \mathrm{Poi}\left((1-\xi)m/n\right)$ and $Z_1 \sim \mathrm{Poi}\left((1+\xi)m/n\right)$. Let $\lambda_0, \lambda_1$ denote the means of $Z_0, Z_1$ respectively. First, we show that $\gamma(x)/\pi(x)$ is bounded when $\gamma \sim Z_0$.

$$\frac{\Pr(Z_0 = x)}{(\Pr(Z_0 = x) + \Pr(Z_1 = x))/2} = \frac{2\Pr(Z_0 = x)}{\Pr(Z_0 = x) + \Pr(Z_1 = x)}$$
$$\leq \frac{2\Pr(Z_0 = x)}{\Pr(Z_0 = x)} = 2.$$

A similar argument holds for $\gamma \sim Z_1$. This concludes the proof of Claim E.9. $\quad\square$

Continuing from our previous calculation, we obtain

$$\left| \gamma(x) P^t(x,y) - \gamma(x)\pi(y) \right| \leq \lambda_*^t \sqrt{2\gamma(x)\pi(y)}.$$

Summing over $x$, applying the triangle inequality and noting that $\sum_x \gamma(x) = 1$, we now have

$$\left| \left( \sum_x \gamma(x) P^t(x,y) \right) - \pi(y) \right| \leq \lambda_*^t \sqrt{2\pi(y)} \sum_x \sqrt{\gamma(x)}.$$

For the remainder of the proof, we will consider the sub-linear and super-linear cases separately.

**Sub-linear Case:** $m \leq n$. Since $m \leq n$, in both cases the Poisson distribution has parameter $\lambda \leq (1 + \xi) \leq 1.1$. By standard Poisson concentration, for both $i \in \{0, 1\}$

$$\Pr(Z_i > \lambda + t) < e^{-t^2/2(\lambda+t)} = e^{-\Omega(t)}.$$

In particular, $\gamma(x + 2) < e^{-\Omega(x)}$ for all $x$. Therefore,

$$\sum_{x=0}^{\infty} \sqrt{\gamma(x)} \leq \sum_{x=0}^{C} \sqrt{\gamma(x)} + \sum_{x=C}^{\infty} \sqrt{\gamma(x)} = C + \sum_{x=C}^{\infty} e^{-\Omega(x/2-1)} = O(1)$$

for some large enough absolute constant $C$. Here, we observe that for $x \geq C$, $e^{-\Omega(x/2-1)} < e^{-x/C'}$ for some constant $C'$ so that the second term is an infinite geometric series with ratio $e^{-1/C'} < 1$. Similarly, we can bound $\sum_{y=0}^{\infty} \pi(y) = O(1)$. Thus, to conclude we sum over $y$ and note that

$$\sum_y \left| \left( \sum_x \gamma(x) P^t(x, y) \right) - \pi(y) \right| \leq \sqrt{2} \lambda_*^t \sum_y \sqrt{\pi(y)} \sum_x \sqrt{\gamma(x)} = O\left(\lambda_*^t\right).$$

In particular, from the initial distribution $\gamma$, the random walk $\mathbf{RW}_\xi$ mixes to $\delta$ in time $O(\log(1/\delta))$.

**Super-linear Case:** $n < m \leq o\left(\frac{n}{\varepsilon^2}\right)$. Recall that $x, y \sim \mathrm{Poi}(\lambda)$ for $\lambda \in \{(1 + \xi)m/n, (1 - \xi)m/n\}$. Using standard Poisson concentration (e.g. Lemma A.1) and noting that $\lambda > 1 - \xi$, we observe that for any $x$, we have $\gamma(x) < e^{-\Omega(|x-\lambda|)}$. As in the sub-linear case, we begin by bounding $\sum \sqrt{\gamma(x)}$ for initial distribution $\gamma$. For sufficiently large constant $C$, we can bound

$$\sum_{x=0}^{\infty} \sqrt{\gamma(x)} \leq \sum_{|x-\lambda| \leq C} \sqrt{\gamma(x)} + \sum_{|x-\lambda| > C} \sqrt{\gamma(x)} = O(1).$$

As above, we observe that for large enough $C$, $\sum_{|x-\lambda| > C} \sqrt{\gamma(x)}$ can be decomposed into two geometric series with ratio strictly less than 1. Since $\pi$ is a mixture of both $\gamma$, we have that $\sum_y \sqrt{\pi(y)} = O(1)$ as well. The conclusion then follows as in the sub-linear case. This concludes the proof of Lemma E.8. $\qquad \square$

*Proof of Lemma 3.8.* The proof follows immediately from Lemma E.8 and Lemma A.6. $\qquad \square$

We are now ready to show that the acceptance probability of the algorithm on sample drawn from $\mathrm{PoiS}(m, \mathbf{p})$ for a random $\mathbf{p} \sim \mathcal{M}_\xi$ is well concentrated assuming that the algorithm is sufficiently replicable in terms of the mixing time of the random walk.

**Lemma E.10.** *Let $K = \tau(0.01)$ and $\xi \in [0, \varepsilon]$. Suppose $\mathcal{A}$ is $\frac{1}{10K}$-replicable with respect to $\mathcal{H}_U$. Then,*

$$\Pr_{\mathbf{p} \sim \mathcal{M}_\xi} \left( \left| \mathbb{E}_{T \sim \mathrm{PoiS}(m,\mathbf{p})} [\mathcal{A}(T)] - \mathbb{E}_{\mathbf{p}' \sim \mathcal{M}_\xi, T' \sim \mathrm{PoiS}(m,\mathbf{p}')} [\mathcal{A}(T')] \right| > \frac{1}{4} \right) \leq \frac{1}{2}.$$

*Proof.* Consider the following sampling process:

1. Sample $\mathbf{p} \sim \mathcal{M}_\xi$.
2. Sample $T_0 \sim \mathrm{PoiS}(m, \mathbf{p})$.
3. For $1 \leq k \leq K := \tau(0.01) = O(\log n)$, sample $T_k \sim \mathbf{RW}_{m,\xi}(T_{k-1})$.

From Lemma 3.9 we know that on average over $\mathbf{p} \sim \mathcal{M}_\xi$, $\mathcal{A}(T_0) = \mathcal{A}(T_1) = \ldots = \mathcal{A}(T_K)$ with probability at least 0.9. By Markov's inequality, for $\frac{1}{2}$-fraction of $S$, we have that $\mathcal{A}(T_0) = \mathcal{A}(T_1) = \ldots = \mathcal{A}(T_k)$ with probability at least 0.8.

We now argue that the distribution of $T_k$ is 0.01-close to the distribution of $T$ drawn from the stationary distribution $\pi$ in total variation distance. Since $\mathbf{RW}_{m,\xi}$ is a product random walk, Lemma A.6 implies that it suffices to argue that each coordinate random walk $\mathbf{RW}_{m,\xi,i}$ mixes to within $\frac{0.01}{n}$ of the stationary distribution on that coordinate in $\tau_i(0.01/n)$ steps. Fix a coordinate $i$. Note that $T_0[i] \sim \mathrm{Poi}((1 + \xi)m/n)$ or $\mathrm{Poi}((1 - \xi)m/n)$. In either case, the initial distribution satisfies the

assumptions of Lemma E.8, so that after $O(\log n)$ steps, the random walk $\mathbf{RW}_{m,\xi,i}$ mixes to within $\frac{0.01}{n}$ of the stationary distribution. Given that $T_k$ is close to the stationary distribution, the data processing inequality says that the probability of acceptance under either distribution cannot differ by more than 0.01.

Here, recall that $T' \sim \pi$ is given by $T' \sim \mathbf{p}$ where $\mathbf{p} \sim \mathcal{M}_\xi$. In particular, since the stationary distribution is exactly the probability of a sample drawn from random $\mathbf{p} \sim \mathcal{M}_\xi$, we have that for $\frac{1}{2}$-fraction of $S$,

$$
\begin{aligned}
\left| \mathbb{E}_{\substack{\mathbf{p} \sim \mathcal{M}_\xi, \\ T_0 \sim \mathbf{p}}} (\mathcal{A}(T_0)) - \mathbb{E}_{T' \sim \pi}(\mathcal{A}(T')) \right| &\leq \left| \mathbb{E}_{\substack{\mathbf{p} \sim \mathcal{M}_\xi, \\ T_0 \sim \mathbf{p}}} (\mathcal{A}(T_0)) - \mathbb{E}_{T_K \sim \mathbf{RW}^K_{m,\xi}(T_0)}(\mathcal{A}(T_K)) \right| \\
&\quad + \left| \mathbb{E}_{T_K \sim \mathbf{RW}^K_{m,\xi}(T_0)}(\mathcal{A}(T_K)) - \mathbb{E}_{T' \sim \pi}(\mathcal{A}(T')) \right| \\
&\leq 0.2 + 0.01 \\
&\leq \frac{1}{4}.
\end{aligned}
$$

$\square$

We are now ready to prove the main lemma of this subsection.

*Proof of Lemma 3.6.* From Lemma 3.8 we note that the mixing time $K = \tau(0.01) = O(\log n)$. Then, since we assume that $\mathcal{A}$ is $(\log n)^{-2}$-replicable with respect to $H_U$, we have $\mathcal{A}$ is $\frac{1}{10K}$-replicable with respect to $H_U$. Applying Lemma E.10, we obtain the desired result noting that $\mathrm{Acc}_m(\mathbf{p}, \mathcal{A}) = \mathbb{E}_{T \sim \mathrm{PoiS}(m,\mathbf{p})}[\mathcal{A}(T)]$. $\square$

### E.3 Random Walk Indistinguishability

In this subsection, we show that a moderately replicable tester in general cannot distinguish a sample from the outcome after one random walk step.

*Proof of Lemma 3.9.* We first show that

$$
\sum_{i=1}^{k} \mathbb{E}_{\mathbf{p} \sim \mathcal{M}_\xi} \left[ \Pr_{T \sim \mathbf{RW}^{i-1}_{m,\xi}(\mathbf{p}), \, T' \sim \mathbf{RW}_{m,\xi}(T)} [\mathcal{A}(T) \neq \mathcal{A}(T')] \right] < k\kappa, \tag{17}
$$

After that, the lemma will follow from Markov's inequality.

Note that if we sample $\mathbf{p} \sim \mathcal{M}_\xi$, and then $T \sim \mathrm{PoiS}(m, \mathbf{p})$, we obtain exactly the stationary distribution of the random walk. Thus, the distribution of $T \sim \mathbf{RW}^{i-1}_{m,\xi}(\mathbf{p})$, $T' \sim \mathbf{RW}^i_{m,\xi}(\mathbf{p})$ is equivalent as $\mathbf{p} \sim \mathcal{M}_\xi, T \sim \mathrm{PoiS}(m, \mathbf{p})$, $T' \sim \mathbf{RW}_{m,\xi}(T)$. If we focus on just the joint distribution of $T, T'$, by the definition of $\mathbf{RW}_{m,\xi}$, this is the same as $T, T' \sim \mathrm{PoiS}(m, \mathbf{p})$, where $\mathbf{p} \sim \mathcal{M}_\xi$. This therefore gives rise to the identity $\mathbb{E}_{\mathbf{p} \sim \mathcal{M}_\xi} \left[ \Pr_{T \sim \mathbf{RW}^{i-1}_{m,\xi}(\mathbf{p}), \, T' \sim \mathbf{RW}_{m,\xi}(T)} [\mathcal{A}(T) \neq \mathcal{A}(T')] \right] = \Pr_{\mathbf{p} \sim \mathcal{M}_\xi, T, T' \sim \mathrm{PoiS}(m,\mathbf{p})} [\mathcal{A}(T) \neq \mathcal{A}(T')] = \kappa$. Summing over all $i$ then concludes the proof of Equation (17) as well as Lemma 3.9. $\square$

### E.4 Lower Bound for Poissonized Tester and Proof of Theorem 1.3

*Proof of Proposition 3.4.* By Lemma 3.5, with probability at least $\Omega(\rho)$ over the choice of $\xi$, we have that

$$
\mathbb{E}_{\mathbf{p} \sim \mathcal{M}_\xi} [\mathrm{Acc}_m(\mathbf{p}, \mathcal{A})] \in (1/3, 2/3). \tag{18}
$$

Conditioned on some $\xi$ satisfying the above, we claim that we must have

$$
\mathbb{E}_{\mathbf{p} \sim \mathcal{M}_\xi} \left[ \Pr_{T, T' \sim \mathrm{PoiS}(m,\mathbf{p})} [\mathcal{A}(T) \neq \mathcal{A}(T')] \right] \geq \log^{-2} n. \tag{19}
$$

We will proceed by a proof of contradiction. Assume that the opposite of Equation (19) holds. In that case, Lemma 3.6 becomes applicable, which gives that

$$\Pr_{\mathbf{p} \sim \mathcal{M}_\xi} \left[ \left| \mathrm{Acc}_m(\mathbf{p}, \mathcal{A}) - \mathbb{E}_{\mathbf{p} \sim \mathcal{M}_\xi} \left[ \mathrm{Acc}_m(\mathbf{p}, \mathcal{A}) \right] \right| > 1/4 \right] \leq 1/2. \tag{20}$$

Combining Equations (18) and (20) then gives that $\mathrm{Acc}_m(\mathbf{p}, \mathcal{A}) \in (1/3 - 1/4, 2/3 + 1/4)$ with probability at least $1/2$ when we choose $\mathbf{p} \sim \mathcal{M}_\xi$. Conditioned on such a $\mathbf{p}$, we immediately have that $\Pr_{T,T' \mathrm{PoiS}(m,\mathbf{p})}[\mathcal{A}(T) \neq \mathcal{A}(T')] \geq \Omega(1)$. This therefore implies that $\mathbb{E}_{\mathbf{p} \sim \mathcal{M}_\xi} \left[ \Pr_{T,T' \sim \mathrm{PoiS}(m,\mathbf{p})}[\mathcal{A}(T) \neq \mathcal{A}(T')] \right] \geq \Omega(1)$, which contradicts the assumption $\mathbb{E}_{\mathbf{p} \sim \mathcal{M}_\xi} \left[ \Pr_{T,T' \sim \mathrm{PoiS}(m,\mathbf{p})}[\mathcal{A}(T) \neq \mathcal{A}(T')] \right] \leq \log^{-2}(n)$. This concludes the proof of Equation (19).

Recall that the meta-distribution $\mathcal{H}_U$ is precisely the distribution of $\mathbf{p}$ if one first chooses $\xi$ from $[0, \varepsilon]$ uniformly at random, and then chooses $\mathbf{p} \sim \mathcal{M}_\xi$. Thus, combining Equations (18) and (19) gives that

$$\Pr_{\mathbf{p} \sim \mathcal{H}_U} \left[ \Pr_{T,T' \sim \mathrm{PoiS}(m,\mathbf{p})} [\mathcal{A}(T) \neq \mathcal{A}(T')] \right] \geq \Omega(\rho \log^{-2} n).$$

Moreover, $\|\mathbf{p}\|_1 \in (1 - \varepsilon, 1 + \varepsilon) \subseteq (0.5, 2)$ with high probability. This therefore concludes the proof of Proposition 3.4. $\qquad \square$

Our lower bound for replicable uniformity test easily follows from Proposition 3.4, and Lemma D.3.

*Proof of Theorem 1.3.* Assume without loss of generality that $m \geq \log(10/\delta) + \log(10 \log^2 n/\rho)$. From Proposition 3.4, any deterministic Poissonized tester with sample complexity $m = \tilde{o}(\sqrt{n}\varepsilon^{-2}\rho^{-1})$ that is 0.1-correct with respect to $\mathcal{M}_0$ and $\mathcal{M}_\varepsilon$ cannot be $\rho \log^{-2} n$-replicable with respect to $\mathcal{H}_U$. Furthermore, any $\mathbf{p} \sim \mathcal{H}_U$ satisfies $\|\mathbf{p}\|_1 \in (0.5, 2)$ with high probability. Thus, even conditioned on $\mathbf{p} \sim \mathcal{H}_U$ satisfying the norm condition, the deterministic Poissonized tester cannot be both 0.1-correct and $\ll \rho \log^{-2} n$-replicable. The conclusion therefore follows from Lemma D.3 (i.e. any randomized tester that is 0.01-correct cannot be $\rho/\mathrm{polylog}(n)$-replicable). $\qquad \square$

# F  Omitted Proofs for Replicable Closeness Testing Lower Bounds

In this section, we give a sample complexity lower bound of $\tilde{\Omega}(n^{2/3}\varepsilon^{-4/3}\rho^{-2/3} + \sqrt{n}\varepsilon^{-2}\rho^{-1} + \varepsilon^{-2}\rho^{-2})$ for replicable closeness testing.

Note that closeness testing is at least as hard as uniformity testing (even when replicability is of concern). Hence, it remains for us to show a lower bound of $\tilde{\Omega}\left(n^{2/3}\varepsilon^{-4/3}\rho^{-2/3}\right)$. Note that this term dominates exactly in the sub-linear regime so it suffices to prove a lower bound in the regime $m \ll n$.

We start by describing the hard instance for replicable closeness testing in this regime. In particular, we construct meta-distributions over pairs of non-negative measures that will be used as inputs to the closeness testing problem.

**Definition F.1** (Closeness Test Hard Instance). *For $\xi \in [0, \epsilon]$, we define $\mathcal{N}_\xi$ to be the distribution over pairs of non-negative measures $\mathbf{p}_\xi, \mathbf{q}_\xi$ generated as follows: $\forall i \in [n]$*

$$(\mathbf{p}_\xi(i), \mathbf{q}_\xi(i)) = \begin{cases} \left( \frac{1-\epsilon}{m}, \frac{1-\epsilon}{m} \right) & w.p. \quad \frac{m}{n} \\ \left( \frac{2\epsilon+\xi}{2(n-m)}, \frac{2\epsilon-\xi}{2(n-m)} \right) & w.p. \quad \frac{n-m}{2n} \\ \left( \frac{2\epsilon-\xi}{2(n-m)}, \frac{2\epsilon+\xi}{2(n-m)} \right) & w.p. \quad \frac{n-m}{2n}. \end{cases} \tag{21}$$

*The meta-distribution $\mathcal{H}_\mathcal{C}$ is the distribution over random pairs of non-negative measures $(\mathbf{p}, \mathbf{q})$ generated as follows: choose $\xi$ uniformly at random from $[0, \epsilon]$, and return $(\mathbf{p}, \mathbf{q}) \sim \mathcal{N}_\xi$.*

Again, thanks to Lemma D.3, after fixing the hard instance to be $\mathcal{H}_\mathcal{C}$, it then suffices for us to show sample complexity lower bounds against deterministic closeness tester $\mathcal{A}$ within the Poisson sampling model.

**Proposition F.2.** *Let $\mathcal{H}_C$ be the meta-distribution defined as in Definition F.1, $\mathcal{A}$ be a deterministic tester that takes as input a sample count vectors $T \in \mathbb{N}^{2n}$,[5] and $m = \tilde{o}(n^{2/3}\varepsilon^{-4/3}\rho^{-2/3} + n)$. Then it holds*

$$\Pr_{(\mathbf{p},\mathbf{q})\sim\mathcal{H}_C}\left[\Pr_{T,T'\sim PoiS(m,\mathbf{p}\oplus\mathbf{q})}[\mathcal{A}(T) \neq \mathcal{A}(T')] \geq \log^{-2} n \text{ and } \|(\mathbf{p}\oplus\mathbf{q})/2\|_1 \in (0.5,2)\right] \geq \rho.$$

In the rest of the section, we focus on showing Proposition F.2.

Define $\mathrm{Acc}_m(\mathbf{p},\mathbf{q},\mathcal{A}) := \Pr_{T\sim \mathrm{PoiS}(m,\mathbf{p}\oplus\mathbf{q})}[\mathcal{A}(T) = \text{Accept}]$. Similar to the argument for replicable uniformity testing lower bound, we begin by showing the intermediate result that the average acceptance probability $\mathbb{E}_{(\mathbf{p},\mathbf{q})\sim\mathcal{N}_\xi}[\mathrm{Acc}_m(\mathbf{p},\mathbf{q},\mathcal{A})]$ is close to $1/2$ with probability at least $\rho$ if $\xi$ is chosen randomly from $[0,\varepsilon]$.

### F.1 Bounding the Average Acceptance Probability for Closeness Testing

We dedicate this section to show the following lemma.

**Lemma F.3.** *Let $\mathcal{A}$ be a deterministic Poissonized tester that's $0.1$-correct w.r.t. $\mathcal{N}_\xi$, then $\Pr_{\xi\sim\mathcal{U}([0,\varepsilon])}\left[\mathbb{E}_{(\mathbf{p},\mathbf{q})\sim\mathcal{N}_\xi}[Acc_m(\mathbf{p},\mathbf{q},\mathcal{A})] \in (1/3,2/3)\right] \geq \rho$ as long as $m = o(n^{2/3}\varepsilon^{-4/3}\rho^{-2/3} + n)$.*

The argument again uses information theory and is similar to the proof of Lemma 3.5. We follow the road map similar to the one to show Lemma 3.5 in Appendix E.1, where the main difference is that we work with a different hard instance. We therefore present the needed lemmas without restating the outline.

**Lemma F.4** (Mutual Information Bound for Closeness Testing Hard Instance)**.** *Let $m < n/2$, $\epsilon_0 < \epsilon_1 \in [0,\epsilon]$ be such that $\epsilon_1 - \epsilon_0 < \epsilon\rho$, $X$ be an unbiased random bit, $(\tilde{\mathbf{p}},\tilde{\mathbf{q}}) \sim \mathcal{N}_{\epsilon_X}$ be defined as in Definition F.1, $T \sim PoiS(m,\tilde{\mathbf{p}}\oplus\tilde{\mathbf{q}})$ as in Proposition F.2 where $\left(T_1^{\tilde{\mathbf{p}}}, T_2^{\tilde{\mathbf{p}}}, \cdots, T_n^{\tilde{\mathbf{p}}}, T_1^{\tilde{\mathbf{q}}}, T_2^{\tilde{\mathbf{q}}}, \cdots, T_n^{\tilde{\mathbf{q}}}\right) = T \in \mathbb{R}^{2n}$ where $T_i^{\tilde{\mathbf{p}}}$ counts the occurrences of element $i$ sampled from $\tilde{\mathbf{p}}/\|\tilde{\mathbf{p}}\|_1$, $T_i^{\tilde{\mathbf{q}}}$ counts the occurrences of element $i$ sampled from $\tilde{\mathbf{q}}/\|\tilde{\mathbf{q}}\|_1$. Then*

$$I\left(X : T_1^{\tilde{\mathbf{p}}}, \cdots, T_n^{\tilde{\mathbf{p}}}, T_1^{\tilde{\mathbf{q}}}, \cdots, T_n^{\tilde{\mathbf{q}}}\right) = O\left(\frac{m^3}{n^2}\epsilon^4\rho^2\right).$$

*Proof of Lemma F.4.* Denote $\delta := \epsilon_1 - \epsilon_0 = O(\epsilon\rho)$. Since $(T_i, N_i)'s$ are conditionally independent on $X$, we have that

$$I\left(X : T_1^{\tilde{\mathbf{p}}}, \cdots, T_n^{\tilde{\mathbf{p}}}, T_1^{\tilde{\mathbf{q}}}, \cdots, T_n^{\tilde{\mathbf{q}}}\right) \leq \sum_{i=1}^n I\left(X : T_i^{\tilde{\mathbf{p}}}, T_i^{\tilde{\mathbf{q}}}\right) =: nI\left(X : T_1^{\tilde{\mathbf{p}}}, T_1^{\tilde{\mathbf{q}}}\right).$$

Therefore, it suffices to show that $I\left(X : T_1^{\tilde{\mathbf{p}}}, T_1^{\tilde{\mathbf{q}}}\right) = O\left(\frac{m^3}{n^3}\epsilon^2\delta^2\right)$. We first note that $\Pr\left[T_1^{\tilde{\mathbf{p}}} = a, T_1^{\tilde{\mathbf{q}}} = b | X = 0\right] = \Theta(1)\Pr\left[T_1^{\tilde{\mathbf{p}}} = a, T_1^{\tilde{\mathbf{q}}} = b | X = 1\right]$ since $\delta = o(\epsilon)$. Therefore, by Claim A.2 it suffices to show that $\bar{I} := \sum_a \frac{\left(\Pr\left[T_1^{\tilde{\mathbf{p}}}=a,T_1^{\tilde{\mathbf{q}}}=b|X=0\right]-\Pr\left[T_1^{\tilde{\mathbf{p}}}=a,T_1^{\tilde{\mathbf{q}}}=b|X=1\right]\right)^2}{\Pr\left[T_1^{\tilde{\mathbf{p}}}=a,T_1^{\tilde{\mathbf{q}}}=b|X=0\right]+\Pr\left[T_1^{\tilde{\mathbf{p}}}=a,T_1^{\tilde{\mathbf{q}}}=b|X=1\right]} =$

---

[5]Note that a closeness tester $\mathcal{A}$ should in principle receive two sets of samples (or two sample count vectors in the Poisson sampling model) — one from $\mathbf{p}$ and the other from $\mathbf{q}$. However, it is not hard to see that do not lose any information if we simply concatenate the two sample count vectors together. For notational convenience, we denote by $(\mathbf{p} \oplus \mathbf{q})$ the non-negative measures over $[2n]$, where the first $n$ entries agree with $\mathbf{p}$ and the last $n$ entries agree with $\mathbf{q}$. Then the distribution over the concatenated sample count vectors is simply given by $\mathrm{Poi}(m,\mathbf{p}\oplus\mathbf{q})$.

$O\left(\frac{m^3}{n^3}\epsilon^2\delta^2\right)$. We next expand $\Pr\left[T_1^{\tilde{\mathbf{p}}} = a, T_1^{\tilde{\mathbf{q}}} = b|X = 0\right]$.

$$
\begin{aligned}
\Pr&\left[T_1^{\tilde{\mathbf{p}}} = a, T_1^{\tilde{\mathbf{q}}} = b|X = 0\right]\\
&= \frac{m}{n}\frac{1}{a!}(1-\epsilon)^a \exp(-(1-\epsilon))\frac{1}{b!}(1-\epsilon)^b \exp(-(1-\epsilon))\\
&\quad + \frac{n-m}{2n}\frac{1}{a!}\left(\frac{m(2\epsilon+\epsilon_0)}{2(n-m)}\right)^a \exp\left(-\frac{m(2\epsilon+\epsilon_0)}{2(n-m)}\right)\frac{1}{b!}\left(\frac{m(2\epsilon-\epsilon_0)}{2(n-m)}\right)^b \exp\left(-\frac{m(2\epsilon-\epsilon_0)}{2(n-m)}\right)\\
&\quad + \frac{n-m}{2n}\frac{1}{b!}\left(\frac{m(2\epsilon+\epsilon_0)}{2(n-m)}\right)^b \exp\left(-\frac{m(2\epsilon+\epsilon_0)}{2(n-m)}\right)\frac{1}{a!}\left(\frac{m(2\epsilon-\epsilon_0)}{2(n-m)}\right)^a \exp\left(-\frac{m(2\epsilon-\epsilon_0)}{2(n-m)}\right)\\
&= \frac{1}{a!b!}\left(\frac{m}{n}(1-\epsilon)^{a+b}\exp(-2(1-\epsilon)) + \frac{n-m}{2n}\left(\frac{m(2\epsilon+\epsilon_0)}{2(n-m)}\right)^a\left(\frac{m(2\epsilon-\epsilon_0)}{2(n-m)}\right)^b\exp\left(\frac{2m\epsilon}{n-m}\right)\right.\\
&\quad \left.+ \frac{n-m}{2n}\left(\frac{m(2\epsilon+\epsilon_0)}{2(n-m)}\right)^b\left(\frac{m(2\epsilon-\epsilon_0)}{2(n-m)}\right)^a\exp\left(\frac{2m\epsilon}{n-m}\right)\right) =: f_{a,b}(\epsilon_0),
\end{aligned}
$$

Then it follows that

$$
\bar{I} = O(1)\sum_{a,b\in\mathbb{Z}_{\geq 0}}\frac{1}{a!b!}\frac{(f_{a,b}(\epsilon_0) - f_{a,b}(\epsilon_1))^2}{f_{a,b}(\epsilon_0) + f_{a,b}(\epsilon_1)} =: O(1)\sum_{a,b\in\mathbb{Z}_{\geq 0}}\bar{I}_{a,b}
$$

where we denote $\bar{I}_{a,b} := \frac{1}{a!b!}\frac{(f_{a,b}(\epsilon_0) - f_{a,b}(\epsilon_1))^2}{f_{a,b}(\epsilon_0) + f_{a,b}(\epsilon_1)}$. To bound $\bar{I}_{a,b}$, we break into 3 cases regarding the value of $a$.

when $a + b = 0$, viz. $a = b = 0$, $\Pr\left[T_1^{\tilde{\mathbf{p}}} = 0, T_1^{\tilde{\mathbf{q}}} = 0|X = 0\right] = \Pr\left[T_1^{\tilde{\mathbf{p}}} = 0, T_1^{\tilde{\mathbf{q}}} = 0|X = 1\right] = \frac{m}{n}\exp(-2(1-\epsilon)) + \frac{n-m}{n}\exp\left(\frac{2m\epsilon}{n-m}\right)$. Therefore, $\bar{I}_{0,0} = 0$.

when $a + b = 1$, wlog $a = 0, b = 1$ (by the symmetry between $a$ and $b$.) We have that

$$
\begin{aligned}
\Pr&\left[T_1^{\tilde{\mathbf{p}}} = 0, T_1^{\tilde{\mathbf{q}}} = 1|X = 0\right] = \Pr\left[T_1^{\tilde{\mathbf{p}}} = 0, T_1^{\tilde{\mathbf{q}}} = 1|X = 1\right] =\\
&= \frac{m}{n}(1-\epsilon)\exp(-2(1-\epsilon)) + \frac{n-m}{2n}\left(\frac{m(2\epsilon-\epsilon_0)}{2(n-m)}\right)\exp\left(\frac{2m\epsilon}{n-m}\right)\\
&\quad + \frac{n-m}{2n}\left(\frac{m(2\epsilon+\epsilon_0)}{2(n-m)}\right)\exp\left(\frac{2m\epsilon}{n-m}\right)\\
&= \frac{m}{n}(1-\epsilon)\exp(-2(1-\epsilon)) + \frac{\epsilon m}{n}\exp\left(-\frac{2m\epsilon}{n-m}\right).
\end{aligned}
$$

Thus, $\bar{I}_{0,1} = \bar{I}_{1,0} = 0$.

when $a + b > 1$, wlog $a \leq b$, then the denominator term $f_{a,b}(\epsilon_0) + f_{a,b}(\epsilon_1) = \Omega\left(\frac{m}{n}(1 - \epsilon)^{a+b}\right)$. On the other hand consider the numerator. When $y \in [\epsilon_0, \epsilon_1]$,

$$\left|\frac{\partial}{\partial y} f_{a,b}(y)\right| = \frac{n - m}{2n} \exp\left(-\frac{2m\epsilon}{n - m}\right) \cdot$$

$$\left|\frac{m}{2(n - m)}\left(a\left(\frac{m(2\epsilon + y)}{2(n - m)}\right)^{a-1}\left(\frac{m(2\epsilon - y)}{2(n - m)}\right)^{b} - b\left(\frac{m(2\epsilon + y)}{2(n - m)}\right)^{a}\left(\frac{m(2\epsilon - y)}{2(n - m)}\right)^{b-1}\right)\right.$$

$$\left.+\frac{m}{2(n - m)}\left(b\left(\frac{m(2\epsilon + y)}{2(n - m)}\right)^{b-1}\left(\frac{m(2\epsilon - y)}{2(n - m)}\right)^{a} - a\left(\frac{m(2\epsilon + y)}{2(n - m)}\right)^{b}\left(\frac{m(2\epsilon - y)}{2(n - m)}\right)^{a-1}\right)\right|$$

$$= \frac{m}{n}\frac{m}{2(n - m)}^{a+b-1}\left|(2\epsilon + y)^{a-1}(2\epsilon - y)^{b-1}[2\epsilon(a - b) - y(a + b)]\right.$$

$$\left.+ (2\epsilon + y)^{b-1}(2\epsilon - y)^{a-1}[2\epsilon(b - a) - y(a + b)]\right|.$$

Let $c > 1$ be such that $n - m = \frac{1}{c}n$, then

$$= O(c^{a+b})\left(\frac{m}{n}\right)^{a+b}\left[2\epsilon(b - a)\left|(2\epsilon + y)^{b-1}(2\epsilon - y)^{a-1} - (2\epsilon + y)^{a-1}(2\epsilon - y)^{b-1}\right|\right.$$

$$\left.+y(a + b)\left|(2\epsilon + y)^{b-1}(2\epsilon - y)^{a-1} + (2\epsilon + y)^{a-1}(2\epsilon - y)^{b-1}\right|\right]$$

$$\leq O(c^{a+b})\left(\frac{m}{n}\right)^{a+b}\epsilon(a + b)\left[(2\epsilon + y)^{b-1}(2\epsilon - y)^{a-1} + (2\epsilon + y)^{a-1}(2\epsilon - y)^{b-1}\right]$$

$$= O\left((4c)^{a+b}\left(\frac{m}{n}\right)^{a+b}\epsilon^{a+b-1}(a + b)\right).$$

This implies that $\forall a + b \geq 2, \bar{I}_{a,b} = O\left(\frac{1}{a!b!}\left(\frac{(4c)^2}{1 - \epsilon}\right)^{a+b}\left(\frac{m}{n}\right)^{2a+2b-1}\epsilon^{2(a+b-1)}(a + b)^2\delta^2\right)$ then

$$\sum_{\substack{a,b\in\mathbb{Z}_{\geq 0} \\ a+b\geq 2 \\ a\leq b}} \bar{I}_{a,b} = O\left(\sum_{a=0}^{\infty}\frac{1}{a!}\left(\frac{(4c)^2}{1 - \epsilon}\right)^{a}\sum_{\substack{b\geq a \\ b\geq 2-a}}\frac{1}{b!}\left(\frac{(4c)^2}{1 - \epsilon}\right)^{b}\left(\frac{m}{n}\right)^{2a+2b-1}\epsilon^{2(a+b-1)}(a + b)^2\delta^2\right),$$

where $\sum_{\substack{b\geq a \\ b\geq 2-a}}\frac{1}{b!}\left(\frac{(4c)^2}{1 - \epsilon}\right)^{b} \leq \exp((4c)^2/(1 - \epsilon))$ and the sum of infinite geometric series is dominated by the first term when common ratio $< 1$, we have that

$$\sum_{\substack{a,b\in\mathbb{Z}_{\geq 0} \\ a+b\geq 2 \\ a\leq b}} \bar{I}_{a,b} = O\left(\sum_{a=0}^{\infty}\frac{1}{a!}\left(\frac{(4c)^2}{1 - \epsilon}\right)^{a}\left(\frac{m}{n}\right)^{3}\epsilon^2\delta^2\right) = O\left(\left(\frac{m}{n}\right)^{3}\epsilon^2\delta^2\right) = \sum_{\substack{a,b\in\mathbb{Z}_{\geq 0} \\ a+b\geq 2 \\ a\geq b}} \bar{I}_{a,b}$$

as desired.

In conclusion, from the above 3 cases we have that

$$\bar{I} \leq \bar{I}_{0,0} + \bar{I}_{0,1} + \bar{I}_{1,0} + \sum_{\substack{a,b\in\mathbb{Z}_{\geq 0} \\ a+b\geq 2 \\ a\geq b}} \bar{I}_{a,b} + \sum_{\substack{a,b\in\mathbb{Z}_{\geq 0} \\ a+b\geq 2 \\ a\leq b}} \bar{I}_{a,b} = O\left(\left(\frac{m}{n}\right)^{3}\epsilon^2\delta^2\right),$$

which concludes the proof of Lemma F.4. $\qquad\square$

**Lemma F.5** (Lipchitzness of Expectation of Acceptance Probability Function). *Assume that $m = o(n^{2/3}\varepsilon^{-4/3}\rho^{-2/3} + n)$, and $\mathcal{A}$ be a deterministic tester takes a sample-count vector over $[n]$ as input and returns 1 if accept 0 otherwise, and recall the acceptance probability function is defined via $Acc_m(\mathbf{p}, \mathbf{q}, \mathcal{A}) := \Pr_{T\sim PoiS(m,\mathbf{p}\oplus\mathbf{q})}[\mathcal{A}(T) = Accept]$. Let $\epsilon_0 < \epsilon_1 \in [0, \epsilon]$ be such that $\epsilon_1 - \epsilon_0 \leq \epsilon\rho$. Then it holds that*

$$\left|\mathbb{E}_{(\mathbf{p},\mathbf{q})\sim\mathcal{N}_{\epsilon_0}}[Acc_m(\mathbf{p}, \mathbf{q}, \mathcal{A})] - \mathbb{E}_{(\mathbf{p},\mathbf{q}))\sim\mathcal{N}_{\epsilon_1}}[Acc_m(\mathbf{p}, \mathbf{q}, \mathcal{A})]\right| < 0.1$$

*Proof of Lemma F.5.* Assume the opposite $|\mathbb{E}_{(\mathbf{p},\mathbf{q})\sim\mathcal{N}_{\epsilon_0}}[\mathrm{Acc}_m(\mathbf{p},\mathbf{q},\mathcal{A})] - \mathbb{E}_{(\mathbf{p},\mathbf{q}))\sim\mathcal{N}_{\epsilon_1}}[\mathrm{Acc}_m(\mathbf{p},\mathbf{q},\mathcal{A})]| \geq 0.1$. Then, let $X$ be an unbiased random bit, and $Y$ be the random variable defined as follows: let $(\mathbf{p},\mathbf{q}) \sim \mathcal{N}_{\epsilon_X}$, $T \sim \mathrm{PoiS}(m,\mathbf{p}\oplus\mathbf{q})$, then $Y = 1$ if $\mathcal{A}(T)$ accept, $Y = 0$ otherwise. From the definition, we notice that $\Pr[Y = 1|X = 0] = \mathbb{E}_{(\mathbf{p},\mathbf{q})\sim\mathcal{N}_{\epsilon_0}}[\mathrm{Acc}_m(\mathbf{p},\mathbf{q},\mathcal{A})]$ and $\Pr[Y = 1|X = 1] = \mathbb{E}_{(\mathbf{p},\mathbf{q})\sim\mathcal{N}_{\epsilon_1}}[\mathrm{Acc}_m(\mathbf{p},\mathbf{q},\mathcal{A})]$, which implies a mutual information bound of $I(X : T) = I(X : Y) = \Omega(1)$. This contradicts with the result from Lemma F.4. $\qquad\square$

We are now ready to prove Lemma F.3.

*Proof of Lemma F.3.* Since $\mathcal{A}$ is $0.1$-correct w.r.t. $\mathcal{N}_0$ and $\mathcal{N}_\varepsilon$, we have that $\mathbb{E}_{(\mathbf{p},\mathbf{q})\sim\mathcal{N}_0}[\mathrm{Acc}_m(\mathbf{p},\mathbf{q},\mathcal{A})] \geq 0.9$ and $\mathbb{E}_{(\mathbf{p},\mathbf{q})\sim\mathcal{N}_\epsilon}[\mathrm{Acc}_m(\mathbf{p},\mathbf{q},\mathcal{A})] < 0.1$. Furthermore, since $\mathbb{E}_{(\mathbf{p},\mathbf{q})\sim\mathcal{N}_\xi}[\mathrm{Acc}_m(\mathbf{p},\mathbf{q},\mathcal{A})]$ is a polynomial in $\xi$, it is continuous in $\xi$, then by mean value theorem there exists $\xi^* \in (0,\epsilon)$ such that $\mathbb{E}_{(\mathbf{p},\mathbf{q})\sim\mathcal{N}_{\xi^*}}[\mathrm{Acc}_m(\mathbf{p},\mathbf{q},\mathcal{A})] = 1/2$. Immediately following from Lemma F.5 that $\forall\xi \in [\xi^* - \rho\epsilon, \xi^* + \rho\epsilon]$ we have that

$$\mathbb{E}_{(\mathbf{p},\mathbf{q})\sim\mathcal{N}_\xi}[\mathrm{Acc}_m(\mathbf{p},\mathbf{q},\mathcal{A})] \in \Big(\mathbb{E}_{(\mathbf{p},\mathbf{q})\sim\mathcal{N}_{\xi^*}}[\mathrm{Acc}_m(\mathbf{p},\mathbf{q},\mathcal{A})] - 0.1, \mathbb{E}_{(\mathbf{p},\mathbf{q})\sim\mathcal{N}_{\xi^*}}[\mathrm{Acc}_m(\mathbf{p},\mathbf{q},\mathcal{A})] + 0.1\Big)$$
$$= (0.4, 0.6) \subset (1/3, 2/3).$$

In conclusion, if we uniformly at randomly select a $\xi \in [0,\epsilon]$, then once it falls in interval $[\xi^* - \rho\epsilon, \xi^* + \rho\epsilon]$ of length $2\rho\epsilon$ we have that $\mathbb{E}_{(\mathbf{p},\mathbf{q})\sim\mathcal{N}_\xi}[\mathrm{Acc}_m(\mathbf{p},\mathbf{q},\mathcal{A})] \in (1/3, 2/3)$ as desired. $\qquad\square$

Conditioned on some $\xi$ satisfying the probabilistic condition in Lemma F.3, we then proceed to show that the acceptance probability $\mathrm{Acc}_m(\mathbf{p},\mathbf{q},\mathcal{A})$ concentrates around the expected acceptance probability $\mathbb{E}_{(\mathbf{p},\mathbf{q})\sim\mathcal{N}_\xi}[\mathrm{Acc}_m(\mathbf{p},\mathbf{q},\mathcal{A})]$.

### F.2  Concentration of Acceptance Probabilities

In this section we prove that the acceptance probabilities concentrate.

**Lemma F.6** (Concentration of Acceptance Probabilities)**.** *Let $\xi \in (0,\varepsilon)$ and $\mathcal{A}$ be a deterministic tester satisfying that is $\log^{-2} n$-replicable with respect to $\mathcal{H}_C$. Then it holds*

$$\Pr_{(\mathbf{p},\mathbf{q})\sim\mathcal{N}_\xi}\left(\left|Acc_m(\mathbf{p},\mathbf{q},\mathcal{A}) - \mathbb{E}_{(\mathbf{p}',\mathbf{q}')\sim\mathcal{N}_\xi}[Acc_m(\mathbf{p}',\mathbf{q}',\mathcal{A})]\right| > \frac{1}{4}\right) \leq \frac{1}{2}.$$

This is achieved by analyzing the sample random walk $\mathbf{RW}_{m,\mathcal{N}_\xi}$ analogous to the one considered in the uniformity testing case.

We begin by defining a random walk on samples drawn from distributions in $\mathcal{N}_\xi$.

**Definition F.7.** *The Sample Random Walk $\mathbf{RW}_{m,\xi}$ is defined on the graph with vertex set $\mathbb{N}^{2n}$ (where each vertex corresponds to sample count vector $T$ drawn from $PoiS(m,\mathbf{p}\oplus\mathbf{q})$,) and transitions $(T_1, T_2)$ are defined by the conditional distribution of $T_2$ given $T_1$ induced by the joint distribution given by the following process:*

*1. $(\mathbf{p},\mathbf{q}) \sim \mathcal{N}_\xi$*
*2. $T_1, T_2$ are independently sampled from $PoiS(m,\mathbf{p}\oplus\mathbf{q})$.*

*For any sample count vector $T$, let $\mathbf{RW}_{m,\xi}(T)$ be the random variable representing the next step of the random walk from $T$.*

Given a sample count vector $T$, we denote $T_\mathbf{p}[i]$ (resp. $T_\mathbf{q}[i]$) the frequency of bucket $i$ from $\mathbf{p}$ (resp. $\mathbf{q}$). Let us analyze the random walk $\mathbf{RW}_{m,\xi}$. As before, we have $T_\mathbf{p}[i] \sim \mathrm{Poi}(m\mathbf{p}_i)$ and $T_\mathbf{q}[i] \sim \mathrm{Poi}(m\mathbf{q}_i)$ independently for all $i$. Thus, $\mathbf{RW}_{m,\xi}$ is the product of $n$ independent random walks $\mathbf{RW}_{m,\xi,i}$ on vertex set $[m] \times [m]$. We describe $\mathbf{RW}_{m,\xi}$ by describing each random walk $\mathbf{RW}_{m,\xi,i}$.

If $S, T$ are drawn from the joint distribution defining $\mathbf{RW}_{m,\xi,i}$,

$$\Pr(S_{\mathbf{p}}[i] = a, T_{\mathbf{p}}[i] = b, S_{\mathbf{q}}[i] = c, T_{\mathbf{q}}[i] = d)$$

$$= \frac{m}{n} \frac{e^{4(1-\varepsilon)} (1-\varepsilon)^{a+b+c+d}}{a!b!c!d!}$$

$$+ \frac{n-m}{2n} \left( \frac{e^{(2\varepsilon+\xi)m/(n-m)} \left( \frac{(2\varepsilon+\xi)m}{2(n-m)} \right)^{a+b}}{a!b!} \frac{e^{(2\varepsilon-\xi)m/(n-m)} \left( \frac{(2\varepsilon-\xi)m}{(n-m)} \right)^{c+d}}{c!d!} \right.$$

$$\left. + \frac{e^{(2\varepsilon-\xi)m/(n-m)} \left( \frac{(2\varepsilon-\xi)m}{(n-m)} \right)^{a+b}}{a!b!} \frac{e^{(2\varepsilon+\xi)m/(n-m)} \left( \frac{(2\varepsilon+\xi)m}{2(n-m)} \right)^{c+d}}{c!d!} \right)$$

$$= \frac{me^{4(1-\varepsilon)} (1-\varepsilon)^{a+b+c+d}}{n \cdot a!b!c!d!} + \frac{n-m}{2n} \left( \frac{m}{2(n-m)} \right)^{a+b+c+d} \frac{e^{4\varepsilon m/(n-m)}}{a!b!c!d!} f_{a,b,c,d}(\xi)$$

where

$$f_{a+b,c+d}(\xi) = (2\varepsilon+\xi)^{a+b} (2\varepsilon-\xi)^{c+d} + (2\varepsilon-\xi)^{a+b} (2\varepsilon+\xi)^{c+d}.$$

Similarly, we compute the marginal distribution as

$$\Pr(S_{\mathbf{p}}[i] = a, S_{\mathbf{q}}[i] = c) = \frac{me^{2(1-\varepsilon)} (1-\varepsilon)^{a+c}}{n \cdot a!c!} + \frac{n-m}{2n} \left( \frac{m}{2(n-m)} \right)^{a+c} \frac{e^{2\varepsilon m/(n-m)}}{a!c!} f_{a,c}(\xi).$$

To describe the random walk transition probability, we compute the conditional distribution

$$P((a,c),(b,d)) = \Pr(T_{\mathbf{p}}[i] = b, T_{\mathbf{q}}[i] = d \mid S_{\mathbf{p}}[i] = a, S_{\mathbf{q}}[i] = c)$$

$$= \frac{\Pr(T_{\mathbf{p}}[i] = b, T_{\mathbf{q}}[i] = d, S_{\mathbf{p}}[i] = a, S_{\mathbf{q}}[i] = c)}{\Pr(S_{\mathbf{p}}[i] = a, S_{\mathbf{q}}[i] = c)}$$

and note that as before, the stationary distribution is given by the probability vector $\pi(a,c) = \Pr(S_{\mathbf{p}}[i] = a, S_{\mathbf{q}}[i] = c)$.

Following identical arguments as Lemma 3.9, we show that over few steps of the random walk, the outcome of the algorithm does not change significantly.

**Lemma F.8.** *Let $\mathcal{A}$ be $1/(10K)$-replicable with respect to $\mathcal{H}_C$. Let $(\mathbf{p}, \mathbf{q}) \sim \mathcal{N}_\xi$ and $T_0 \sim PoiS(m, \mathbf{p} \oplus \mathbf{q})$. For $1 \leq k \leq K$, let $T_k \sim \mathbf{RW}_{m,\xi}(T_{k-1})$. Then,*

$$\Pr_{T_0,\ldots,T_k} \left( \bigcup_{k=1}^{K} \{\mathcal{A}(T_{k-1}) \neq \mathcal{A}(T_k)\} \right) < \frac{1}{10}.$$

We now bound the mixing time of the random walk $\mathbf{RW}_{m,\xi}$. As in the argument for uniformity, we begin by bounding the (constant) mixing time of a single coordinate.

**Lemma F.9.** *Suppose $m \leq n/10$. The random walk $\mathbf{RW}_{m,\xi,i}$ has mixing time $\tau(0.11) = O(1)$.*

*Proof.* Let $Y_0 = (\ell_1, \ell_2)$ denote the current step. Let $X \sim \{A, B, C\}$ be drawn randomly with probabilities $\frac{m}{n}$, $\frac{n-m}{2n}$, $\frac{n-m}{2n}$ respectively. Let $Z_1^A \sim \text{Poi}(1-\varepsilon)$, $Z_1^B \sim \text{Poi}\left( \frac{(2\varepsilon+\xi)m}{2(n-m)} \right)$, $Z_1^C \sim \text{Poi}\left( \frac{(2\varepsilon-\xi)m}{2(n-m)} \right)$ and $Z_2^A \sim \text{Poi}(1-\varepsilon)$, $Z_2^B \sim \text{Poi}\left( \frac{(2\varepsilon-\xi)m}{2(n-m)} \right)$, $Z_2^C \sim \text{Poi}\left( \frac{(2\varepsilon+\xi)m}{2(n-m)} \right)$. The next step of the random walk $Y_1 \sim \mathbf{RW}_{m,\xi,i}(Y_0)$ is taken according to the distribution

$$P((\ell_1, \ell_2),(k_1, k_2)) = \Pr(X = A \mid Y_0 = (\ell_1, \ell_2)) \Pr(Z_1^A = k_1, Z_2^A = k_2)$$

$$+ \Pr(X = B \mid Y_0 = (\ell_1, \ell_2)) \Pr(Z_1^B = k_1, Z_2^B = k_2)$$

$$+ \Pr(X = C \mid Y_0 = (\ell_1, \ell_2)) \Pr(Z_1^C = k_1, Z_2^C = k_2).$$

We show that regardless of the current state $(\ell_1, \ell_2)$, the random walk reaches state $(0,0)$ with reasonable probability. Since $\varepsilon < 0.1$,

$$\Pr(Z_1^A = 0, Z_2^A = 0) = \left(e^{-(1-\varepsilon)}\frac{(1-\varepsilon)^0}{0!}\right)^2 = e^{-2(1-\varepsilon)} \geq 0.13,$$

$$\Pr(Z_1^B = 0, Z_2^B = 0) = e^{-\frac{(2\varepsilon+\xi)m}{2(n-m)}} e^{-\frac{(2\varepsilon-\xi)m}{2(n-m)}} = e^{-\frac{2\varepsilon m}{n-m}} \geq e^{-2\varepsilon} > 0.8,$$

$$\Pr(Z_1^C = 0, Z_2^C = 0) > 0.8$$

where we have used $m < n/10$ in the second bound. In particular, note that $\frac{m}{n-m} \leq \frac{1}{9}$ (here we only use that $1/9 < 1$). The last follows identically. Then, regardless of $(\ell_1, \ell_2)$, we have $P((\ell_1, \ell_2), (0,0)) \geq 0.13$. In particular, after $O(1)$ steps, we can guarantee that with probability $0.99$ we reach the state $(0,0)$. We can then assume without loss of generality that $\ell_1 = \ell_2 = 0$.

We now examine the distribution of $X$ conditioned on $Y_0 = (0,0)$. First, note that

$$\Pr(Y_0 = (0,0)) = \frac{m}{n}e^{-2(1-\varepsilon)} + \frac{n-m}{2n}\left(e^{-2\varepsilon m/(n-m)} + e^{-2\varepsilon m/(n-m)}\right)$$

$$= \frac{m}{n}e^{-2(1-\varepsilon)} + \frac{n-m}{n}e^{-2\varepsilon m/(n-m)}.$$

Then, we argue that distribution of $X$ conditioned on $Y_0 = (0,0)$ is reasonably random.

$$\Pr(X = B \mid Y_0 = (0,0)) = \frac{\Pr(X = B, Y_0 = (0,0))}{\Pr(Y_0 = (0,0))}$$

$$= \frac{\frac{n-m}{2n}e^{-2\varepsilon/(n-m)}}{\frac{m}{n}e^{-2(1-\varepsilon)} + \frac{n-m}{n}e^{-2\varepsilon/(n-m)}}$$

$$= \frac{1/2}{\frac{m}{n-m}\exp\left(\frac{2\varepsilon m}{n-m} - 2(1-\varepsilon)\right) + 1}.$$

Observe $\frac{m}{10n} \leq \frac{m}{n}\exp(-2) \leq \frac{m}{n-m}\exp\left(\frac{2\varepsilon m}{n-m} - 2(1-\varepsilon)\right) \leq \frac{2m}{n}\exp\left(\frac{\varepsilon}{9} - 1.8\right) \leq \frac{m}{n}$ so that applying $0.5 - x \leq \frac{0.5}{1+x} \leq 0.5 - x/3$ for small $x > 0$,

$$0.5 - \frac{m}{n} \leq \Pr(X = B \mid Y_0 = (0,0)) \leq 0.5 - \frac{m}{30n}.$$

As a result, we can conclude

$$\frac{m}{15n} \leq \Pr(X = A \mid Y_0 = (0,0)) \leq \frac{2m}{n}.$$

In the stationary distribution, we have

$$\Pr(X = A) = \frac{m}{n} \ , \ \ \Pr(X = B) = \Pr(X = C) = \frac{n-m}{2n}.$$

In particular, the total variation distance between $X$ in the stationary distribution and $X$ conditioned on $Y_0 = (0,0)$ is at most $\frac{m}{n} \leq \frac{1}{10}$ using our assumption $m \leq n/10$. Thus, from initial state $(0,0)$, the random walk mixes to within $0.1$ total variation distance to the stationary distribution. We union bound with the $0.01$ probability of not reaching the stationary distribution in $O(1)$ steps to conclude the argument. $\qquad\square$

Thus, we can bound the relaxation time of $\mathbf{RW}_{m,\xi,i}$ as

$$t_{\text{rel}} \leq \frac{\tau(0.11)}{\log(1/0.22)} + 1 = O(1).$$

We now bound the mixing time from the initial distribution.

**Lemma F.10.** *Let $m \leq n/10$. Let $\gamma(x) \sim \mathrm{Poi}(1-\varepsilon) \otimes \mathrm{Poi}(1-\varepsilon)$, $\gamma(x) \sim \mathrm{Poi}\left(\frac{(2\varepsilon+\xi)m}{2(n-m)}\right) \otimes$ $\mathrm{Poi}\left(\frac{(2\varepsilon-\xi)m}{2(n-m)}\right)$, or $\gamma(x) \sim \mathrm{Poi}\left(\frac{(2\varepsilon-\xi)m}{2(n-m)}\right) \otimes \mathrm{Poi}\left(\frac{(2\varepsilon+\xi)m}{2(n-m)}\right)$ denote the initial distribution. The random walk $\mathbf{RW}_{m,\xi,i}$ has mixing time $\tau(\delta) = O(\log(n/\delta))$.*

*Proof.* As in Lemma E.8, we begin with following the inequality for any pair of states $x = (x_1, x_2)$ and $y = (y_1, y_2)$.

$$\left|\gamma(x)P^t(x,y) - \gamma(x)\pi(y)\right| \le \frac{\lambda_*^t \gamma(x)\sqrt{\pi(y)}}{\sqrt{\pi(x)}}.$$

We bound the ratio $\gamma(x)/\pi(x)$.

**Claim F.11.** *For all states $x = (x_1, x_2)$,*

$$\frac{\gamma(x)}{\pi(x)} \le \frac{n}{m}.$$

*Proof.* We split into the cases $\gamma \sim Z^A, \gamma \sim Z^B, \gamma \sim Z^C$ as defined in Lemma F.9. First, we write

$$\pi(x) = \frac{m}{n}\Pr(Z^A = x) + \frac{n-m}{2n}\Pr(Z^B = x) + \frac{n-m}{2n}\Pr(Z^C = x)$$

Then

$$\frac{\Pr(Z^A = x)}{\pi(x)} \le \frac{\Pr(Z^A = x)}{\frac{m}{n}\Pr(Z^A = x)} \le \frac{n}{m}$$

$$\frac{\Pr(Z^B = x)}{\pi(x)} \le \frac{\Pr(Z^B = x)}{\frac{n-m}{2n}\Pr(Z^B = x)} \le \frac{2n}{n-m} \le 4$$

$$\frac{\Pr(Z^C = x)}{\pi(x)} \le 4$$

where in the second and third cases we used $n - m \ge n/2$. Finally, we conclude by observing $\frac{n}{m} \ge 4$. $\square$

Then, we sum over $x$ to observe

$$\left|\left(\sum_x \gamma(x)P^t(x,y)\right) - \pi(y)\right| \le \lambda_*^t \sqrt{\frac{n}{m}\pi(y)} \sum_x \sqrt{\gamma(x)}.$$

We now bound $\sum_x \sqrt{\gamma(x)}$. In all three cases $Z^A, Z^B, Z^C$, we have that the Poisson distribution (in both distributions) has parameter $\lambda \le 1$. By standard Poisson concentration, for any $i \in \{1, 2\}$ and $D \in \{A, B, C\}$ we have

$$\Pr(Z_i^D > x) = \Pr(Z_i^D > \lambda + (x-1)) = e^{-\Omega(x-1)}.$$

Then, we bound for any $X \in \{A, B, C\}$

$$\sum_x \sqrt{\gamma(x)} = \sum_{x_1=0}^{\infty}\sum_{x_2=0}^{\infty} \sqrt{\Pr\left(Z_1^X = x_1\right)\Pr\left(Z_2^X = x_2\right)}$$

$$= \sum_{x_1=0}^{\infty} \sqrt{\Pr\left(Z_1^X = x_1\right)} \sum_{x_2=0}^{\infty} \sqrt{\Pr\left(Z_2^X = x_2\right)}$$

$$= O(1).$$

where the first equality follows by definition of $\gamma$ and independence of $\mathbf{p}, \mathbf{q}$, and the second and third equalities follow as $\sum_x \sqrt{\Pr(Z_i^X = x)}$ converges absolutely, which we showed in Lemma E.8. Thus, we arrive at the inequality

$$\left|\left(\sum_x \gamma(x)P^t(x,y)\right) - \pi(y)\right| = O\left(\lambda_*^t\sqrt{\frac{n}{m}\pi(y)}\right).$$

Summing over $y$ and applying a similar argument (see Lemma E.8 for details), we obtain

$$\sum_y \left|\left(\sum_x \gamma(x)P^t(x,y)\right) - \pi(y)\right| = O\left(\lambda_*^t\sqrt{\frac{n}{m}}\right).$$

Thus, since $\lambda_* < 1$ is an absolute constant less than 1, we conclude that from any of the three initial distributions, the random walk $\mathbf{RW}_{m,\xi,i}$ mixes to within $\delta$ of the stationary distribution in time $\tau(\delta) = O(\log((n/m)/\delta)) = O(\log(n/\delta))$.

$\square$

Now, using identical arguments as in Lemma E.10, we can conclude with the following lemma.

**Lemma F.12.** *Let $K = \tau(0.01)$ and $\xi \in [0, \varepsilon]$. Suppose $\mathcal{A}$ is $\frac{1}{10K}$-replicable with respect to $\mathcal{H}_C$. Then,*

$$\Pr_{(\mathbf{p},\mathbf{q}) \sim \mathcal{N}_\xi} \left( \left| \mathbb{E}_{T \sim (\mathbf{p},\mathbf{q})} \left[ \mathcal{A}(T) \right] - \mathbb{E}_{(\mathbf{p}_2,\mathbf{q}_2) \sim \mathcal{N}_\xi, T' \sim (\mathbf{p}_2,\mathbf{q}_2)} \left[ \mathcal{A}(T') \right] \right| > \frac{1}{4} \right) \le \frac{1}{2}.$$

We now prove Lemma F.6.

*Proof.* From Lemma F.10, we have that $\tau(0.01/n) = O(\log n))$. Since $\mathcal{A}$ is $\log^{-2} n$-replicable w.r.t. $\mathcal{H}_C$, it is also $1/(10K)$-replicable w.r.t. $\mathcal{H}_C$. The conclusion follows. $\square$

## F.3 Proof of Proposition F.2 and Theorem 1.4

Combining Lemma F.3 and Lemma F.6 then yields the proof of Proposition F.2.

*Proof of Proposition F.2.* The proof follows using analogous arguments as Proposition 3.4, applying Lemma F.3 and Lemma F.6 where appropriate. $\square$

We are ready to prove Theorem 1.4. The theorem follows from Proposition F.2 and Lemma D.3.

*Proof of Theorem 1.4.* Note that a lower bound of $\tilde{\Omega}(\sqrt{n}\varepsilon^{-2}\rho^{-1} + \varepsilon^{-2}\rho^{-2})$ follows immediately from lower bounds for uniformity testing and bias estimation respectively. It suffices to show a lower bound of $\tilde{\Omega}(n^{2/3}\varepsilon^{-4/3}\rho^{-2/3})$.

Proposition F.2 says that any deterministic tester that is 0.01-correct takes Poissonized samples with sample complexity $m = \tilde{o}(n^{2/3}\varepsilon^{-4/3}\rho^{-2/3})$ is not $\rho/(\log n)^2$-replicable with respect to the hard instance $\mathcal{H}_C$. Then, from Lemma D.3 we may conclude that any randomized tester with fixed sample complexity $m = \tilde{o}(n^{2/3}\varepsilon^{-4/3}\rho^{-2/3})$ is not $\rho/\mathrm{polylog}(n)$-replicable with respect to $\mathcal{H}_C$, concluding the proof. $\square$

