# OpenReview forum: "Replicable Distribution Testing"
_NeurIPS.cc/2025/Conference — NeurIPS 2025 spotlight_

### Official Review · Reviewer_6dkn · 2025-07-03

**Clarity:** 4
**Significance:** 4
**Originality:** 4
**Rating:** 6
**Confidence:** 2

**Summary:**

The focus is on replicable testing (uniformity, closeness, etc); that is we are interested in testing algorithms that with high probability output the same thing on two different independent draws (using the same randomness).

The main result of the paper is thm 1.3, setling down (up to polylog factors) the sample complexity of replicable uniformity testing, showing that the existing algorihm is in fact nearly optimal. As an aditional application of the developed techniques (which to some extent mimic the previous work up to a new random-walk argument), the authors provide lower bound on the sample complexity of closeness testing and give a (linear time) algorithm matching this lower bound up to polylog factor. The final interesting point of the paper I want to discuss here is the observation that replicable testing has suspiciously similar complexities with high probability testing and I like that it is emphasized as an open question.

**Questions:**

-

**Ethical Concerns:**

["NO or VERY MINOR ethics concerns only"]

**Final Justification:**

-

**Limitations:**

-

**Paper Formatting Concerns:**

-

**Quality:**

4

**Strengths And Weaknesses:**

The paper is very dense and was not that easy to read, but this is to be expected as it is very technical and I do not have suggestions in this regard. The studied problem/field is rather new and is somewhat niche, but I believe the replicability is an important property of testers and should be understood. To this end, It is likely that this paper will become a canonical reference.

Overall, I consider the presented results in the paper to be very strong and (unless there is an error that I missed, **I haven't checked technical details**) should be clearly accepted.


typos in alg 1, line "2: ".

---

> ### Author Rebuttal · Authors · 2025-07-31
>
> Thanks for the detailed response. We are encouraged by the positive comments about our tight sample complexity bounds for fundamental problems and the novelty of the random walk argument.

---

### Official Review · Reviewer_4xTX · 2025-07-03

**Clarity:** 4
**Significance:** 3
**Originality:** 3
**Rating:** 5
**Confidence:** 4

**Summary:**

This paper studies replicable property testing of discrete distributions, under the shared-randomness replicability definition proposed by Impagliazzo et al. Recent work of Liu and Ye studied the specific problem of replicable uniformity testing. This paper strengthens and generalizes their results in several dimensions. First, it introduces a framework for proving replicable distribution testing lower bounds, which gives nearly tight results for both uniformity and closeness testing. The framework takes the same conceptual approach as Liu and Ye, who were able to prove lower bounds for symmetric testers, but bypasses the need for this assumption by identifying and studying a particular random walk to show that an arbitrary tester behaves as though it were symmetric in the right way.

Second, it designs new replicable testers for closeness and independence testing. Building again on the work of Liu and Ye, it argues more generally that testers which involve thresholding a test statistic can be made replicable by randomizing the threshold. Getting this to work for independence testing involves some additional ideas for stabilizing the test statistic, which were originally introduced for differential privacy.

**Questions:**

* Does the lower bound framework give anything nontrivial for replicable independence testing?

**Ethical Concerns:**

["NO or VERY MINOR ethics concerns only"]

**Final Justification:**

This submission makes a solid technical contribution, and discussion with the authors did not raise anything to significantly change that impression.

**Limitations:**

Yes.

**Paper Formatting Concerns:**

None.

**Quality:**

4

**Strengths And Weaknesses:**

(+) Distribution testing offers an important and well-studied class of problems as fodder for identifying optimal replicable algorithms.

(+) We're still lacking strong / general-purpose techniques for proving replicability lower bounds, so the framework this paper offers (in particular, showing that the approach suggested by Liu and Ye does not require as strong assumptions about the structure of the learner as previously thought) is quite welcome.

(+) Both the positive and negative results identify broadly applicable principles for replicable distribution testing and algorithmic replicability more generally.

(+) The writing is clear and appears sound (though I did not check details carefully).

(-) The paper's novelty lies in its technical contributions (which are nice), but it doesn't offer much that is new conceptually.

---

> ### Author Rebuttal · Authors · 2025-07-31
>
> Thanks for the detailed response. We are encouraged by the positive comments about our tight sample complexity bounds for fundamental problems and the novelty of the random walk argument. In what follows we will address the concerns of the reviewer regarding the significance of our conceptual contribution and questions regarding related proofs.
>
> “doesn’t offer much new conceptually”
>
> While the closeness testing algorithm follows the general paradigm in replicable algorithm design of comparing a randomized threshold to a strongly concentrated test statistic, our independence tester is significantly more involved as known test statistics do not satisfy the required concentration bounds. Furthermore, our main technical contribution is the first unconditional lower bounds for replicable distribution testing (removing the previous work’s assumption that the given algorithm is symmetric and proving a new new-optimal lower bound for closeness testing). Our random walk argument to show that any algorithm’s output under a given family of distributions is well concentrated is novel in this context, specifically does not appear in prior work within this field.
>
> “lower bound framework … replicable independence testing”
>
> We believe that our lower bound framework likely yields non-trivial (and quite possibly optimal) lower bounds for independence testing. However, the hard instance for independence testing is much more complicated than those for uniformity and closeness. Thus, the computation of mutual information (to show similar families cannot be distinguished) and the analysis of the eigenvalues of the random walk (to show that the algorithm output is well concentrated within any given family) would be significantly more challenging. We believe that this is certainly an interesting direction for future research.

---

> > ### Comment · Reviewer_4xTX · 2025-08-06
> >
> > Thank you for addressing my comments. I have no further questions and maintain my positive impression of the submission.

---

### Official Review · Reviewer_hQvc · 2025-07-03

**Clarity:** 3
**Significance:** 4
**Originality:** 3
**Rating:** 5
**Confidence:** 3

**Summary:**

The paper studies the problem of distribution testing with replicability constraints. Given samples from k distributions, the distribution testing problem aims, in general, aims to design a "replicable" tester (algorithm) that distinguishes between the cases when all k distributions satisfy a certain property or if one of them is far from the property. This work in particular focuses on uniformity testing, closeness testing, and independence testing problems. These distribution testing problems without the replicability constraint has been well studied. The main results show tight tradeoffs in sample complexity with this added constraint.

**Questions:**

Please see the previous section for connections to tolerant testing and private testing.

**Ethical Concerns:**

["NO or VERY MINOR ethics concerns only"]

**Final Justification:**

I am convinced by the author responses, and recommend accepting the paper.

**Limitations:**

Yes

**Paper Formatting Concerns:**

Formatting is good.

**Quality:**

3

**Strengths And Weaknesses:**

The results are a good addition to the literature on distribution testing, and the techniques while similar to ref 27 (in the manuscript) require certain delicate balancing and are quite novel. The paper overall is quite well-written. The main technical differences from [27] while adequately addressed is a bit confusing at first without reading the definitions that follow in later sections. In my personal opinion, a slight rephrasing could help, but is not a major concern. Hence i would recommend accepting.

Connection to Tolerant Testing: It is however unclear why a tolerant tester that distinguishes between $\xi(\rho)$, and $\epsilon-\xi(\rho)$ for independence testing is not sufficient. At a high level, it seems intuitive that two independent sets of samples from the same distribution will be close, and hence from triangle inequality, should ensure that two induced distribution be either close or far from the property.

Connection to Private Testing: Another connection that has been leveraged, but not discussed is the connection to differentially private testing. This stems from the intuition that modifying even a single sample should not change the tester's output with high probability. Will the lower bounds of differentially private independence testing carry over to this setting?

---

> ### Author Rebuttal · Authors · 2025-07-31
>
> Thanks for the detailed response. We are encouraged by the positive comments about our tight sample complexity bounds for fundamental problems and the novelty of the random walk argument. In what follows we will address the concerns of the reviewer regarding the significance of our conceptual contribution and questions regarding related proofs.
>
> “similar to [27]”
>
> While the closeness testing algorithm follows the general paradigm in replicable algorithm design of comparing a randomized threshold to a strongly concentrated test statistic, our independence tester is significantly more involved as known test statistics do not satisfy the required concentration bounds. Furthermore, our main technical contribution is the first unconditional lower bounds for replicable distribution testing (removing the previous work’s assumption that the given algorithm is symmetric and proving a new new-optimal lower bound for closeness testing). Our random walk argument to show that any algorithm’s output under a given family of distributions is well concentrated is novel in this context, specifically does not appear in prior work within this field.
>
> Connections to Tolerant Testing
>
> While tolerant testers do imply replicable testers, our algorithms are significantly more sample efficient than any tester obtained in this way (see also discussion in [27]). For example, a natural approach to a replicable test is the following: sample a random threshold r between 0 and eps, and run a tolerant tester with thresholds r - \rho \eps, r + \rho \eps. Even for constant tolerance, the sample complexity is barely sub-linear in the domain size n. Thus, even for constant \rho, and \eps the sample complexity of a replicable tester will be barely sublinear in n. In contrast, all our algorithms have strongly sub-linear dependency on the domain size parameter.
>
> Connections to Private Testing
>
> While m-sample private testers give rise to m^2-sample replicable testers [12] the sample complexity is suboptimal by a polynomial factor (as discussed in line 52), and in fact linear in n. We do mention that our algorithm for independence testing is inspired by the randomized flattening technique that has been leveraged for designing private independence testing algorithms (see line 173).
>
> On the lower bound side, while private testing lower bounds imply replicable testing lower bounds [12], they will not give any non-trivial lower bounds (beyond standard lower bounds for uniformity testing). This is due to the fact that the known equivalence [12] will only give lower bounds for constant replicability \rho. (In particular, they show a constant replicability algorithm already implies a private algorithm).
>
> Although the current connection is not strong enough to obtain sample-optimal bounds, the connection between replicable and differentially private testing remains an interesting one, and it is possible that a stronger quantitative relationship than what was proven in [12] would allow us to obtain tight bounds.

---

> > ### Comment · Reviewer_hQvc · 2025-08-06
> >
> > Thanks for the clarifications. The paper makes an interesting read, and the results open up some interesting questions for future works. I will retain my recommendation to accept.

---

### Official Review · Reviewer_E3L9 · 2025-07-10

**Clarity:** 2
**Significance:** 3
**Originality:** 3
**Rating:** 5
**Confidence:** 3

**Summary:**

Given sample access to a distribution, property testing is the problem of determining if the distribution has property $P$ or is far from any distribution with probability $P$. An algorithm is replicable if repeated runs of the algorithm give the same answer whp. In this case, that corresonds to consistently classifying a particular distribution the same way whp, regardless of the exact set of samples you get for the distribution. The paper gives upper and lower bounds for replicable property-testing algorithms for a couple of key properties, including tight bounds for closeness testing and n-item uniformity testing.

**Questions:**

There's an indirectly related proof of lower bounds for replicability in https://arxiv.org/pdf/2304.03757, https://arxiv.org/pdf/2503.15294 and https://arxiv.org/pdf/2304.02240v1 Both show roughly that you can't replicably distinguish between things if things more or less continuously change into each other. There's a follow-up with sample complexity stuff here https://openreview.net/pdf?id=2lL7s5ESTj.
Those papers required that there is a set $S$ of size $k$ s.t. $A[samples, randomness]\inS$ whp, which is slightly different but related to the constraint that $A[sample1, randomness] = A[sample2,same randomness]$ whp. But I think it could simplify your lower bound proofs.

From what I can tell, your main technical contribution is the random walk analysis to establish the space of similar distributions. Is that fair to say?

**Ethical Concerns:**

["NO or VERY MINOR ethics concerns only"]

**Final Justification:**

I still think that the sample complexity bounds are interesting because they are on fundamental problems for replicability. The random walk argument continues to be neat. I still have not verified the proofs, which are still extremely long and technical

The authors and other reviewers have convinced me that the paper is more original than I thought. Therefore I have updated my Originality score.

**Limitations:**

Yes

**Quality:**

3

**Strengths And Weaknesses:**

+ Tight sample complexity bounds for replicability for some fundamental problems
+ Neat random walk argument to establish that on similar distributions the tester will behave similarly

? I did not verify the proofs, which are extremely long and technical

---

> ### Author Rebuttal · Authors · 2025-07-31
>
> Thanks for the detailed response. We are encouraged by the positive comments about our tight sample complexity bounds for fundamental problems and the novelty of the random walk argument. In what follows we will address the concerns of the reviewer regarding the significance of our conceptual contribution and questions regarding related proofs.
>
> Mostly an improvement of existing arguments and algorithms using known techniques
>
> While the closeness testing algorithm follows the general paradigm in replicable algorithm design of comparing a randomized threshold to a strongly concentrated test statistic, our independence tester is significantly more involved as known test statistics do not satisfy the required concentration bounds. Furthermore, our main technical contribution is the first unconditional lower bounds for replicable distribution testing (removing the previous work’s assumption that the given algorithm is symmetric and proving a new new-optimal lower bound for closeness testing). Our random walk argument to show that any algorithm’s output under a given family of distributions is well concentrated is novel in this context, specifically does not appear in prior work within this field.
>
> “can’t distinguish things… continuously change”
>
> While our lower bound relies on the general framework that similar distributions cannot be distinguished (also observed e.g. for mean estimation [33, 34]), the hard instances for distribution testing consist of similar families of distributions (\mathcal{M}_{\xi} in the text) instead of single distributions. A barrier from previous work (which we resolve) is to show that the algorithm behaves similarly among all distributions in a particular family.
>
> Connections to List Replicability (Randomness and Sample Complexity)
>
> Finally, list replicability does not seem to have any immediate implications for distribution testing as there is a 0-sample algorithm that is replicable with list size 2 (output both {0, 1}).

---

> > ### Comment · Reviewer_E3L9 · 2025-08-08
> >
> > Sorry, I was trying to sketch out my idea in more detail and then I got busy.
> > Either way, I've changed my score to "accept", removed my statement about known techniques, and adjusted my scores.

---

> > ### Comment · Reviewer_E3L9 · 2025-08-08
> >
> > I think there still may be a connection, but it's certainly not an obvious one.
> >
> > I have been convinced that your techniques are more original than I thought initially, and I have updated my score to match.

---

### Decision · Program_Chairs · 2025-09-17

**Decision:**

Accept (spotlight)

**Comment:**

This manuscript applies replicability in learning introduced by Impagliazzo et al @ SOTC'22 to testing properties of distributions and  answers the open question related to replicable uniformity testing posed in Liu & Ye @ NeurIPS'24. While the proofs in this work are quite long and technical [E3L9] and there are no conceptual novelties [4xTX], the work is considered a good addition to the literature on distribution testing [hQvc, 4xTX] as it clarifies technical differences from prior work [hQvc] and offers technical novelties [4xTX] and results are perceived as both relevant and elegant [E3L9].